# Understanding and Extending Subgraph GNNs by Rethinking Their Symmetries

**Fabrizio Frasca**[*]
Imperial College London & Twitter
ffrasca@twitter.com

**Beatrice Bevilacqua**[*]
Purdue University
bbevilac@purdue.edu

**Michael M. Bronstein**
University of Oxford & Twitter
mbronstein@twitter.com

**Haggai Maron**
NVIDIA Research
hmaron@nvidia.com

## Abstract

Subgraph GNNs are a recent class of expressive Graph Neural Networks (GNNs) which model graphs as collections of subgraphs. So far, the design space of possible Subgraph GNN architectures as well as their basic theoretical properties are still largely unexplored. In this paper, we study the most prominent form of subgraph methods, which employs node-based subgraph selection policies such as ego-networks or node marking and deletion. We address two central questions: (1) *What is the upper-bound of the expressive power of these methods?* and (2) *What is the family of equivariant message passing layers on these sets of subgraphs?*. Our first step in answering these questions is a novel symmetry analysis which shows that modelling the symmetries of node-based subgraph collections requires a significantly smaller symmetry group than the one adopted in previous works. This analysis is then used to establish a link between Subgraph GNNs and Invariant Graph Networks (IGNs). We answer the questions above by first bounding the expressive power of subgraph methods by 3-WL, and then proposing a general family of message-passing layers for subgraph methods that generalises all previous node-based Subgraph GNNs. Finally, we design a novel Subgraph GNN dubbed SUN, which theoretically unifies previous architectures while providing better empirical performance on multiple benchmarks.

## 1 Introduction

Message Passing Neural Networks (MPNNs) are arguably the most commonly used version of Graph Neural Networks (GNNs). The limited expressive power of MPNNs [36, 55] has led to a plethora of works aimed at designing expressive GNNs while maintaining the simplicity and scalability of MPNNs [11, 39, 49, 30]. Several recent studies have proposed a new class of such architectures [14, 59, 7, 61, 43, 42], dubbed *Subgraph GNNs*, which apply MPNNs to collections ('bags') of subgraphs extracted from the original input graph and then aggregate the resulting representations. Subgraphs are selected according to a predefined policy; in the most popular ones, each subgraph is tied to a specific node in the original graph, for example by deleting it or extracting its local ego-network. Subgraph GNNs have demonstrated outstanding empirical performance, with state-of-the-art results on popular benchmarks like the ZINC molecular property prediction [61, 7].

While offering great promise, it is fair to say that we still lack a full understanding of Subgraph GNNs. Firstly, on the theoretical side, it is known that subgraph methods are strictly stronger than

---

[*]Equal contribution. Author ordering determined by coin flip.

36th Conference on Neural Information Processing Systems (NeurIPS 2022).

the Weisfeiler-Leman (WL) test [54, 39], but an upper-bound on their expressive power is generally unknown. Secondly, on a more practical level, Subgraph GNN architectures differ considerably in the way information is aggregated and shared across the subgraphs, and an understanding of the possible aggregation and sharing rules is missing. Both aspects are important: an understanding of the former can highlight the limitations of emerging architectures, a study of the latter paves the way for improved Subgraph GNNs.

**Main contributions.** The goal of this paper is to provide a deeper understanding of node-based Subgraph GNNs in light of the two aforementioned aspects. The main theoretical tool underpinning our contributions is a novel analysis of the symmetry group that acts on the sets of subgraphs. While several previous approaches [43, 14, 7] have (often implicitly) assumed that a subgraph architecture should be equivariant to independent node and subgraph permutations, we leverage the fact that node-based policies induce an inherent bijection between the subgraphs and the nodes. This observation allows us to align the two groups and model the symmetry with a single (smaller) permutation group that acts on nodes and subgraphs *jointly*. Other works [61, 56, 59] have (again, implicitly) recognised such node-subgraph correspondence but without studying the implications on the symmetry group, and resorting, as a result, to a partial and heuristic choice of equivariant operations.

The use of this stricter symmetry group raises a fruitful connection with $k$-order Invariant Graph Networks (`k-IGNs`) [33, 32], a well studied family of architectures for processing graphs and hypergraphs designed to be equivariant to the same symmetry group. This connection allows us to transfer and reinterpret previous results on `IGNs` to our Subgraph GNN setup. As our first contribution we show that the expressive power of Subgraph GNNs with node-based policies is bounded by that of the `3-WL` test. This is shown by proving that all previous Subgraph GNNs can be implemented by a `3-IGN` and by leveraging the fact that the expressive power of these models is bounded by `3-WL` [21, 5].

Our second contribution is the proposal of a general layer formulation for Subgraph GNNs, based on the observation that these methods maintain an $n \times n$ representation of $n$ subgraphs with $n$ nodes, following the same symmetry structure of `2-IGNs` (same permutation applied to both rows and columns of this representation). We propose a novel extension of `2-IGNs` capturing both local (message-passing-like) and global operations. This extension easily recovers previous methods facilitating their comparison. Also, we present a number of new operations that previous methods did not implement. We build upon these observations to devise a new Subgraph GNN dubbed `SUN`, (*Subgraph Union Network*). We prove that `SUN` generalises all previous node-based Subgraph GNNs and we empirically compare it to these methods, showing it can outperform them.

## 2   Previous and related work

**Expressive power of GNNs.** The expressive power of GNNs is a central research focus since it was realised that message-passing type GNNs are constrained by the expressivity of the WL isomorphism test [36, 55]. Other than the aforementioned subgraph-based methods, numerous approaches for more powerful GNNs have been proposed, including positional and structural encodings [1, 45, 11, 17, 28, 31], higher-order message-passing schemes [36, 38, 10, 9], equivariant models [24, 33, 32, 53, 15, 51, 40]. We refer readers to the recent survey by Morris et al. [39] for additional details. Finally we note that, in a related and concurrent work, Qian et al. [46] propose a theoretical framework to study the expressive power of subgraph-based GNNs by relating them to the `k-WL` hierarchy, and explore how to sample subgraphs in a data-driven fashion.

**Invariant graph networks.** `IGNs` were recently introduced in a series of works by Maron et al. [33, 32, 34] as an alternative to MPNNs for processing graph and hyper-graph data. For $k \geq 2$, `k-IGNs` represent hyper-graphs with hyper-edges up to size $k$ with $k$-order tensor $\mathcal{Y} \in \mathbb{R}^{n^k}$, where each entry holds information about a specific hyper-edge. On these they apply linear $S_n$-equivariant layers interspersed with pointwise nonlinearities. These models have been thoroughly studied in terms of: (i) their expressive power; (ii) the space of their equivariant linear layers. As for (i), `IGNs` were shown to have exactly the same graph separation power as the `k-WL` graph isomorphism test [32, 5, 21] and, for sufficiently large $k$, to have a universal approximation property w.r.t. $S_n$-invariant and equivariant functions [34, 26, 47]. Concerning (ii), the work in [33] completely characterised the space of linear layers equivariant to $S_n$ from $\mathbb{R}^{n^k}$ to $\mathbb{R}^{n^{k'}}$: the authors derived a basis of $\text{bell}(k + k')$ linear operators consisting of indicator tensors of equality patterns over the multi-index set $\{1, \ldots, n\}^{k+k'} = [n]^{k+k'}$. Albooyeh et al. [2] showed these layers can be (re-)written as sums of pooling-broadcasting operations

between elements of $\mathcal{Y}$ indexed by the orbits [2] of the action of $S_n$ on $[n]^k$ and $[n]^{k'}$. Take, e.g., $k = k' = 2$. In this case there are only two orbits: $\{i, i\}$, $i \in [n]$ corresponding to on-diagonal terms, and $\{i, j\}$, $i \neq j \in [n]$, off-diagonal terms. According to Albooyeh et al. [2] any equivariant linear layer $L : \mathbb{R}^{n^2} \to \mathbb{R}^{n^2}$ can be represented as a composition of pooling and broadcasting operations on the elements indexed by these orbits. One example is the linear map that sums the on-diagonal elements and broadcasts the result to the off-diagonal ones: $L(\mathcal{Y})_{ij} = \sum_k \mathcal{Y}_{kk}$ for $i \neq j$, 0 otherwise. See Appendix B, for additional details. These results particularly important as they underpin most of our theoretical derivations. Lastly, a more comprehensive coverage of IGNs can be found in [39].

**Subgraph GNNs.** Despite motivated by diverse premises, a collection of concurrent methods share the overarching design whereby graphs are modelled through the application of a GNN to their subgraphs. Bevilacqua et al. [7] first explicitly formulated the concept of bags of subgraphs generated by a predefined policy and studied layers to process them in an equivariant manner: the same GNN can encode each subgraph independently (DS-GNN), or information can be shared between these computations in view of the alignment of nodes across the bag [35] (DSS-GNN). Building upon the Reconstruction Conjecture [25, 52], Reconstruction GNNs [14] obtain node-deleted subgraphs, process them with a GNN and then aggregate the resulting representations by means of a set model. Nested GNNs [59] and GNN-As-Kernel models (GNN-AK) [61] shift their computation from rooted subtrees to rooted subgraphs, effectively representing nodes by means of GNNs applied to their enclosing ego-networks. Similarly to DSS-GNNs [7], GNN-AK models may feature information sharing modules aggregating node representations across subgraphs. ID-GNNs [56] also process ego-network subgraphs, but their roots are 'marked' so to specifically alter the exchange of messages involving them. Intuitively, the use of subgraphs implicitly breaks those local symmetries which determine the notorious expressiveness bottleneck of MPNNs. We note that other works can be interpreted as Subgraph GNNs, including those by Papp et al. [43], Papp and Wattenhofer [42].

# 3 Node-based Subgraph GNNs

**Notation.** Let $G = (A, X)$ be a member of the family $\mathcal{G}$ of *node-attributed*, undirected, finite, simple graphs[3]. The *adjacency matrix* $A \in \mathbb{R}^{n \times n}$ represents $G$'s edge set $E$ over its set of $n$ nodes $V$. The *feature matrix* $X \in \mathbb{R}^{n \times d}$ gathers the node features; we denote by $x_j \in \mathbb{R}^{d \times 1}$ the features of node $j$ corresponding to the $j$-th row of $X$. $B_G$ is used to denote a multiset (bag) of $m$ subgraphs of $G$. Adjacency and feature matrices for subgraphs in $B_G$ are arranged in tensors $\mathcal{A} \in \mathbb{R}^{m \times n \times n}$ and $\mathcal{X} \in \mathbb{R}^{m \times n \times d}$. Superscript $^{i,(t)}$ refers to representations on subgraph $i$ at the $t$-th layer of a stacking, as in $x_j^{i,(t)}$. Finally, we denote $[n] = \{1, \ldots, n\}$. All proofs are deferred to Appendices B and D.

**Formalising Subgraph GNNs.** Subgraph GNNs compute a representation of $G \in \mathcal{G}$ as

$$(A, X) \mapsto \big(\mu \circ \rho \circ \mathcal{S} \circ \pi\big)(A, X). \tag{1}$$

Here, $\pi : G \mapsto \{G^1, ..., G^m\} = \{(A^1, X^1), ..., (A^m, X^m)\} = B_G^{(0)}$ is a *selection policy* generating a bag of subgraphs from $G$; $\mathcal{S} = L_T \circ \ldots \circ L_1 : B_G^{(0)} \mapsto B_G^{(T)}$ is a *stacking* of $T$ (node- and subgraph-) permutation equivariant layers; $\rho : (G, B_G^{(T)}) \mapsto x_G$ is a permutation invariant *pooling function*, $\mu$ is an MLP. The layers in $\mathcal{S}$ comprise a *base-encoder* in the form of a GNN applied to subgraphs; throughout this paper, we assume it to be a 1-WL maximally expressive MPNN such as the one in Morris et al. [36]. Subgraph GNNs differ in the implementation of $\pi$, $\mathcal{S}$ and, in some cases, $\rho$. For example, in (n-1)-Reconstruction GNNs [14], $\pi$ selects node-deleted subgraphs and $\mathcal{S}$ applies a Siamese MPNN to each subgraph independently. To exemplify the variability in $\mathcal{S}$, DSS-GNN [7] extends this method with cross-subgraph node and connectivity aggregation. More details are on how currently known Subgraph GNNs are captured by Equation (1) can be found in Appendix A.

**Node-based selection policies.** In this work, we focus on a specific family of *node-based* subgraph selection policies, wherein every subgraph is associated with a unique node in the graph. Formally, we call a subgraph selection policy *node-based* if it is of the form $\pi(G) = \{f(G, v)\}_{v \in V}$, for some *selection function* $f(G, v)$ that takes a graph $G$ and a node $v$ as inputs and outputs a subgraph $G^v$. In the following, we refer to $v$ as the *root* of subgraph $G^v$. We require $f$ to be a bijection and we

---

[2]For group $G$ acting on set $X$, the orbits of the action of $G$ on $X$ are defined as $\{G \cdot x \mid x \in X\}$. These *partition* $X$ into subsets whose elements can (only) reach all other elements in the subset via the group action.

[3]We do not consider edge features, although an extension to such a setting would be possible.

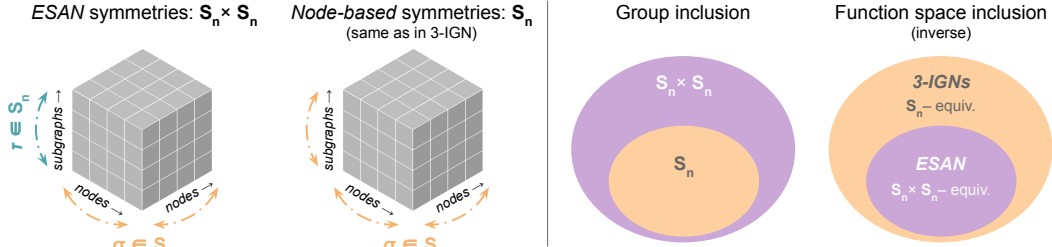

Figure 1: Symmetries of bags of subgraphs (left) and corresponding function space diagrams (right). In ESAN [7] symmetries are modelled as a direct product of node and subgraph permutation groups; however, node-based policies enable the use of one single permutation group, the same as in `3-IGNs`. `3-IGNs` are less constrained, thus more expressive than ESAN and other Subgraph GNNs. See diagram on the right and formal statement in Section 5.

note that such policies produce $m = n$ different subgraphs. Amongst the most common examples are *node-deletion* (ND), *node-marking* (NM), and *ego-networks* (EGO) policies. For input graph $G$, $f_{\text{ND}}(G, v)$ removes node $v$ and the associated connectivity; $f_{\text{NM}}(G, v)$ adds a special 'mark' attribute to $v$'s features (with no connectivity alterations), and $f_{\text{EGO}(h)}(G, v)$ returns the subgraph induced by the $h$-hop-neighbourhood around the root $v$. EGO policies can be 'marked': $f_{\text{EGO}+(h)}(G, v)$ extracts the $h$-hop ego-net around $v$ *and* marks this node as done by $f_{\text{NM}}$. For convenience, we denote the class of such node-based selection policies by $\Pi$:

**Definition 1** (Known node-based selection policies $\Pi$). *Let $\Sigma$ be the set of all node-based subgraph selection policies operating on $\mathcal{G}$. Class $\Pi \subset \Sigma$ collects the node-based policies node-deletion (ND), node-marking (NM), ego-nets (EGO) and marked ego-nets (EGO+) of any depth:* $\Pi = \{\pi_{\text{ND}}, \pi_{\text{NM}}, \pi_{\text{EGO}(h)}, \pi_{\text{EGO}+(h)} \,|\, h > 0\}$.

**Node-based Subgraph GNNs** are those Subgraph GNNs which, implicitly or explicitly, process bags generated by node-based policies. We group known formulations in the following family:

**Definition 2** (Known node-based Subgraph GNNs $\Upsilon$). *Let $\Xi$ be the set of all node-based Subgraph GNNs. Class $\Upsilon \subset \Xi$ collects known Subgraph GNNs when equipped with `1-WL` base-encoders:* $\Upsilon = \{\text{(n-1)-Reconstr.GNN, GNN-AK, GNN-AK-ctx, NGNN, ID-GNN, DS-GNN}_{\Pi}, \text{DSS-GNN}_{\Pi}\}$. DS-GNN$_{\Pi}$, DSS-GNN$_{\Pi}$ *refer to* DS- *and* DSS-GNN *models equipped with any* $\pi \in \Pi$.

Importantly, all these methods apply MPNNs to subgraphs of the original graph, but differ in the way information is shared between subgraphs/nodes. In all cases, their expressive power is strictly larger than `1-WL`, but an *upper*-bound is currently unknown.

## 4 Symmetries of node-based subgraph selection policies

In an effort to characterise the representational power of node-based Subgraph GNNs, we first study the symmetry group of the objects they process: 'bags of subgraphs' represented as tensors $(\mathcal{A}, \mathcal{X}) \in \mathbb{R}^{m \times n \times n} \times \mathbb{R}^{m \times n \times d}$, assuming $n$ nodes across $m$ subgraphs. Previous approaches [14, 7, 43] used two permutation groups: one copy of the symmetric group $S_n$ models *node permutations*, while another copy $S_m$ models *subgraph permutations* in the bag. These two were combined by a group product[4] acting *independently* on the nodes and subgraphs in $(\mathcal{A}, \mathcal{X})$. For example, Bevilacqua et al. [7] model the symmetry as:

$$((\tau, \sigma) \cdot \mathcal{A})_{ijk} = \mathcal{A}_{\tau^{-1}(i)\sigma^{-1}(j)\sigma^{-1}(k)}, \quad ((\tau, \sigma) \cdot \mathcal{X})_{ijl} = \mathcal{X}_{\tau^{-1}(i)\sigma^{-1}(j)l}, \quad (\tau, \sigma) \in S_m \times S_n \quad (2)$$

Our contributions stem from the following crucial observation: When using node-based policies, the subgraphs in $(\mathcal{A}, \mathcal{X})$ can be *ordered consistently with the nodes* by leveraging the bijection $f : v \mapsto G_v$ characterising this policy class. In other words, $f$ suggests a node-subgraph alignment inducing a new structure on $(\mathcal{A}, \mathcal{X})$, whereby the subgraph order is not independent of that of nodes

---

[4]Bevilacqua et al. [7] use a direct-product, assuming nodes in subgraphs are consistently ordered. Cotta et al. [14] use the larger wreath-product assuming node ordering in the subgraph is unknown.

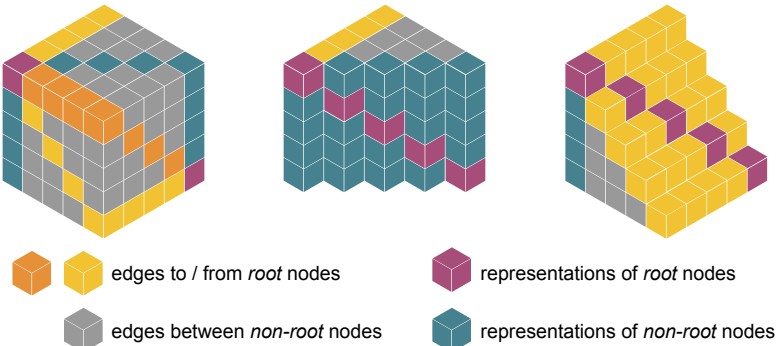

Figure 2: Depiction of cubed tensor $\mathcal{Y}$, its orbit-induced partitioning and the related semantics when $\mathcal{Y}$ is interpreted as a bag of node-based subgraphs, $n = 5$. Elements in the same partition are depicted with the same colour. Left: the whole tensor. Middle and right: sections; elements in purple and green constitute sub-tensor $\mathcal{X}$, the remaining ones sub-tensor $\mathcal{A}$.

anymore. Importantly, this new structure is preserved *only* by those permutations operating identically on both nodes and subgraphs. Following this observation, the symmetry of a node-based bag of subgraphs is modelled more accurately using only *one single permutation group* $S_n$ jointly acting on both nodes and subgraphs:

$$(\sigma \cdot \mathcal{A})_{ijk} = \mathcal{A}_{\sigma^{-1}(i)\sigma^{-1}(j)\sigma^{-1}(k)}, \quad (\sigma \cdot \mathcal{X})_{ijl} = \mathcal{X}_{\sigma^{-1}(i)\sigma^{-1}(j)l}, \qquad \sigma \in S_n \tag{3}$$

It should be noted that $S_n$ is *significantly smaller* than $S_n \times S_n$[5]. Informally, the latter group contains many permutations which are not in the former: those acting differently on nodes and subgraphs and, thus, not preserving the new structure of $(\mathcal{A}, \mathcal{X})$. Since they are restricted by a smaller set of equivariance constraints, we expect GNNs designed to be equivariant to $S_n$ to be be more expressive than those equivariant to the larger groups considered by previous works [34] (see Figure 1).

The insight we obtain from Equation (3) is profound: it reveals that the symmetry structure of $\mathcal{A}$ exactly matches the symmetries of third-order tensors used by `3-IGNs`, and similarly, that the symmetry structure for $\mathcal{X}$ matches the symmetries of second-order tensors used by `2-IGNs`. In the following, we will make use of this insight and the fact that `IGNs` are well-studied objects to prove an upper-bound on the expressive power of Subgraph GNNs and to design principled extensions to these models. We remark that bags of node-based subgraphs can also be represented as tensors $\mathcal{Y} \in \mathbb{R}^{n^3 \times d}$, the same objects on which `3-IGNs` operate. Here, $\mathcal{X}$ is embedded in the main diagonal plane of $\mathcal{Y}$, $\mathcal{A}$ in its remaining entries. Within this context, it is informative to study the semantics of the 5 orbits induced by the action of $S_n$ on $\mathcal{Y}$'s multi-index set $[n]^3$: each of these uniquely identify root nodes, non-root nodes, edges to and from root nodes as well as edges between non-root nodes (see Figure 2 and additional details in Appendix B.1.1). We build upon this observation, along with the layer construction by Albooyeh et al. [2], to prove many of the results presented in the following.

## 5 A representational bound for Subgraph GNNs

In this section we prove that the expressive power of known node-based Subgraph GNNs is bounded by `3-WL` by showing that they can be implemented by `3-IGNs`, which have the same expressive power as `3-WL`. Underpinning the possibility of `IGNs` to upper-bound a certain Subgraph GNN $\mathcal{N}$ in its expressive power is the ability of `IGNs` to (i) implement $\mathcal{N}$'s subgraph selection policy ($\pi$) and (ii) implement $\mathcal{N}$'s (generalised) message-passing and pooling equations ($\mu \circ \rho \circ \mathcal{S}$). This would ensure that whenever $\mathcal{N}$ assigns distinct representations to two non-isomorphic graphs, an `IGN` implementing $\mathcal{N}$ would do the same. We start by introducing a recurring, useful concept.

**Definition 3** ("implements"). *Let $f : D_f \to C_f$, $g : D_g \to C_g$ be two functions and such that $D_g \subseteq D_f, C_g \subseteq C_f$. We say $f$ implements $g$ (and write $f \cong g$) when $\forall x \in D_g, f(x) = g(x)$.*

---

[5]More formally, $S_n$'s orbits on the indices in (3) refine the orbits of the product group in (2).

Our first result shows that 3-IGNs can implement the selection policies in class $\Pi$ (Definition 1), which, to the best of our knowledge, represent *all known* node-based policies utilised by previously proposed Subgraph GNNs.

**Lemma 4** (3-IGNs implement known node-based selection policies). *For any $\pi \in \Pi$ there exists a stacking of 3-IGN layers $\mathcal{M}_\pi$ s.t. $\mathcal{M}_\pi \cong \pi$.*

Intuitively, 3-IGNs start from a $\mathbb{R}^{n^2}$ representation of $G$ and, first, move to a $\mathbb{R}^{n^3}$ tensor 'copying' this latter along its first (subgraph) dimension. This is realised via an appropriate broadcast operation. Then, they proceed by adding a 'mark' to the features of some nodes and/or by nullifying elements corresponding to some edges. We refer readers to Figure 2 and Appendix B.1.2 for additional details on how nodes in each subgraph are represented in 3-IGNs. Next, we show 3-IGNs can implement layers of any model $\in \Upsilon$.

**Lemma 5** (3-IGNs implement Subgraph GNN layers). *Let $G_1, G_2$ be two graphs in $\mathcal{G}$ and $\mathcal{N}$ a model in family $\Upsilon$ equipped with Morris et al. [36] message-passing base-encoders. Let $B_1^{(t)}, B_2^{(t)}$ be bags of subgraphs in the input of some intermediate layer $L$ in $\mathcal{N}$. Then there exists a stacking of 3-IGN layers $\mathcal{M}_L$ for which $\mathcal{M}_L(B_i^{(t)}) = B_i^{(t+1)} = L(B_i^{(t)})$ for $i = 1, 2$.*

Lemmas 4 and 5 allow us to upper-bound the expressive power of all known instances of node-based Subgraph GNNs by that of 3-IGNs:

**Theorem 6** (3-IGNs upper-bound node-based Subgraph GNNs). *For any pair of non-isomorphic graphs $G_1, G_2$ in family $\mathcal{G}$ and Subgraph GNN model $\mathcal{N} \in \Upsilon$ equipped with Morris et al. [36] message-passing base-encoders, if there exists weights $\Theta$ such that $G_1$, $G_2$ are distinguished by instance $\mathcal{N}_\Theta$, then there exist weights $\Omega$ for a 3-IGN instance $\mathcal{M}_\Omega$ such that $G_1, G_2$ are distinguished by $\mathcal{M}_\Omega$ as well.*

Theorem 6 has profound consequences in the characterisation of the expressive power of node-based Subgraph GNNs, as we show in the following

**Corollary 7** (3-WL upper-bounds node-based Subgraph GNNs). *Let $G_1, G_2 \in \mathcal{G}$ be two non-isomorphic graphs and $\mathcal{N}_\Theta \in \Upsilon$ one instance of model $\mathcal{N}$ with weights $\Theta$. If $\mathcal{N}_\Theta$ distinguishes $G_1, G_2$, then the 3-WL algorithm does so as well.*

*Proof idea*: If there is a pair of graphs undistinguishable by 3-WL, but for which there exists a Subgraph GNN separating them, there must exists a 3-IGN separating these (Theorem 6). This is in contradiction with the result by Geerts [21], Azizian and Lelarge [5][6].

# 6 A design space for Subgraph GNNs

As discussed, different formulations of Subgraph GNNs differ primarily in the specific rules for updating node representations across subgraphs. However, until now it is not clear whether existing rules exhaust all the possible equivariant options. We devote this section to a systematic characterisation of the 'layer space' of Subgraph GNNs.

In the spirit of the previous Section 5, where we "embedded" Subgraph GNNs in 3-IGNs, one option would be to consider all bell$(6) = 203$ linear equivariant operations prescribed by this formalism. However, this choice would be problematic for three main reasons: (i) This layer space is *too vast* to be conveniently explored; (ii) It includes operations involving $\mathcal{O}(n^3)$ space complexity, impractical in most applications; (iii) The linear IGN basis does not directly support local message passing, a key operation in subgraph methods. Following previous Subgraph GNN variants, which use $\mathcal{O}(n^2)$ storage for the representation of $n$ nodes in $n$ subgraphs, we set the desideratum of $\mathcal{O}(n^2)$ memory complexity as our main constraint, and use this restriction to reduce the design space. Precisely, we are interested in modelling $S_n$-equivariant transformations on the subgraph-node tensor $\mathcal{X}$.

## 6.1 Extended 2-IGNs

As we have already observed in Equation 3 in Section 4, such a second order tensor $\mathcal{X}$ abides by the same symmetry structure of 2-IGNs. We therefore gain intuition from the characterisation of linear equivariant mappings as introduced by Maron et al. [33], and propose an extension of this formalism.

---

[6]k-WL is equivalent to (k-1)-FWL, i.e. the "Folklore" WL test, see Morris et al. [36].

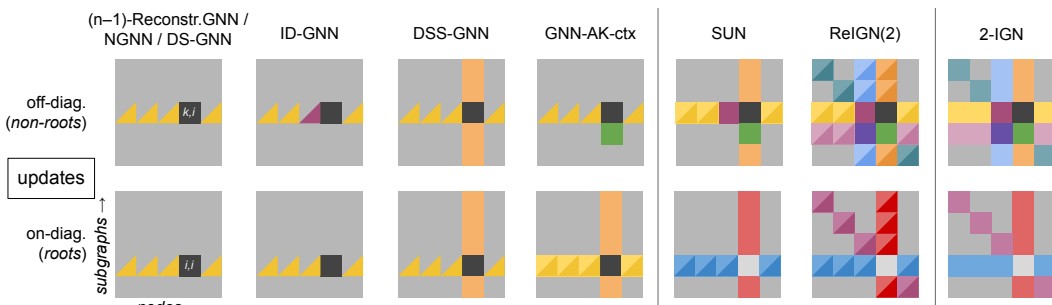

Figure 3: Comparison of aggregation and update rules in Subgraph GNNs, illustrated on an $n \times n$ matrix ($n$ subgraphs with $n$ nodes). Top row: off-diagonal updates; bottom row: diagonal (root node) updates. Each colour represents a different parameter. Full squares: global sum pooling; triangles: local pooling; two triangles: both local and global pooling. See Appendix C for more details.

`2-IGN` **layer space.** A `2-IGN` layer $L_\Theta$ updates $\mathcal{X} \in \mathbb{R}^{n \times n \times d}$ as $\mathcal{X}^{(t+1)} = L_\Theta\big(\mathcal{X}^{(t)}\big)$ by applying a specific transformation to on- ($x_i^i$) and off-diagonal terms ($x_j^i, i \neq j$):

$$x_i^{i,(t+1)} = \upsilon_{\theta_1}\big(x_i^{i,(t)}, \underset{j}{\square}\, x_j^{j,(t)}, \underset{j \neq i}{\square}\, x_j^{i,(t)}, \underset{h \neq i}{\square}\, x_i^{h,(t)}, \underset{h \neq j}{\square}\, x_j^{h,(t)}\big) \tag{4}$$

$$x_i^{k,(t+1)} = \upsilon_{\theta_2}\big(x_i^{k,(t)}, x_k^{i,(t)}, \underset{h \neq j}{\square}\, x_j^{h,(t)}, \underset{h \neq i}{\square}\, x_i^{h,(t)}, \underset{j \neq k}{\square}\, x_j^{k,(t)}, \underset{j \neq i}{\square}\, x_j^{i,(t)}, \underset{h \neq k}{\square}\, x_k^{h,(t)}, x_k^{k,(t)}, x_i^{i,(t)}, \underset{j}{\square}\, x_j^{j,(t)}\big)$$

Here, $\square$ indicates a permutation invariant aggregation function, $\upsilon_{\theta_1}, \upsilon_{\theta_2}$ apply a specific $d \times d'$ linear transformation to each input term and sum the outputs including bias terms.

`ReIGN(2)` **layer space.** As `2-IGN` layers are linear, the authors advocate setting $\square \equiv \sum$, performing pooling as *global* summation. Here, we extend this formulation to additionally include different *local* aggregation schemes. In this new extended formalism, entry $x_i^k$ represents node $i$ in subgraph $k$; accordingly, each aggregation in Equation 4 can be also performed locally, i.e. extending only over $i$'s neighbours, as prescribed by the connectivity of subgraph $k$ or of the original input graph. As an example, when updating entry $x_i^{k,(t)}$, term $\square_{j \neq k}\, x_j^{k,(t)}$ is expanded as $\big(\square_{j \neq k}\, x_j^{k,(t)}, \square_{j \sim_k i}\, x_j^{k,(t)}, \square_{j \sim i}\, x_j^{k,(t)}\big)$, with $\sim_k$ denoting adjacency in subgraph $k$, and $\sim$ that in the original graph connectivity. Each term in the expansion is associated with a specific learnable linear transformation. We report a full list of pooling operations in Appendix D, Table 3. These local pooling operations allow to readily recover sparse message passing, which constitutes the main computational primitive of all popular (Subgraph) GNNs. Other characteristic Subgraph GNN operations are also recovered by this formalism: for example, $\square_{h \neq i}\, x_i^{h,(t)}$ operates global pooling of node $i$'s representations across subgraphs, as previously introduced in Bevilacqua et al. [7], Zhao et al. [61]. We also note that additional, novel, operations are supported, e.g. the transpose $x_k^{i,(t)}$. We generally refer to this framework as `ReIGN(2)` ("Rethought 2-IGN").

`ReIGN(2)` **architectures.** `ReIGN(2)` induces (linear) layers in the same form of Equation 4, but where $\square$ terms are expanded to both local and global operations, as explained. These layers can operate on any bag generated by a node-based selection policy $\bar{\pi}$, and can be combined together in `ReIGN(2)` stacks of the form $\mathcal{S}_\mathcal{R} = L^{(T)} \circ \sigma \circ L^{(T-1)} \circ \sigma \circ \ldots \circ \sigma \circ L^{(1)}$, where $\sigma$'s are pointwise nonlinearities and $L$'s are `ReIGN(2)` layers. This allows us to define `ReIGN(2)` models as Subgraph GNNs in the form of Equation 1, where $\mathcal{S}$ is a `ReIGN(2)` layer stacking and $\pi$ is node-based: $\mathcal{R}_{\bar{\pi}} = \mu \circ \rho \circ \mathcal{S}_\mathcal{R} \circ \bar{\pi}$.

More generally, `ReIGN(2)` induces a 'layer space' for node-based Subgraph GNNs: the expanded terms in its update equations represent a pool of atomic operations that can be selected and combined to define new equivariant layers. Compared to that of `3-IGNs`, this space is of tractable size, yet it recovers previously proposed Subgraph GNNs and allows to define novel interesting variants.

**Recovering previous Subgraph GNNs.** The following result states that the `ReIGN(2)` generalises all known subgraph methods in $\Upsilon$, as their layers are captured by a `ReIGN(2)` stacking.

**Theorem 8** (`ReIGN(2)` implements node-based Subgraph GNNs). *Let $\mathcal{N}$ be a model in family $\Upsilon$ equipped with Morris et al. [36] message-passing base-encoders. For any instance $\mathcal{N}_\Theta$, there exists `ReIGN(2)` instance $\mathcal{R}_\Omega$ such that $\mathcal{R}_\Omega \cong \mathcal{N}_\Theta$.*

This shows that known methods are generalised without resorting to the $\mathcal{O}(n^3)$ computational complexity of `3-IGN`s. Figure 3 illustrates the aggregation and sharing rules used by previous Subgraph GNNs to update root and non-root nodes, and compare them with those of `ReIGN(2)` and `2-IGN`s. We visualise these on the subgraph-node sub-tensor gathering node representations across subgraphs; here, root nodes occupy the main diagonal, non-root nodes all the remaining off-diagonal entries. As for to the `2-IGN` Equations 4, the elements in these two partitions may be updated differently, so we depict them separately in, respectively, the bottom and top rows. In each depiction we colour elements depending on the set of weights parameterising their contribution in the update process, with two main specifications: (i) Elements sharing the same colour are pooled together; (ii) Triangles indicate such pooling is performed locally based on the subgraph connectivity at hand (two triangles indicate both local and global pooling ops are performed). E.g., note how DS-GNN equivalently updates the representations of root and non-root nodes via the same (local) message-passing layer (triangles, yellow, leftmost picture). By illustrating how `ReIGN(2)` generalises previous node-based methods, this figure is to be interpreted as visual support for the Proof of Theorem 8 (see Appendix D). Additional details and discussions on Figure 3 are found in Appendix C.

Notably, as methods in $\Upsilon$ have been shown to be strictly stronger than `2-WL` [7, 14, 61, 59, 56], Theorem 8 implies the same lower bound for `ReIGN(2)`. Nevertheless, when employing policies in $\Pi$ and `3-IGN`-computable invariant pooling functions $\rho$ (as those used by models in $\Upsilon$), `ReIGN(2)`s are upper-bounded by `3-IGN`s:

**Proposition 9** (`3-IGN`s implement `ReIGN(2)`). *For any pair of non-isomorphic graphs $G_1, G_2$ in family $\mathcal{G}$, if there exist policy $\bar{\pi} \in \Pi$, parameters $\Theta$ and `3-IGN`-computable invariant pooling function $\rho$ such that the `ReIGN(2)` instance $\mathcal{R}_{\rho,\Theta,\bar{\pi}}$ distinguishes $G_1$, $G_2$, then there exist weights $\Omega$ for a `3-IGN` instance $\mathcal{M}_\Omega$ such that $G_1, G_2$ are distinguished by $\mathcal{M}_\Omega$ as well.*

This proposition entails an upper-bound on the expressive power of `ReIGN(2)`.

**Corollary 10** (`3-WL` upper-bounds `ReIGN(2)`). *The expressive power of a `ReIGN(2)` model with policy $\pi \in \Pi$ and `3-IGN`-computable invariant pooling function $\rho$ is upper-bounded by `3-WL`.*

We note that there may be layers equivariant to $S_n$ over $\mathbb{R}^{n^2}$ not captured by `ReIGN(2)`. Yet, previously proposed Subgraph GNN layers do not exhaust the `ReIGN(2)` design space, which remains largely unexplored. One, amongst possible novel constructions, is introduced next.

### 6.2 A unifying architecture: Subgraph Union Networks

We now show how the `ReIGN(2)` layer space can guide the design of novel, expressive, Subgraph GNNs. Our present endeavour is to conceive a computationally tractable architecture subsuming known node-based models: in virtue of this latter desideratum, we will dub this architecture "Subgraph Union Network" (SUN). To design the base equivariant layer for SUN, we select and combine specific aggregation terms suggested by the `ReIGN(2)` framework:

$$x_i^{i,(t+1)} = \sigma\Big(\upsilon_{\theta_1}\big(x_i^{i,(t)}, \sum_{j \sim_i i} x_j^{i,(t)}, \sum_j x_j^{i,(t)}, \sum_h x_i^{h,(t)}, \sum_{j \sim i} \sum_h x_j^{h,(t)}\big)\Big) \tag{5}$$

$$x_i^{k,(t+1)} = \sigma\Big(\upsilon_{\theta_2}\big(x_i^{k,(t)}, \sum_{j \sim_k i} x_j^{k,(t)}, x_i^{i,(t)}, x_k^{k,(t)}, \sum_j x_j^{k,(t)}, \sum_h x_i^{h,(t)}, \sum_{j \sim i} \sum_h x_j^{h,(t)}\big)\Big) \tag{6}$$

where $\upsilon$'s sum their inputs after applying a specific linear transformations to each term. One of the novel features of SUN is that roots are transformed by a *different set of parameters* ($\theta_1$) than the other nodes [7] ($\theta_2$, see Figure 2). In practice, the first and last two terms in each one of Equations (5) and (6) can be processed by maximally expressive MPNNs [36, 55], the remaining terms by MLPs. We test these variants in our experiments, with their formulations in Appendix G. SUN remains an instantiation of the `ReIGN(2)` framework:

**Proposition 11** (A `ReIGN(2)` stacking implements SUN layers). *For any SUN layer $L$ defined according to Equations 5 and 6, there exists a `ReIGN(2)` layer stacking $\mathcal{S}_L$, such that $\mathcal{S}_L \cong L$.*

---

[7]As a result, the architecture can mark root nodes, for example.

Table 1: Test mean MAE on the Counting Substructures and ZINC-12k datasets. All Subgraph GNNs employ a GIN base-encoder. [†]This version of GNN-AK+ does not follow the standard evaluation procedure.

| Method | Counting Substructures (MAE $\downarrow$) | | | |
|---|---|---|---|---|
| | Triangle | Tailed Tri. | Star | 4-Cycle |
| GCN [27] | 0.4186 | 0.3248 | 0.1798 | 0.2822 |
| GIN [55] | 0.3569 | 0.2373 | 0.0224 | 0.2185 |
| PNA [13] | 0.3532 | 0.2648 | 0.1278 | 0.2430 |
| PPGN [32] | 0.0089 | 0.0096 | 0.0148 | **0.0090** |
| GNN-AK [61] | 0.0934 | 0.0751 | 0.0168 | 0.0726 |
| GNN-AK-CTX [61] | 0.0885 | 0.0696 | 0.0162 | 0.0668 |
| GNN-AK+ [61] | 0.0123 | 0.0112 | 0.0150 | 0.0126 |
| **SUN (EGO)** | 0.0092 | 0.0105 | 0.0064 | 0.0140 |
| **SUN (EGO+)** | **0.0079** | **0.0080** | **0.0064** | 0.0105 |

| Method | ZINC (MAE $\downarrow$) |
|---|---|
| GCN [27] | $0.321 \pm 0.009$ |
| GIN [55] | $0.163 \pm 0.004$ |
| PNA [13] | $0.133 \pm 0.011$ |
| GSN [11] | $0.101 \pm 0.010$ |
| CIN [9] | $\mathbf{0.079} \pm 0.006$ |
| NGNN [59] | $0.111 \pm 0.003$ |
| DS-GNN (EGO) [7] | $0.115 \pm 0.004$ |
| DS-GNN (EGO+) [7] | $0.105 \pm 0.003$ |
| DSS-GNN (EGO) [7] | $0.099 \pm 0.003$ |
| DSS-GNN (EGO+) [7] | $0.097 \pm 0.006$ |
| GNN-AK [61] | $0.105 \pm 0.010$ |
| GNN-AK-CTX [61] | $0.093 \pm 0.002$ |
| GNN-AK+ [61][†] | $0.086 \pm ???$ |
| GNN-AK+ [61] | $0.091 \pm 0.011$ |
| **SUN (EGO)** | $0.083 \pm 0.003$ |
| **SUN (EGO+)** | $0.084 \pm 0.002$ |

Finally, we show that a stacking of SUN layers can implement any layer of known node-based Subgraph Networks, making this model a principled generalisation thereof.

**Proposition 12** (A SUN stacking implements known Subgraph GNN layers). *Let $\mathcal{N}$ be a model in family $\Upsilon$ employing Morris et al. [36] as a message-passing base-encoder. Then, for any layer $L$ in $\mathcal{N}$, there exists a stacking of SUN layers $\mathcal{S}_L$ such that $\mathcal{S}_L \cong L$.*

**Beyond SUN.** As it can be seen in Figure 3, SUN does not use all possible operations in the ReIGN(2) framework. Notably, two interesting operations that are not a part of SUN are: (i) The 'transpose': $x_i^k = \upsilon_\theta(x_k^i)$, which shares information between the $i$-th node in the $k$-th subgraph and the $k$-th node in the $i$-th subgraph; (ii) Local vertical pooling $x_i^k = \upsilon_\theta(\sum_{h \sim i} x_i^h)$. The exploration of these and other operations is left to future work.

## 7 Experiments

We experimentally validate the effectiveness of one ReIGN(2) instantiation, comparing SUN to previously proposed Subgraph GNNs[8]. We seek to verify whether its theoretical representational power practically enables superior accuracy in expressiveness tasks and real-world benchmarks. Concurrently, we pay attention to the *generalisation ability* of models in comparison. SUN layers are less constrained in their weight sharing pattern, resulting in a more complex model. As this is traditionally associated with inferior generalisation abilities in low data regimes, we deem it important to additionally assess this aspect. Our code is also available.[9]

**Synthetic.** Counting substructures and regressing graph topological features are notoriously hard tasks for GNNs [12, 17, 13]. We test the representational ability of SUN on common benchmarks of this kind [12, 13]. Table 1 reports results on the substructure counting suite, on which SUN attains state-of-the-art results in 3 out of 4 tasks. Additional results on the regression of global, structural properties are reported in Appendix G.

**Real-world.** On the molecular ZINC-12k benchmark (constrained solubility regression) [50, 22, 16], SUN exhibits best performance amongst all domain-agnostic GNNs under the 500k parameter budget, including other Subgraph GNNs (see Table 1). A similar trend is observed on the large-scale Molhiv dataset from the OGB [23] (inhibition of HIV replication). Results are in Table 2. Remarkably, on both datasets, SUN either outperforms or approaches HIMP [19], GSN [11] and CIN [9], GNNs which explicitly model rings. We experiment on smaller-scale TUDatasets [37] in Appendix G, where we also compare selection policies.

---

[8]For GNN-AK variants [61], we run the code provided by the authors, for which the 'context' and 'subgraph' embeddings sum only over ego-network nodes.

[9]https://github.com/beabevi/SUN

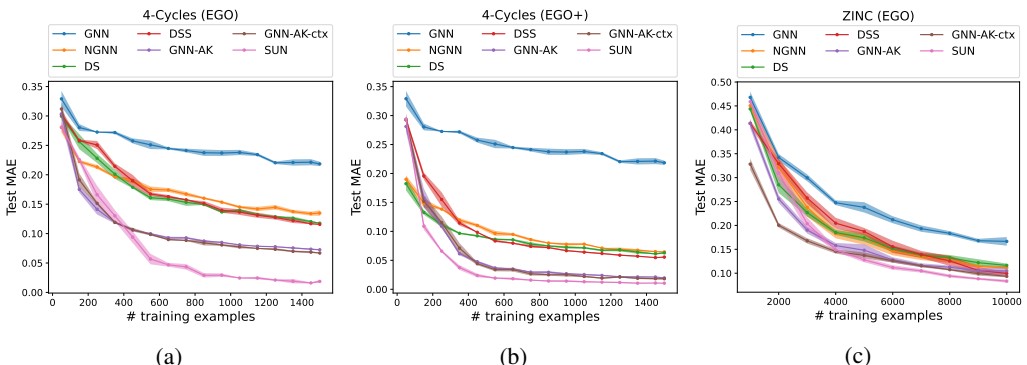

Figure 4: Generalisation capabilities of Subgraph GNNs in the counting prediction task (Figures 4a and 4b) and in the ZINC-12k dataset (Figure 4c).

**Generalisation from limited data.** In this set of experiments we compare the test performance of Subgraph GNNs when trained on increasing fractions of the available training data. Each architecture is selected by tuning the hyperparameters with the entire training and validation sets. We run this experiment on the 4-cycle counting task and the real-world ZINC-12k. We illustrate results in Figures 4a to 4c. Except for a short initial phase in the EGO policy, `SUN` generalises better than other Subgraph GNNs on cycle-counting. On ZINC-12k, `SUN` outruns DSS-, DS-GNN and GNN-AK variants from, respectively, 20, 30 and 40% of the samples. These results demonstrate that `SUN`'s expressiveness is not at the expense of sample efficiency, suggesting that its modelled symmetries guarantee strong representational power while retaining important inductive biases for learning on graphs.

Table 2: Test results for OGB dataset. GIN base-encoder for each Subgraph GNN.

| Method | OGBG-MOLHIV ROC-AUC (%) |
|---|---|
| GCN [27] | 76.06±0.97 |
| GIN [55] | 75.58±1.40 |
| PNA [13] | 79.05±1.32 |
| DGN [6] | 79.70±0.97 |
| HIMP [19] | 78.80±0.82 |
| GSN [11] | 80.39±0.90 |
| CIN [9] | **80.94**±0.57 |
| RECONSTR.GNN [14] | 76.32±1.40 |
| DS-GNN (EGO+) [7] | 77.40±2.19 |
| DSS-GNN (EGO+) [7] | 76.78±1.66 |
| GNN-AK+ [61] | 79.61±1.19 |
| **SUN (EGO+)** | 80.03±0.55 |

## 8 Conclusions

Our work unifies, extends, and analyses the emerging class of Subgraph GNNs. Notably, we demonstrated that the expressive power of these methods is bounded by `3-WL`. Towards a systematic study of models whose expressivity lies between `1-` and `3-WL`, we proposed a new family of layers for the class of Subgraph GNNs and, unlike most previous works on the expressive power of GNNs, we also investigated the generalisation abilities of these models, for which `SUN` shows considerable improvement. Appendix E lists several directions for future work, including an extension of our work to higher-order node-based policies.

**Societal impact.** We do not envision any negative, immediate societal impact originating from our theoretical results, which represent most of our contribution. Experimentally, our model has shown promising results on molecular property prediction tasks and strong generalisation ability in low-data regimes. This leads us to believe our work may contribute to positively impactful pharmaceutical research, such as drug discovery [20, 3].

## Acknowledgments and Disclosure of Funding

The authors are grateful to Joshua Southern, Davide Eynard, Maria Gorinova, Guadalupe Gonzalez, Katarzyna Janocha for valuable feedback on early versions of the manuscript. They would like to thank Bruno Ribeiro and Or Litany for helpful discussions, Giorgos Bouritsas for constructive conversations about the generalisation experiments and, in particular, Marco Ciccone for the precious exchange on sharpness-aware optimisation and Neapolitan pizza. MB is supported in part by ERC Consolidator grant no 724228 (LEMAN). No competing interests are declared.

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
