# Supplementary Materials for
# Understanding and Extending Subgraph GNNs by Rethinking Their Symmetries

## A  Subgraph GNNs

### A.1  Review of Subgraph GNN architectures

Here we review a series of previously proposed Subgraph GNNs, showing how the proposed architectures are captured by the formulation of Equation 1. We report this here for convenience:

$$(A, X) \mapsto \big(\mu \circ \rho \circ \mathcal{S} \circ \pi\big)(A, X).$$

$(n-k)$-**Reconstruction GNN** by Cotta et al. [14] is the simplest Subgraph GNN: it applies a Siamese MPNN base-encoder $\gamma$ to $k$-node-deleted subgraphs of the original graph and then processes the resulting representations with a set network. When $k = 1$, these models are *node-based* Subgraph GNNs. More formally, $\pi = \pi_{\mathrm{ND}}$, a DeepSets network [58] implements $\mu \circ \rho$, and $\mathcal{S}$ is realised with layers of the form:

$$X^{i,(t+1)} = \gamma_t(A^i, X^{i,(t)}) \tag{7}$$

**Equivariant Subgraph Aggregation Network (ESAN)** by Bevilacqua et al. [7] extends Reconstruction GNNs in two main ways: First, introducing subgraph selection policies that allow for more general sets of subgraphs such as edge-deleted policies. Second, performing an in-depth equivariance analysis which advocates the use of the DSS layer structure introduced by Maron et al. [35]. This choice gives rise to a more expressive architecture that shares information between subgraphs. Formally, in ESAN, $\mathcal{S}$ is defined as a sequence of equivariant layers which process subgraphs as well as the aggregated graph $G_{\mathrm{agg}} = \big(A^{\mathrm{agg}}, X^{\mathrm{agg}}\big) = \sum_{G^i \in B_G} G^i$. Each layer in $\mathcal{S}$ is of the following form:

$$X^{i,(t+1)} = \gamma_t^0(A^i, X^{i,(t)}) + \gamma_t^1(A^{\mathrm{agg}}, X^{\mathrm{agg},(t)}) \tag{8}$$

with $\gamma_t^0, \gamma_t^1$ being two *distinct* MPNN base-encoders. The above architecture is referred to as DSS-GNN. Bevilacqua et al. [7] also explore disabling component $\gamma^1$ and term this simplified model DS-GNN (which reduces to a Reconstruction GNN of Cotta et al. [14] under node-deletion policies[10]). In the same work, the considered policies are edge-covering [7, Definition 7], that is, each edge in the original connectivity appears in the connectivity of at least one subgraph. In view of this observation, the authors consider and implement a simplified version of DSS-GNN, whereby $\gamma^1$'s operate on the original connectivity $A$, rather than $A^{\mathrm{agg}}$, that is:

$$X^{i,(t+1)} = \gamma_t^0(A^i, X^{i,(t)}) + \gamma_t^1(A, X^{\mathrm{agg},(t)}). \tag{9}$$

Both DS- and DSS-GNN are *node-based* Subgraph GNNs when equipped with policies in $\Pi$. Also, these policies are clearly edge-covering and, in this work, we will consider DSS-GNN as defined by Equation 9.

**GNN as Kernel (GNN-AK).** Zhao et al. [61] employs an ego-network policy ($\pi = \pi_{\mathrm{EGO}}$), while each layer in $\mathcal{S}$ is structured as $A \circ S$, where $S$ is a stacking of layers in the form of Equation 7 and $A$ is an aggregation/pooling block in the form:

$$x_j^{i,(t+1)} = \phi\big(h_j^{j,(t)}, \sum_\ell h_\ell^{j,(t)}\big) \tag{10}$$

with $\phi$ either concatenation or summation, $h_j^{i,(t)} = \big(\gamma_t(A^i, X^{i,(t)})\big)_j^\top$, for MPNN $\gamma_t$. The authors introduce an additional variant (GNN-AK-ctx in the following) which also pools information from nodes in other subgraphs:

$$x_j^{i,(t+1)} = \phi\big(h_j^{j,(t)}, \sum_\ell h_\ell^{j,(t)}, \sum_\ell h_j^{\ell,(t)}\big). \tag{11}$$

In this paper we consider a more general case of global summation in Equations (10)-(11) [11].

---

[10]For this reason, we will only consider DS-GNN in the following proofs.

[11]The original paper considers summation over each ego network which is specific to a particular policy. Such summations can be dealt with by adding masking node features.

**Nested GNN (NGNN).** Zhang and Li [59] also uses $\pi = \pi_{\text{EGO}}$ and applies an independent MPNN to each ego-network, effectively structuring $\mathcal{S}$ as a stack of layers in the form of Equation 7. This architecture differs in the way block $\rho$ is realised, namely by pooling the obtained local representations and running an additional MPNN $\gamma_\rho$ on the original graph:

$$x_j^{(\rho)} = \sum_\ell x_\ell^{j,(T)}, \qquad x_G = \sum_j \left(\gamma_\rho(A, X^{(\rho)})\right)_j \tag{12}$$

**ID-GNN.** You et al. [56] proposes distinguishing messages propagated by ego-network roots. This architecture uses $\pi = \pi_{\text{EGO}+(T)}$ policy and $\mathcal{S}$ as a $T$-layer stacking performing independent *heterogeneous* message-passing on each subgraph:

$$x_j^{i,(t+1)} = \upsilon_t\big(x_j^{i,(t)}, \sum_{\ell \sim_i j, \ell \neq i} \mu_{0,t}(x_\ell^{i,(t)}) + \mathbb{1}_{[i \sim_i j]} \cdot \mu_{1,t}(x_i^{i,(t)})\big) \tag{13}$$

where $\mathbb{1}_{[i \sim_i j]}$ if $i \sim_i j$, 0 otherwise, and $\sim_i$ denotes the connectivity induced by $A^i$. GNN-AK, GNN-AK-ctx, NGNN and ID-GNN are all, intrinsically, *node-based* Subgraph GNNs.

Interestingly, we notice that the contemporary work by Papp and Wattenhofer [42] suggested using a node marking policy as a more powerful alternative to node deletion. Finally, we note that the model by Vignac et al. [53] may potentially be considered as a Subgraph GNN as well.

## A.2 The computational complexity of Subgraph GNNs

Other than proposing Subgraph GNN architectures, the works by Bevilacqua et al. [7], Zhang and Li [59], Zhao et al. [61] also study their computational complexity. In particular, Bevilacqua et al. [7] describe both the space and time complexity of a subgraph method equipped with generic subgraph selection policy and an MPNN as a base encoder. Given the inherent locality of traditional message-passing, the authors derive asymptotic bounds accounting for the sparsity of input graphs. Let $n, d$ refer to, respectively, the number of nodes and *maximum node degree* of an input graph generating a subgraph bag of size $b$. The forward-pass asymptotic time complexity amounts to $\mathcal{O}(b \cdot n \cdot d)$, while the memory complexity to $\mathcal{O}(b \cdot (n + n \cdot d))$. For a node-based selection policy, $b = n$ so these become, respectively, $\mathcal{O}(n^2 \cdot d)$ and $\mathcal{O}(n \cdot (n + nd))$. The authors stress the explicit dependence on $d$, which is, on the contrary, lacking in 3-IGNs. As we show in Appendix B, 3-IGNs subsume node-based subgraph methods, but at the cost of a cubic time and space complexity ($\mathcal{O}(n^3)$) [7]. Amongst others, this is one reason why Subgraph GNNs may be preferable when applied to sparse, real world graphs (where we typically have $d \ll n$).

As we have noted in the above, more sophisticated Subgraph GNNs layers may feature "global" pooling terms other than local message-passing: see, e.g., term $X^{\text{agg}}$ for DSS-GNN [7] in Equations (8) and (9) or the "subgraph" and "context" encodings in the GNN-AK-ctx model [61] (second and third term in the summation of Equation (11)). In principle, each of these operations require a squared asymptotic computational complexity ($\mathcal{O}(n^2)$). However, we note that these terms are shared in the update equations of nodes / subgraphs: in practice, it is only sufficient to perform the computation once. In this case, the asymptotic time complexity would amount to $\mathcal{O}(n^2 \cdot d + n^2)$, i.e., still $\mathcal{O}(n^2 \cdot d)$. Therefore, these Subgraph GNNs retain the same asymptotic complexity described above.

Our proposed SUN layers involve the same "local" message-passing and "global" pooling operations: the above analysis is directly applicable, yielding the same asymptotic bounds.

We conclude this section by noting that, for some specific selection policies, these bounds can be tightened. In particular, let us consider ego-networks and refer to $c$ as the maximum ego-network size. As observed by Zhang and Li [59], the time complexity of a Subgraph GNN equipped with such policy becomes $\mathcal{O}(n \cdot c \cdot d)$. Importantly, When ego-networks are of limited depth, the size of the subgraphs may be significantly smaller than that of the input graph; in other words $c \ll n$, reducing the overall forward-pass complexity.

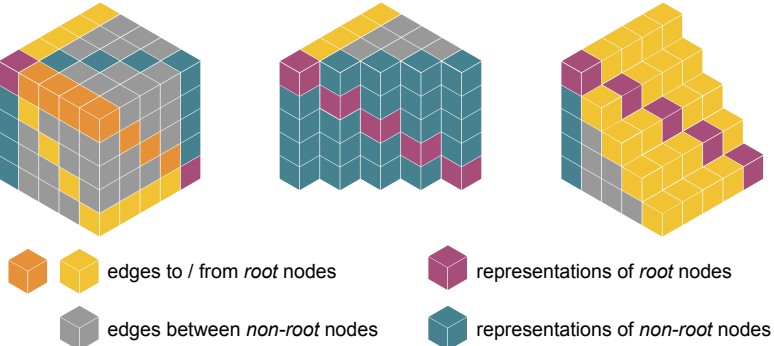

Figure 5: Depiction of cubed tensor $\mathcal{Y}$, its orbit-induced partitioning and the related semantics when $\mathcal{Y}$ is interpreted as a bag of node-based subgraphs, $n = 5$. Elements in the same partition are depicted with the same colour. Left: the whole tensor. Middle and right: sections displaying orbit representations $X_{iii}, X_{ijj}, X_{iij}$ in their entirety.

# B   Proofs for Section 5 – Subgraph GNNs and 3-IGNs

## B.1   3-IGNs as computational models

Before proving the results in Section 5, we first describe here a list of simple operations that are computable by `3-IGNs`. These opearations are to be intended as 'computational primitives' that can then be invoked and reused together in a way to program these models to implement more complex functions. We believe this effort not only serves our need to define those atomic operations required to simulate Subgraph GNNs, but also, it points out to an interpretation of `IGNs` as (abstract) comprehensive computational models beyond deep learning on hypergraphs. We start by describing the objects on which `3-IGNs` operate, and how they can be interpreted as bags of subgraphs.

### B.1.1   The 3-IGN computational data structure

The main object on which a `3-IGN` operates is a 'cubed' tensor in $\mathbb{R}^{n^3 \times d}$, typically referred to as $\mathcal{Y}$ in the following. An $S_n$ permutation group act on the first three dimensions of this tensor as:

$$\big(\sigma \cdot \mathcal{Y}\big)_{ijkl} = \mathcal{Y}_{\sigma^{-1}(i)\sigma^{-1}(j)\sigma^{-1}(k)l} \quad \forall \sigma \in S_n$$

whereas the last dimension ($l$ above) hosts $d$ 'channels' not subject to the permutation group.

The action of the permutation group on $[n]^k$ decomposes it into orbits, that is equivalence classes associated with the relation $\sim_S$ defined as:

$$\forall x, y \in [n]^k, x \sim_S y \iff \exists \sigma \in S_n : \sigma(x) = y$$

Orbits induce a partitioning of $[n]^k$. In particular, for $k = 3$ we have:

$$[n]^3 = \mathbf{o}_{iii} \sqcup \mathbf{o}_{iij} \sqcup \mathbf{o}_{iji} \sqcup \mathbf{o}_{ijj} \sqcup \mathbf{o}_{ijk} \tag{14}$$
$$\mathbf{o}_{iii} = \big\{(i,i,i) \,|\, i \in [n]\big\},$$
$$\mathbf{o}_{iij} = \big\{(i,i,j) \,|\, i,j \in [n], i \neq j\big\},$$
$$\mathbf{o}_{iji} = \big\{(i,j,i) \,|\, i,j \in [n], i \neq j\big\},$$
$$\mathbf{o}_{ijj} = \big\{(i,j,j) \,|\, i,j \in [n], i \neq j\big\},$$
$$\mathbf{o}_{ijk} = \big\{(i,j,k) \,|\, i,j \in [n], i \neq j \neq k\big\}$$

As studied in Albooyeh et al. [2][12], each of these orbits indexes a specific face-vector, that is a (vectorised) sub-tensor of $\mathcal{Y}$ with a certain number of free index variables, which determines its 'size'.

---

[12]The authors study the more general case of incidence tensors of any order, for which our three-way cubed tensor is a special case.

Importantly, this entails that the partitioning in Equation 14 induces the same partitioning on $\mathcal{Y}$, so that we can interpret the cubed tensor $\mathcal{Y}$ as a disjoint union of following face-vectors[13]:

$$\mathcal{Y} \cong X_{iii} \sqcup X_{iij} \sqcup X_{iji} \sqcup X_{ijj} \sqcup X_{ijk} \qquad (15)$$

$$X_{iii} = \left(\mathcal{Y}\right)_{\mathbf{o}_{iii}} \qquad \text{(size 1)},$$

$$X_{iij} = \left(\mathcal{Y}\right)_{\mathbf{o}_{iij}} \qquad \text{(size 2)},$$

$$X_{iji} = \left(\mathcal{Y}\right)_{\mathbf{o}_{iji}} \qquad \text{(size 2)},$$

$$X_{ijj} = \left(\mathcal{Y}\right)_{\mathbf{o}_{ijj}} \qquad \text{(size 2)},$$

$$X_{ijk} = \left(\mathcal{Y}\right)_{\mathbf{o}_{ijk}} \qquad \text{(size 3)}$$

or, more compactly, as $\mathcal{Y} \cong \bigsqcup_\omega X_\omega, \omega \in \Omega = \{iii, iij, iji, ijj, ijk\}$. According to this notation, we consider subscripts in $\Omega$ for $X$ as indexing variables for $\mathcal{Y}$. Importantly, since they directly reflect the set of indexes in $\mathbf{o}_{\mathbf{iii}}, \mathbf{o}_{\mathbf{iij}}, \mathbf{o}_{\mathbf{iji}}, \mathbf{o}_{\mathbf{ijj}}, \mathbf{o}_{\mathbf{ijk}}$, when subscripting $X$, $i, j, k$ are always distinct amongst each other. At the same time, as it can be observed from the definition above, we keep duplicate indexing variables in the subscripts of $X$'s to highlight the characteristic equality pattern of the corresponding orbit. As an example, element $\left(\mathcal{Y}\right)_{1,1,1} \in \mathbb{R}^d$ uniquely belongs to face-vector $X_{iii}$, elements $\left(\mathcal{Y}\right)_{1,1,2}, \left(\mathcal{Y}\right)_{1,2,1}, \left(\mathcal{Y}\right)_{1,2,2}$ to, respectively, face-vectors $X_{iij}, X_{iji}, X_{ijj}, \left(\mathcal{Y}\right)_{1,2,3}$ to $X_{ijk}$. Since each of these face-vectors is uniquely associated with a particular orbit, we will more intuitively refer to them as 'orbit representations' in the following.

From the considerations above, it is interesting to notice that orbit representations have a precise, characteristic collocation within the cubed tensor $\mathcal{Y}$, directly induced by the equality patterns of the orbits they are indexed by. In particular, $X_{iii}$ corresponds to elements on the main diagonal of the cube, $X_{iij}, X_{iji}, X_{ijj}$ to its three diagonal planes (with main-diagonal excluded), while $X_{ijk}$ corresponds to the the overall adjacency cube (with the main diagonal and the three diagonal planes excluded). The described partitioning is visually depicted in Albooyeh et al. [2, Figure in page 6] and Figure 5 (corresponding to Figure 2 in the main paper).

### B.1.2 Bag-of-subgraphs interpretation

An important observation underpinning the majority of our results is that the described cubed tensor $\mathcal{Y}$ can represent bags of node-based subgraphs — this is in contrast with standard interpretations whereby this tensor represents a 3-hypergraph [33, 32]. In particular, as depicted in Figure 1, we arrange subgraphs on the first axis, whereas the second and third axes index nodes.

Accordingly, the adjacency matrix and node representations for subgraph $\bar{i}$ are in sub-tensor $\mathcal{Z}_{\bar{i}} = \left(\mathcal{Y}\right)_{\bar{i},j,k,l} \in \mathbb{R}^{n^2 \times d}$, with $j, k = 1, \ldots, n, l = 1, \ldots, d$. Here, coherently with [33], we assume node representations are stored in the on-diagonal entries ($j = k$) of $\mathcal{Z}_{\bar{i}}$, while off-diagonal terms ($j \neq k$) host edges, i.e. connectivity information.

We note that this interpretation of $\mathcal{Y}$ assigns meaningful semantics to orbit representations (see Figure 5 for a visual illustration):

- $X_{iii}$ stores representations for root nodes;
- $X_{ijj}$ stores representations for non-root nodes;
- $X_{iij}$ stores connections incoming into root nodes;
- $X_{iji}$ stores connections outgoing from root nodes;
- $X_{ijk}$ stores connections between non-root nodes.

To come back to the examples above, and consistently with the aforementioned semantics, $\left(\mathcal{Y}\right)_{1,1,1}$ represents subgraph 1's root node (that is node 1); $\left(\mathcal{Y}\right)_{1,1,2}, \left(\mathcal{Y}\right)_{1,2,1}$ the connectivity between such root node and node 2 in the same subgraph; $\left(\mathcal{Y}\right)_{1,2,2}$ represents node 2 in subgraph 1; $\left(\mathcal{Y}\right)_{1,2,3}$ the connectivity from node 3 to node 2 in subgraph 1.

---

[13]The use of symbol '$\cong$', referring to an isomorphism, follows Albooyeh et al. [2].

It is important to note how this interpretation induces a correspondence between the 3-IGN tensor $\mathcal{Y}$ and tensors $\mathcal{A}, \mathcal{X}$ introduced in the main paper, Section 4. $\mathcal{X}$ gathers node features across subgraphs and is therefore in correspondence with $X_{iii} \sqcup X_{ijj}$; $\mathcal{A}$ hosts subgraph connectivity information and is thus in correspondence with $X_{iij} \sqcup X_{iji} \sqcup X_{ijk}$.

As a last note, this interpretation already preludes the more general, less constrained weight sharing pattern advocated by the ReIGN(2) framework, which prescribes, for example, parameters specific to root- and non-root-updates. See Figure 3. This will become more clear in the following (see Equation 18).

### B.1.3 Updating orbit representations

A 3-IGN architecture has the following form:

$$(A, X) \mapsto \big(m \circ h \circ \mathcal{I}\big)(A, X), \qquad \mathcal{I} = L^{(T)} \circ \sigma \ldots \circ \sigma \circ L^{(1)} \tag{16}$$

where $m$ is an MLP, $h$ is an invariant linear 'pooling' layer and $L$'s are equivariant linear k-IGN-layers, with k $\leq 3$. From here onwards we will assume $\sigma$'s are ReLU non-linearities.

Albooyeh et al. [2] show that, effectively, a linear 3-IGN-layer $L$ updates each orbit representation such that output $X_{\omega'}$ is obtained as the sum of linear equivariant transformations of input orbit representations $X_\omega$[14]:

$$\mathcal{Y}^{(t+1)} = L(\mathcal{Y}^t) \cong \bigsqcup_{\omega' \in \Omega} X_{\omega'}^{(t+1)} \tag{17}$$

$$X_{\omega'}^{(t+1)} = \sum_{\omega \in \Omega} \boldsymbol{W}^{\omega \to \omega'}(X_\omega^{(t)}) \tag{18}$$

where, as the authors show, each $\boldsymbol{W}^{\omega \to \omega'}$ corresponds to a linear combination of all pooling-broadcasting operations defined between input-output orbit representation $X_\omega, X_{\omega'}$:

$$\boldsymbol{W}^{\omega \to \omega'}(X_\omega) = \sum_{\substack{\mathbb{P} \subseteq [m] \\ \mathbb{B} \subseteq \langle 1, \ldots, m' \rangle \\ |\mathbb{B}| = m - |\mathbb{P}|}} W_{\mathbb{B}, \mathbb{P}}^{\omega \to \omega'} \texttt{Broad}_{\mathbb{B}, m'}\big(\texttt{Pool}_{\mathbb{P}}(X_\omega)\big) \tag{19}$$

with $m$ the size of input orbit representation $X_\omega$ and $m'$ the size of the output one. `Broad` and `Pool` are defined in [2] as follows.

**Pooling** Let $X_{i_1, \ldots i_m}$ be a generic face-vector of size $m$ indexed by $i_1, \ldots i_m$. Let $\mathbb{P} = \{p_1, \ldots, p_\ell\} \subseteq [m]$. $\texttt{Pool}_{\mathbb{P}}$ sums over the indexes in $\mathbb{P}$:

$$\texttt{Pool}_{\mathbb{P}}\big(X_{i_1, \ldots i_m}\big) = \sum_{\substack{i_{p_1} \neq \ldots \neq i_{p_\ell} \\ i_{p_1} \in [n], \ldots, i_{p_\ell} \in [n]}} X_{i_1, \ldots i_m} \tag{20}$$

where the inequality constraints in the summation derive from the fact that $X_{i_1, \ldots i_m}$ represents an orbit of the permutation group. To shorten the notation, we will write pooling operations as:

$$\texttt{Pool}_{\mathbb{P}}\big(X_{i_1, \ldots i_m}\big) = \boldsymbol{\pi}_{out}\, X_{i_1, \ldots i_m}$$

where $out = \{i_p \,|\, p \in [m] \setminus \mathbb{P}\}$ i.e. the set of indexes which are *not* pooled over, from which it follows that the cardinality of $out$ states the size of the resulting object. For example, $\boldsymbol{\pi}_{i_1}\, X_{i_1, i_2}$ returns a size-1 object from a size-2 face-vector by summing over index variable $i_2$.

The pooling operation applies in the same way when we repeat index variables in the subscript of the input orbit representation: as more concrete examples, when interpreting $\mathcal{Y}$ as a bag of subgraphs, $\boldsymbol{\pi}_i\, X_{ijj}$ sums the representations of all non-root nodes (subgraph readout) as:

$$\Big(\texttt{Pool}_j\big(X_{ijj}\big)\Big)_1 = \big(\boldsymbol{\pi}_i\, X_{ijj}\big)_1 = \Big(\sum_{j \in [n], j \neq i} (\mathcal{Y})_{i,j,j}\Big)_1 = \sum_{j \in [n], j \neq 1} (\mathcal{Y})_{1,j,j}$$

Similarly, $\boldsymbol{\pi}_j\, X_{ijj}$ sums non-root node representations across subgraphs (cross-subgraph aggregation). Set $out$ can be empty, in which case pooling amounts to global summation: e.g., $\boldsymbol{\pi}\, X_{iii}$ sums the representations of all root nodes across the bag. Pooling boils down to the identity operation when $out$ replicates the free indexes in $X$ – as no pooling is effectively performed. We will refrain from explicitly writing this operation.

---

[14]Even if omitted, each face-vector update equation includes a bias term.

**Broadcasting** Let $X_{i_1,\ldots i_m}$ be a generic face-vector of size $m$ indexed by $i_1,\ldots i_m$ and $\mathbb{B} = (b_1,\ldots,b_m)$ a tuple of $m$ indexes, with $b_j \in [m'], j = 1,\ldots m, m' \geq m$. Operation $\texttt{Broad}_{\mathbb{B},m'}$ "broadcasts" $X$ over a target size-$m'$ face-vector in the sense that it identifies $X$ by the target index sequence $\mathbb{B}$ and broadcasts across the remaining $m' - m$ indexes:

$$\left(\texttt{Broad}_{\mathbb{B},m'}\big(X_{i_1,\ldots,i_m}\big)\right)_{i_1,\ldots,i_{m'}} = X_{i_{b_1},\ldots,i_{b_m}} \tag{21}$$

For example, if input $X_{i_1,i_2}$ is a size-2 face-vector and output $Y_{i_1,i_2,i_3}$ is a size-3 face-vector, $\texttt{Broad}_{(1,2),3}\big(X_{i_1,i_2}\big)$ maps $X$ onto the first two indexes of $Y$, and broadcasts along the third. As another example, for output size-2 face-vector $Z_{i_1,i_2}$, $\texttt{Broad}_{(2,1),2}\big(X_{i_1,i_2}\big)$ effectively implements the 'transpose' operation. Similarly as above, we shorten the notation. Let us define $\iota$ such that, $\forall \ell \in [m']$:

$$\iota(\ell) = \begin{cases} j \text{ s.t. } b_j = \ell & \text{if } \ell \in \mathbb{B}, \\ * & \text{otherwise.} \end{cases}$$

where $*$ indicates an index over which broadcasting is performed. We write $\mathbb{C} = (i_{\iota(1)},\ldots,i_{\iota(m')})$ and rewrite the broadcast operation as:

$$\texttt{Broad}_{\mathbb{B},m'}\big(X_{i_1,\ldots,i_m}\big) = \boldsymbol{\beta}_{\mathbb{C}}\, X_{i_1,\ldots,i_m}$$

Accordingly, the examples above are rewritten as $\boldsymbol{\beta}_{i_1,i_2,*}\, X_{i_1,i_2}, \boldsymbol{\beta}_{i_2,i_1}\, X_{i_1,i_2}$:

$$\big(Y_{i_1,i_2,i_3}\big)_{1,2,3} = \big(\texttt{Broad}_{(1,2),3}\big(X_{i_1,i_2}\big)\big)_{1,2,3} = \big(\boldsymbol{\beta}_{i_1,i_2,*}\, X_{i_1,i_2}\big)_{1,2,3} = \big(X_{i_1,i_2}\big)_{1,2}$$

$$\big(Z_{i_1,i_2}\big)_{1,2} = \big(\texttt{Broad}_{(2,1),2}\big(X_{i_1,i_2}\big)\big)_{1,2} = \big(\boldsymbol{\beta}_{i_2,i_1}\, X_{i_1,i_2}\big)_{1,2} = \big(X_{i_1,i_2}\big)_{2,1}$$

In the subscript of $\boldsymbol{\beta}$, we can conveniently retain the equality pattern of the orbit representation where we are broadcasting onto. For a concrete example, $X_{ijj} = \boldsymbol{\beta}_{i,*j,*j}\, X_{iii}$ broadcasts the root node representations over non-root ones, in a way that each non-root node $j$ in subgraph $i$ gets the representation of root node $i$. It has to be noted that, in these cases, the length of tuple $\mathcal{C}$ may not correspond to the size of the output face-vector, i.e. in those cases where indexes repeat, as above. Finally, let us note that, when both input and target face-vectors have size $m$ and $\mathbb{B} = [m]$, the broadcasting operation boils down to the identity, e.g. $\boldsymbol{\beta}_{ijk}\, X_{ijk}$. We will omit it in writing in these cases.

The above results suggest that a way to describe a 3–IGN stacking $\mathcal{I} = L^{(T)} \circ \sigma \ldots \circ \sigma \circ L^{(1)}$ is to specify how each orbit representation $X_{\omega'}$ is updated from time step $(t)$ to $(t+1)$, according to Equation 18, expanded as per Equation 19. In other words, stacking $\mathcal{I}$ is described by specifying, for each layer $L^{(t)}$, every linear operator $W_{\mathbb{B},\mathbb{P},t}^{\omega \rightarrow \omega'}$ in Equation 19. This is the main strategy we adopt in the proofs described in the following.

As a last note, in an effort to ease the notation, we will:

1. Describe updates for $X_{\omega'}$'s (in the form of Equation 18) only when non-trivial;

2. (For each of the above) specify only the non-null linear operators $W_{\mathbb{B},\mathbb{P},t}^{\omega \rightarrow \omega'}$ and corresponding terms.

For example, if layer $L^{(t)}$ only applies linear transformation $W_{(1),\varnothing,t}^{iii \rightarrow iii} = W$ to orbit representation $X_{iii}^{(t)}$, we will simply write:

$$X_{iii}^{(t+1)} = W \cdot X_{iii}^{(t)}$$

implying that, according to 2., all other terms in Equation 19 are nullified by operator $\mathbf{0}$ and, according to 1., all other orbit updates read as $X_{\omega}^{(t+1)} = I_d \cdot X_{\omega}^{(t)}$.

### B.1.4 Pointwise operations

We define as 'pointwise' those operations which only apply to the feature dimension(s) and implement a form of channel mixing. By operating independently on the feature dimension(s), these are trivially equivariant. As these operations can be performed on any orbit representation, we will not specify

them in the following descriptions, and will simply assume to be working with generic tensors $X, Y, Z$.

Linear transformations are the most natural pointwise operations the `3-IGN` framework supports. Some of these are of particular interest as they will be heavily used in our proofs. We describe them below and define convenient notation for them.

**Copy/Routing**   Pointwise linear operators can copy some specific feature channels in the input tensor and route them into some other output channels. Let $X$ refer to a tensor representing orbit elements with $d_{in}$ channels, and $Y$ to an output tensor with $d_{out}$ channels. We will write:

$$Y = \boldsymbol{\kappa}^{a:b}_{e:f} X$$

to refer to the operation which writes channels $a$ to $b$ (included) in $X$ into channels $e$ to $f$ (included) in $Y$, with $a \leq b \leq d_{in}$, $e \leq f \leq d_{out}$, $b - a = f - e$. Here, $\boldsymbol{\kappa}^{a:b}_{e:f}$ is used to conveniently denote operator $W$: a matrix $d_{out} \times d_{in}$ where the square submatrix $W_{e:f,a:b} = I_{b-a+1}$ and other entries are 0. When omitting left indices we start from the first channel, i.e., $\boldsymbol{\kappa}^{:b}_{e:f} = \boldsymbol{\kappa}^{1:b}_{e:f}$. When omitting right indices we end at the last channel, i.e., $\boldsymbol{\kappa}^{a:b}_{e:} = \boldsymbol{\kappa}^{a:b}_{e:f}$ for $f$ output channels. We will use notation $\boldsymbol{\kappa}^{a:b}_{c:d/(e)}$ to specify the target has $e$ channels, whenever not clear from the context.

**Selective copy/routing**   With a slight notation overload, we also write $\boldsymbol{\kappa}^{in}_{out}$ — with $in, out$ being index tuples of the same cardinality $\ell$ — to refer to specific channels in, respectively, input and output tensors. For instance, $\boldsymbol{\kappa}^{a,b,c}_{d,e,f}$ copies (or routes) input channels $a, b, c$ into output channels $d, e, f$, with such indices being not necessarily contiguous. Again, this operation is linear and is implemented by a matrix $W$ having 1 in all entries in the form $(out_k, in_k), k \in [\ell]$, 0 elsewhere.

**Concatenation**   Two (compatible) operands can be concatenated by means of the copy/routing operation above along with summation. For example, if the two are $d$ dimensional, it is sufficient to route them into the two distinct halves of a $2d$-dimensional output tensor and then sum the two:

$$Z = \boldsymbol{\kappa}^{:}_{:d} X + \boldsymbol{\kappa}^{:}_{d+1:2d} Y$$

**Concurrent linear transformations**   We note that one single linear operator can apply multiple linear transformations on (a specific subset of) channels of the input tensor. Let $W_1, W_2$ be two operators of size $d_{out} \times d_{in}$. Obtained as the vertical stacking of $W_1, W_2$, operator $W$ can be applied to tensor $X$ with $d_{in}$ channels. It produces an output tensor $Y$ with $2 \cdot d_{out}$ channels, where the first $d_{out}$ channels store the result of applying $W_1$, channels $d_{out} + 1$ to $2 \cdot d_{out}$ store that from the application of $W_2$. We write:

$$Y = [W_1 \ \ W_2] X$$

Pre-multiplying routing operators allows these transformations to be applied to a selection of a subset of input channels. As an example, let $X$ be an input with $2 \cdot d$ channels, and $W_1, W_2$ be two $d \times d$ linear operators. Expression:

$$Y = [W_1 \cdot \boldsymbol{\kappa}^{:d}_{:d} \ \ W_2 \cdot \boldsymbol{\kappa}^{d+1:2d}_{:d}] X$$

effectively applies $W_1$ to the first $d$ channels of $X$, $W_2$ to channels $d+1$ to $2d$. In fact, $[W_1 \cdot \boldsymbol{\kappa}^{:d}_{:d} \ \ W_2 \cdot \boldsymbol{\kappa}^{d+1:2d}_{:d}]$ consists, as a whole, of a block diagonal matrix made up of operators $W_1$ (leftmost upper block) and $W_2$ (rightmost lower block). Clearly, these operations can be extended to more than two concurrent linear operators.

We note that this notation also captures the following 'replication' operation:

$$Y = [I_d \ \ I_d] X$$

which outputs two (stacked) copies of $d$-dimensional input tensor $X$.

**Concurrent transformations and channel-wise summation**   Let $X$ be an tensor with $2 \cdot d$ channels, and $W_1, W_2$ two $d \times d$ linear operators. The following expression applies $W_1$ to the first $d$ channels of $X$, $W_2$ to channels $d+1$ to $2d$ *and* sums the result channel by channel:

$$Y = [W_1 \,\|\, W_2] X$$

In fact, $[W_1 \,\|\, W_2]$ can be interpreted as single linear operator $d \times 2 \cdot d$ obtained by horizontally stacking $W_1, W_2$. As a particular case, we can simply sum the two halves of $X$ by $[I_d \,\|\, I_d] X$.

**A note on non-linearities**   A `3-IGN`-layer stacking is in the form $\mathcal{I} = L^{(T)} \circ \sigma \ldots \circ \sigma \circ L^{(1)}$, where $L$'s are linear equivariant layers and $\sigma$'s are ReLU non-linearities. In principle, these non-linearities in between layers may alter the result of linear computation they perform. For example, in order to perform an exact copy of input representation $X$, it may not be sufficient to simply choose an identity weight matrix $I_d$: the following ReLU non-linearity would clip negative entries to 0, thus invalidating the correctness of the operation. However, we note that the identity function can be implemented by a ReLU-network, as $y = x = \sigma(x) - \sigma(-x)$. This means that the copy operation can be realised by a `3-IGN`-layer stacking as:

$$Y = [I_d \,\|\, -I_d] \cdot \Big( \sigma\big( [I_d \;\; -I_d] \, X \big) \Big)$$

This effectively provides us with a way to 'choose', in a `3-IGN` layer stacking, when to apply $\sigma$'s and when not to. Indeed, we can always work with $2d$ channels where all entries are non-negative: the positive entries in the first $d$ channels store the originally positive ones, those in the second $d$ channels store the originally negative ones, negated. This expansion can be realised by one layer as $Y = \sigma\big([I_d \;\; -I_d] \, X\big)$ and is such that ReLU activations act neutrally. Any linear transformation $W$ is now applied as $Y = \sigma\big(UX\big)$, with $U = [I_d \;\; -I_d] \cdot W \cdot [I_d \,\|\, -I_d]$. Whenever computation requires the application of ReLUs after linear transformation $W$, it is sufficient to perform the following: $Y = [I_d \;\; -I_d] \cdot \Big( \sigma\big(VX\big) \Big)$, with $V = W \cdot [I_d \,\|\, -I_d]$. Operator $[I_d \;\; -I_d]$ will effectively be absorbed by the linear transformation in the following layer. This doubled representation also allows to apply non-linearities to some particular channels only. This can be realised by $Y = \sigma\big(U'X\big)$, with $U'$ constructed as $U$ above, but with the difference being that entries in its $(d+e)$-th row are set to 0, if $e$ is an output channel where the ReLU non-linearity takes effect.

These considerations show that, in fact, interleaving linear (equivariant) layers with ReLU non-linearities does not hinder the possibility of performing (partially) linear computation, which can always be recovered at the cost of having additional channels and/or layers. In view of the above observations, and in an effort to ease the notation, in the proofs reported in the following sections we will thus assume to be allowed to stack linear layers with no ReLU activations in between.

**Multi-Layer Perceptron**   `3-IGNs` can naturally implement the application of a Multi-Layer Perceptron (MLP) to the feature dimension(s) of a particular orbit representation, as it stacks linear layers, interspersed with non-linearities. Each of these dense linear layers is trivially equivariant, and so is the overall stacking. We generally write:

$$Y = \varphi^{(f)} \, X$$

to indicate the application of an MLP implementing (or approximating) function $f$. Additionally, we use the notation $\varphi^{(f)in}_{out}$ to indicate that $\varphi^{(f)}$ acts on the $in$ channels of its input and writes over channels $out$ of the output. This behaviour can be obtained by multiplying the MLP inputs by operator $\kappa^{in}$ and its output by operator $\kappa_{out}$.

**MLP-copy**   We can also combine copy and MLP operations together. In other words, we can apply an MLP on some particular channels while copying some others; we write this operation as

$$Y = [\kappa^{in}_{out} \;\; \varphi^{in'}_{out'}] \, X$$

This operation can be implemented by 'embedding' the weights of the MLP in appropriately sized matrices where remaining entries perform copy/routing operations and are not affected by non-linearities (see discussion above). More in specific, to see how this is implemented, let the MLP $\varphi^{in'}_{out'}$ above have weight matrices $(W_1, W_2, \ldots, W_L)$ with $W_1 \in \mathbb{R}^{d_1 \times |in'|}$, $W_l \in \mathbb{R}^{d_l \times d_{l-1}}, \forall l = 2 \ldots L-1$, $W_L \in \mathbb{R}^{|out'| \times d_{L-1}}$. Then, $Y$ is obtained by stacking:

$$Y = \big(U_L \circ \sigma \ldots \circ \sigma \circ U_1\big) X$$

where $U_l, l \in [L]$ are linear operators obtained as follows.

$$U_1 = [I_{(1)} \;\; -I^*_{(1)}] \cdot [\kappa^{in}_{:} \;\; W_1 \cdot \kappa^{in'}_{:}]$$
$$I_{(1)} = I_{|in|+d_1}$$
$$I^*_{(1)} = [[I_{|in|} \,\|\, \mathbf{0}_{|in|\times d_1}] \;\; \mathbf{0}_{d_1 \times (|in|+d_1)}]$$

For any $l = 2 \ldots L - 1$:

$$U_l = [I_{(l)} \quad -I^*_{(l)}] \cdot V_l \cdot [I_{|in|+d_{l-1}} \,||\, -I_{|in|+d_{l-1}}]$$
$$V_l = [[I_{|in|} \,||\, \mathbf{0}_{|in|\times d_{l-1}}] \quad [\mathbf{0}_{d_l \times |in|} \,||\, W_l]\,]$$
$$I_{(l)} = I_{|in|+d_l}$$
$$I^*_{(l)} = [[I_{|in|} \,||\, \mathbf{0}_{|in|\times d_l}] \quad \mathbf{0}_{d_l \times (|in|+d_l)}]$$

Finally:

$$U_L = \boldsymbol{\kappa}_{(out||out')} \cdot [I_{(L)} \quad -I^*_{(L)}] \cdot V_L \cdot [I_{|in|+d_{L-1}} \,||\, -I_{|in|+d_{L-1}}]$$
$$V_L = [[I_{|in|} \,||\, \mathbf{0}_{|in|\times d_{L-1}}] \quad [\mathbf{0}_{|out'|\times |in|} \,||\, W_L]\,]$$
$$I_{(L)} = I_{|in|+|out'|}$$
$$I^*_{(L)} = [[I_{|in|} \,||\, \mathbf{0}_{|in|\times |out'|}] \quad \mathbf{0}_{|out'|\times(|in|+|out'|)}]$$

with $(out||out')$ the concatenation of the two output index tuples $out$ and $out'$.

**Logical AND**   For inputs in $\{0,1\}$, it is possible to *exactly* implement the logical AND function $y = f(a,b) = a \wedge b$ as $y = \sigma\big(+2\cdot a + 2\cdot b - 3\big)$, with $\sigma$ being the ReLU non-linearity. As `3-IGNs` can implement pointwise MLPs, this operation can be performed as well when operands are on two distinct channels of the same orbit representation. In particular, for a $d$-dimensional input $X$ we write:

$$Y = \boldsymbol{\varphi}_c^{(\wedge)a,b}\, X$$

to indicate the operation that applies a logical AND between channels $a$ and $b$ of input $X$ and writes the result into channel $c$ in output $Y$, with $a,b,c \in [d]$. For this operation in particular, we also imply that the rest of $X$ is written in the remaining output channels, that is, we imply we are, in fact, performing $[\boldsymbol{\kappa}_{[d]\backslash\{c\}}^{[d]\backslash\{c\}} \quad \boldsymbol{\varphi}_c^{(\wedge)a,b}\,]$ .

**Clipping**   An MLP equipped with one hidden layer and ReLU activation can exactly implement the 1-clipping function $f_\downarrow = \min(x,1)$. In practice, $f_\downarrow$ clips inputs at the value of 1. For a scalar input, this is realised as:

$$y = -\sigma\big(-x + 1\big) + 1$$

An MLP $\boldsymbol{\varphi}^{(\downarrow)}$ implementing this function on each channel of a multi-dimensional input $X$ is constructed as:

$$Y = -I_d\, \sigma\big(-I_d\, X + \mathbf{1}_d\big) + \mathbf{1}_d$$

where $\mathbf{1}_d$ is the $d$-dimensional one-vector. The existence of such an MLP entails that `3-IGNs` can exactly implement the clipping function as well, as a pointwise operation.

## B.2   Implementing node-based selection policies

We now show how `IGNs` can compute node-based selection policies, proving Lemma 4. We will make use of concepts introduced above; the partitioning into orbits and the computational primitives in particular.

*Proof of Lemma 4.* We assume to be given an $n$-node graph $G$ in input, represented by tensor $A \in \mathbb{R}^{n\times n\times d}$. $A$ is subject to $S_n$ symmetry, which partitions it into two distinct orbits: on-diagonal terms $A_{ii}$ and off-diagonal terms $A_{ij}, i \neq j$. $A_{ii}$ store $d$-dimensional node features; $A_{ij}$ the binary graph connectivity in its first channels, with others being 0-padded.

The first operation, common to the implementation of all the policies, consists of lifting the two-way tensor $A$ to the three-way tensor $\mathcal{Y} \in \mathbb{R}^{n^3\times d}$ interpreted as a bag of subgraphs and partitioned into orbit tensors as described above in Section B.1.1. This step is realised by the following broadcasting

operations[15]:

$$X_{iii}^{(0)} = \boldsymbol{\beta}_{i,i,i} \, A_{ii}$$

$$X_{jii}^{(0)} = \boldsymbol{\beta}_{*,i,i} \, A_{ii}$$

$$X_{iij}^{(0)} = \boldsymbol{\beta}_{i,i,j} \, A_{ij}$$

$$X_{iji}^{(0)} = \boldsymbol{\beta}_{i,j,i} \, A_{ij}$$

$$X_{kij}^{(0)} = \boldsymbol{\beta}_{*,i,j} \, A_{ij}$$

We now focus on each of the considered policies in particular.

[***node-deletion***] Given that $X_{iij}, X_{iji}$ contain all and only those connections involving root nodes, it is sufficient to zero them out to recover a node-deleted bag. We thus perform the following operations:

$$X_{iij} = \mathbf{0} \cdot X_{iij}^{(0)}$$

$$X_{iji} = \mathbf{0} \cdot X_{iji}^{(0)}$$

[***node-marking***] adds a special 'mark' to root nodes only. We implement it by adding one additional dimension to node features and by setting that to 1 only for root nodes via the bias term:

$$X_{iii} = \boldsymbol{\kappa}_{1:d/(d+1)}^{1:d} \, X_{iii}^{(0)} + \mathbb{1}_{d+1}$$

$$X_{ijj} = \boldsymbol{\kappa}_{1:d/(d+1)}^{1:d} \, X_{ijj}^{(0)} + \mathbf{0}_{d+1}$$

$$X_{iij} = \boldsymbol{\kappa}_{1:d/(d+1)}^{1:d} \, X_{iij}^{(0)} + \mathbf{0}_{d+1}$$

$$X_{iji} = \boldsymbol{\kappa}_{1:d/(d+1)}^{1:d} \, X_{iji}^{(0)} + \mathbf{0}_{d+1}$$

$$X_{ijk} = \boldsymbol{\kappa}_{1:d/(d+1)}^{1:d} \, X_{ijk}^{(0)} + \mathbf{0}_{d+1}$$

where $\mathbb{1}_{d+1}$ is a (one-hot) $(d+1)$-dimensional vector being 1 in dimension $d+1$, 0 elsewhere; $\mathbf{0}_{d+1}$ is the $(d+1)$-dimensional zero vector.

[***ego-networks(h)***] So far, we have mostly made use of the orbit-partitioning that is induced by the $S_n$ symmetry group, which has allowed us to implement the *node-deletion* and *node-marking* policies with one single 3-IGN layer. We now show that, in order to implement the *ego-networks* policy, multiple layers are required, as a 3-IGN effectively needs to perform Breadth-First-Search for each node in the graph to construct $h$-hop neighbourhoods. We illustrate the required steps below, which mainly articulate in (i) the construction of $h$-hop neighbourhoods around nodes; (ii) extraction of egonets from such neighbourhoods. The yet-to-describe 3-IGN will realise part (i) by storing a 'reachability' patterns in an additional channel in $X_{ijj}$: element $\left(X_{ijj}^{d+1}\right)_{i=v,j=w}$ will be set to 1 if node $w$ is reachable from node $v$ in $h$ hops, 0 otherwise. In part (ii), these patterns will be utilised, for each subgraph, to nullify the connectivity involving unreachable nodes, thus effectively extracting $h$-hop egonets. We start by describing the layers implementing part (i).

(*Immediate neighbourhood*) For immediate neighbours, the reachability pattern is already (implicitly) stored in $X_{iij}$, as it contains the direct connectivity involving root nodes. We therefore copy this information into an additional channel in $X_{ijj}$. This value will be updated iteratively as we explore higher-order neighborhoods. In the following, we conveniently set $e = d + 1$ to ease the notation.

$$X_{ijj}^{(1)} = \boldsymbol{\kappa}_{:d/(e)}^{:d} \, X_{ijj}^{(0)} + \boldsymbol{\kappa}_{e:}^{:1} \, \boldsymbol{\beta}_{i,j,j} \, X_{iij}^{(0)}$$

(*Higher-order neighbourhoods*) We repeat the following steps $(h-1)$ times, and describe the generic $l$-th step. We first broadcast the current reachability pattern into $X_{ijk}$, writing it into the second channel (the first contains the original graph connectivity). Essentially, this operation 'propagates' the subgraph-wise reachability pattern copying it row-by-row.

$$X_{ijk}^{(l,1)} = X_{ijk}^{(l-1)} + \boldsymbol{\kappa}_{2:2}^{e:} \, \boldsymbol{\beta}_{i,*,j} \, X_{ijj}^{(l-1)}$$

---

[15]This layer effectively implements the *null* policy, see Appendix G.

Having placed the pattern as described above, we perform a logical AND between the first two channels of $X_{ijk}$ and write back the result into the second channel.

$$X_{ijk}^{(l,2)} = \boldsymbol{\varphi}_{2:2/(e)}^{(\wedge)1:2}\, X_{ijk}^{(l,1)}$$

For a specific node $w$ in a given subgraph $v$, this operation spots the neighbours of $w$ that are currently marked in $v$, effectively propagating the reachability information one hop farther. At this point, pooling over rows counts the number of such neighbours: if at least one is reachable, then node $w$ becomes reachable as well, and its corresponding entry must be set to 1 in the updated reachability pattern. We therefore complete the $l$-th hop step by updating the pattern accordingly and by clipping it to 1.

$$X_{ijj}^{(l,3)} = X_{ijj}^{(l,2)} + \boldsymbol{\kappa}_{e:}^{2:2}\, \boldsymbol{\beta}_{i,j,j}\, \boldsymbol{\pi}_{i,j}\, X_{ijk}^{(l,2)}$$
$$X_{ijj}^{(l)} = [\,\boldsymbol{\kappa}_{:d/(e)}^{:d}\quad \boldsymbol{\varphi}_{e:e}^{(\downarrow)e:e}\,]\, X_{ijj}^{(l,3)}$$

(*Egonet extraction*) We complete the implementation of the policy by leveraging the computed reachability pattern to extract the required egonets. We do this by nullifying the connectivity entries in $X_{ijk}$ for those nodes still unreached, i.e. we zero out row- and column-elements for nodes whose entry in the reachability pattern is 0. To perform this operation we appropriately broadcast the pattern and use it as an argument into a logical AND:

$$X_{ijk}^{(x,1)} = X_{ijk}^{(h)} + \boldsymbol{\kappa}_{2:2}^{e:}\, \boldsymbol{\beta}_{i,j,*}\, X_{ijj}^{(h)}$$
$$X_{ijk}^{(x,2)} = \boldsymbol{\varphi}_{2:2}^{(\wedge)1:2}\, X_{ijk}^{(x,1)}$$
$$X_{ijk}^{(x,3)} = X_{ijk}^{(x,2)} + \boldsymbol{\kappa}_{3:3}^{e:}\, \boldsymbol{\beta}_{i,*,j}\, X_{ijj}^{(x,2)}$$
$$X_{ijk}^{(x,4)} = \boldsymbol{\varphi}_{3:3}^{(\wedge)1:3}\, X_{ijk}^{(x,3)}$$
$$X_{ijk}^{(x)} = \boldsymbol{\varphi}_{:1}^{(\wedge)2:3}\, X_{ijk}^{(x,4)}$$

Finally, we save the reachability pattern in channel 2 of $X_{iij}$; this information effectively conveys, for each subgraph, the membership of each node to that specific subgraph. At the same time, we bring all other orbit tensors to the original dimension $d$:

$$X_{iii} = \boldsymbol{\kappa}_{:d}^{:d}\, X_{iii}^{(x)}$$
$$X_{ijj} = \boldsymbol{\kappa}_{:d}^{:d}\, X_{ijj}^{(x)}$$
$$X_{iij} = \boldsymbol{\kappa}_{:1/(d)}^{:1}\, X_{iij}^{(x)} + \boldsymbol{\kappa}_{2:2/(d)}^{e:}\, \boldsymbol{\beta}_{i,i,j}\, X_{ijj}^{(x)}$$
$$X_{iji} = \boldsymbol{\kappa}_{:d}^{:d}\, X_{iji}^{(x)}$$
$$X_{ijk} = \boldsymbol{\kappa}_{:d}^{:d}\, X_{ijk}^{(x)}$$

[**Retaining original connectivity**] In all the derivations above we have overwritten the first channel of orbit representations $X_{iij}, X_{iji}, X_{ijk}$ with the computed subgraph connectivity. However, certain Subgraph GNNs may require to retain the original graph connectivity, see, e.g., Equations (9) and (12). In that case it is sufficient to first replicate the first channel of the aforementioned orbit representations into another one before altering it to obtain the subgraph connectivity. $\qquad\square$

### B.3 Implementing Subgraph GNN layers

Before proceeding to prove Lemma 5, we report the equation of a GNN layer in the form due to Morris et al. [36], which we assume is the base-encoder of the Subgraph GNN layers considered in the following.

$$x_i^{(t+1)} = \sigma\big(W_{1,t} \cdot x_i^{(t)} + W_{2,t} \cdot \sum_{j \sim i} x_j^{(t)}\big) \tag{22}$$

*Proof of Lemma 5.* We will implement the update equations for a generic bag $B \in \{B_1^{(t)}, B_2^{(t)}\}$ represented by $\mathbb{R}^{n^3 \times d}$ tensor $\mathcal{Y}^{(t)} \cong X_{iii}^{(t)} \sqcup X_{ijj}^{(t)} \sqcup X_{iij}^{(t)} \sqcup X_{iji}^{(t)} \sqcup X_{ijk}^{(t)}$. When necessary, we will

use an additional subscript to indicate which of the two input bags the tensor is representing, as in $X^{(t)}_{(1),ijk}$. In the following we assume $B_1$ is the bag for graph $G_1$ of $n_1$ nodes, $B_2$ is the bag for graph $G_2$ of $n_2$ nodes.

[**DS-GNN**] When equipped with Morris et al. [36] base-encoder, DS-GNN updates the representation of node $i$ in subgraph $k$ as:

$$x_i^{k,(t+1)} = \sigma\big(W_{1,t} \cdot x_i^{k,(t)} + W_{2,t} \cdot \sum_{j \sim_k i} x_j^{k,(t)}\big) \tag{23}$$

([1] *Message broadcasting*) One first `3-IGN` layer propagates the current node representations in a way to prepare them for the following aggregation. Node representations $X_{ijj}$ are written over $X_{ijk}, X_{iij}$; the ones in $X_{iii}$ over $X_{iji}$: This will allow matching them with the subgraph connectivity stored in the first channel of such tensors.

$$X_{iij}^{(t,1)} = \boldsymbol{\kappa}_{:d/(2d)}^{:d} \, X_{iij}^{(t)} + \boldsymbol{\kappa}_{d+1:2d}^{:} \, \boldsymbol{\beta}_{i,i,j} \, X_{ijj}^{(t)}$$
$$X_{iji}^{(t,1)} = \boldsymbol{\kappa}_{:d/(2d)}^{:d} \, X_{iji}^{(t)} + \boldsymbol{\kappa}_{d+1:2d}^{:} \, \boldsymbol{\beta}_{i,*,i} \, X_{iii}^{(t)}$$
$$X_{ijk}^{(t,1)} = \boldsymbol{\kappa}_{:d/(2d)}^{:d} \, X_{ijk}^{(t)} + \boldsymbol{\kappa}_{d+1:2d}^{:} \, \boldsymbol{\beta}_{i,*,j} \, X_{ijj}^{(t)}$$

([2] *Message sparsification & aggregation*) `3-IGN`s only possess *global* pooling as computational primitive, while message-passing requires a form of local aggregation in accordance to the connectivity at hand. For each node, we realise this by first nullifying messages from non-adjacent nodes followed by global summation. We make use of a result by Yun et al. [57] to show that the aforementioned nullification, over the two bags in input, can be exactly implemented by a (small) ReLU network memorising a properly assembled dataset. Let us first report the result of interest [57, Theorem 3.1]:

**Theorem 13** (Theorem 3.1 from Yun et al. [57]). *Consider any datasaset $\{(x_i, y_i)\}_{i=1}^N$ such that all $x_i$'s are distinct and all $y_i \in [-1, +1]^{d_y}$. If a 3-layer ReLU-like MLP $f_\theta$ satisfies $4\lfloor d_1/4 \rfloor \lfloor d_2/(4d_y) \rfloor \geq N$, then there exists a parameter $\theta$ such that $y_i = f_\theta(x_i)$ for all $i \in [N]$.*

The theorem guarantees the existence of a properly sized ReLU network able to memorise an input dataset satisfying the reported conditions. We note that one of these conditions can, in a sense, be relaxed:

**Proposition 14** (Memorisation). *Consider any dataset $D = \{(x_i, y_i)\}_{i=1}^N$ such that all $x_i$'s $\in \mathbb{R}^{d_x}$ are distinct and all $y_i \in \mathbb{R}^{d_y}$. There exists a 3-layer ReLU-like MLP $\varphi^{(D)}$ such that $y_i = \varphi^{(D)}(x_i)$ for all $i \in [N]$.*

*Proof.* Let $M = \max\{\max_{i \in [N]} |y_i|, \mathbf{1}_{d_y}\}$, where $\max$ is intended to be applied element-wise. Let $\tilde{D} = \{(x_i, \tilde{y}_i) \,|\, \tilde{y}_i = y_i \oslash M\}_{i=1}^N$, where $\oslash$ is element-wise division. Dataset $\tilde{D}$ satisfies the assumptions in Theorem 13, as targets $\tilde{y}_i$ are now all necessarily in $[-1, +1]^{d_y}$. Hence, there exists a 3-layer ReLU MLP $f_\theta$ memorising $\tilde{D}$. Let $(W_l, b_l)$ refer to its weight matrix and bias vector at the $l$-th layer, $l = 1, 2, 3$. Let $\bar{W} = \mathrm{diag}(M)$, i.e. a diagonal matrix in $\mathbb{R}^{d_y \times d_y}$ such that $\bar{W}_{ii} = M_i, i \in [d_y]$. To conclude the proof it is sufficient to construct a 3-layer ReLU MLP $\varphi^{(D)}$ with parameter stacking $\{(W_l^D, b_l^D)\}_{l=1,2,3}$ such that: $(W_1^D, b_1^D) = (W_1, b_1), (W_2^D, b_2^D) = (W_2, b_2), (W_3^D, b_3^D) = (\bar{W} \cdot W_3, \bar{W} \cdot b_3)$. $\square$

We now continue our proof. Formally, we seek to find two MLPs $\varphi^{(\odot_{iij})}, \varphi^{(\odot_{ijk})}$ implementing, respectively, functions $f_{iij}^\odot, f_{ijk}^\odot$ satisfying the following. $f_{iij}^\odot$ is such that $\forall a, b \in [n], a \neq b$:

$$f_{iij}^\odot\big(X_{aab}^{(t,1)}\big) = \begin{cases} \mathbf{0}_d & \text{if } X_{aab}^{(t,1),1} = 0, \\ X_{aab}^{(t,1),d+1:} & \text{otherwise.} \end{cases}$$

Likewise, $f_{ijk}^\odot$ is such that: $\forall a, b, c \in [n], a \neq b \neq c$:

$$f_{ijk}^\odot\big(X_{abc}^{(t,1)}\big) = \begin{cases} \mathbf{0}_d & \text{if } X_{abc}^{(t,1),1} = 0, \\ X_{abc}^{(t,1),d+1:} & \text{otherwise.} \end{cases}$$

We construct two datasets:

$$D_{iij} = \left\{ \left(x, f_{iij}^{\odot}(x)\right) \mid x = X_{(1),aab}^{(t,1)} \;\; \forall a,b \in [n_1], a \neq b \right\} \cup$$

$$\cup \left\{ \left(x, f_{iij}^{\odot}(x)\right) \mid x = X_{(2),aab}^{(t,1)} \;\; \forall a,b \in [n_2], a \neq b \right\}$$

$$D_{ijk} = \left\{ \left(x, f_{ijk}^{\odot}(x)\right) \mid x = X_{(1),abc}^{(t,1)} \;\; \forall a,b,c \in [n_1], a \neq b \neq c \right\} \cup$$

$$\cup \left\{ \left(x, f_{ijk}^{\odot}(x)\right) \mid x = X_{(2),abc}^{(t,1)} \;\; \forall a,b,c \in [n_2], a \neq b \neq c \right\}$$

Here, as all targets are the output of a well-defined function, these datasets satisfy, by construction, the hypothesis of Preposition 14, which we apply on both. This guarantees the existence of $\varphi^{(\odot_{iij})}$, $\varphi^{(\odot_{ijk})}$; their application allows global pooling to effectively recover sparse message aggregation. We also notice that, when updating representations of non-root nodes, roots may be amongst their neighbours, so that it may be needed to additionally sum their representations in $X_{iii}$ to $X_{ijj}$, conditioned on the subgraph connectivity information stored in $X_{iji}$. Accordingly, let us define function $f_{iji}^{\odot}$:

$$f_{iji}^{\odot}\left(X_{aba}^{(t,1)}\right) = \begin{cases} \mathbf{0}_d & \text{if } X_{aba}^{(t,1),1} = 0, \\ X_{aba}^{(t,1),d+1:} & \text{otherwise.} \end{cases}$$

We construct dataset:

$$D_{iji} = \left\{ \left(x, f_{iji}^{\odot}(x)\right) \mid x = X_{(1),aba}^{(t,1)} \;\; \forall a,b \in [n_1], a \neq b \right\} \cup$$

$$\cup \left\{ \left(x, f_{iji}^{\odot}(x)\right) \mid x = X_{(2),aba}^{(t,1)} \;\; \forall a,b \in [n_2], a \neq b \right\}$$

Invoking Proposition 14 on $D_{iji}$ guarantees the existence of $\varphi^{(\odot_{iji})}$ memorising $D_{iji}$. We let the described 3-IGN implement such networks:

$$X_{iij}^{(t,2)} = [\boldsymbol{\kappa}_{:d}^{:d} \;\; \boldsymbol{\varphi}_{d+1:}^{(\odot_{iij}):}] X_{iij}^{(t,1)}$$

$$X_{iji}^{(t,2)} = [\boldsymbol{\kappa}_{:d}^{:d} \;\; \boldsymbol{\varphi}_{d+1:}^{(\odot_{iji}):}] X_{iji}^{(t,1)}$$

$$X_{ijk}^{(t,2)} = [\boldsymbol{\kappa}_{:d}^{:d} \;\; \boldsymbol{\varphi}_{d+1:}^{(\odot_{ijk}):}] X_{ijk}^{(t,1)}$$

This last layer completes message aggregation:

$$X_{iii}^{(t,3)} = \boldsymbol{\kappa}_{:d}^{:d} X_{iii}^{(t,2)} + \boldsymbol{\kappa}_{d+1:}^{d+1:} \boldsymbol{\beta}_{iii} \boldsymbol{\pi}_i X_{iij}^{(t,2)} \tag{24}$$

$$X_{ijj}^{(t,3)} = \boldsymbol{\kappa}_{:d}^{:d} X_{ijj}^{(t,2)} + \boldsymbol{\kappa}_{d+1:}^{d+1:} \boldsymbol{\beta}_{ijj} X_{iji}^{(t,2)} + \boldsymbol{\kappa}_{d+1:}^{d+1:} \boldsymbol{\beta}_{ijj} \boldsymbol{\pi}_{ij} X_{ijk}^{(t,2)} \tag{25}$$

([3] *Update*) We finally describe the statements implementing linear transformations operated by parameters $W_{1,t}, W_{2,t}$, other than bringing the other orbit representations to dimension $d$:

$$X_{iii}^{(t+1)} = \sigma\left([W_{1,t} \,\|\, W_{2,t}] X_{iii}^{(t,3)}\right) \tag{26}$$

$$X_{ijj}^{(t+1)} = \sigma\left([W_{1,t} \,\|\, W_{2,t}] X_{ijj}^{(t,3)}\right) \tag{27}$$

$$X_{iij}^{(t+1)} = \boldsymbol{\kappa}_{:1/(d)}^{:1} X_{iij}^{(t,3)}$$

$$X_{iji}^{(t+1)} = \boldsymbol{\kappa}_{:1/(d)}^{:1} X_{iji}^{(t,3)}$$

$$X_{ijk}^{(t+1)} = \boldsymbol{\kappa}_{:1/(d)}^{:1} X_{ijk}^{(t,3)}$$

[**DSS-GNN**] When equipped with Morris et al. [36] base-encoder, DSS-GNN updates representation of node $i$ in subgraph $k$ as:

$$x_i^{k,(t+1)} = \sigma\left(W_{1,t}^1 \cdot x_i^{k,(t)} + W_{2,t}^1 \cdot \sum_{j \sim_k i} x_j^{k,(t)} + W_{1,t}^2 \cdot \sum_h x_i^{h,(t)} + W_{2,t}^2 \cdot \sum_{j \sim i} \sum_h x_j^{h,(t)}\right) \tag{28}$$

([0] *Cross-bag aggregation*) We start by performing those operations above of the form $\sum_h$:

$$X_{iii}^{(t,0)} = [I_d \ I_d] \, X_{iii}^{(t)} + \boldsymbol{\kappa}_{d+1:2d}^{:} \, \boldsymbol{\beta}_{j,j,j} \, \boldsymbol{\pi}_j \, X_{ijj}^{(t)}$$

$$X_{ijj}^{(t,0)} = \boldsymbol{\kappa}_{:d}^{:} \, X_{ijj}^{(t)} + \boldsymbol{\kappa}_{d+1:2d}^{:} \, \boldsymbol{\beta}_{*,j,j} \, \boldsymbol{\pi}_j \, X_{ijj}^{(t)} + \boldsymbol{\kappa}_{d+1:2d}^{:} \, \boldsymbol{\beta}_{*,i,i} \, X_{iii}^{(t)}$$

([1] *Message broadcasting*) Similarly as in DS-GNN, we propagate node representations — and their cross-bag aggregated counterparts — on those orbits storing (sub)graph connectivity.

$$X_{iij}^{(t,1)} = \boldsymbol{\kappa}_{:d/(3d)}^{:d} \, X_{iij}^{(t,0)} + \boldsymbol{\kappa}_{d+1:3d}^{:} \, \boldsymbol{\beta}_{i,i,j} \, X_{ijj}^{(t,0)}$$

$$X_{iji}^{(t,1)} = \boldsymbol{\kappa}_{:d/(3d)}^{:d} \, X_{iji}^{(t,0)} + \boldsymbol{\kappa}_{d+1:3d}^{:} \, \boldsymbol{\beta}_{i,*,i} \, X_{iii}^{(t,0)}$$

$$X_{ijk}^{(t,1)} = \boldsymbol{\kappa}_{:d/(3d)}^{:d} \, X_{ijk}^{(t,0)} + \boldsymbol{\kappa}_{d+1:3d}^{:} \, \boldsymbol{\beta}_{i,*,j} \, X_{ijj}^{(t,0)}$$

([2] *Message sparsification & aggregation*) We now follow the same rationale as for DS-GNN, and construct datasets that allow the invocation of Proposition 14. This will guarantee the existence of an MLP that can be applied to retain, for each node, only those messages coming from direct neighbours, according to the subgraph connectivity (stored in channel 1) or the original one (which we assume to be stored in channel 2). Precisely, we would like to memorise the following functions:

$$f_{iij}^{\odot;\sim i}\big(X_{aab}^{(t,1)}\big) = \begin{cases} \mathbf{0}_d & \text{if } X_{aab}^{(t,1),1} = 0, \\ X_{aab}^{(t,1),d+1:2d} & \text{otherwise.} \end{cases} \quad f_{iji}^{\odot;\sim i}\big(X_{aba}^{(t,1)}\big) = \begin{cases} \mathbf{0}_d & \text{if } X_{aba}^{(t,1),1} = 0, \\ X_{aba}^{(t,1),d+1:2d} & \text{otherwise.} \end{cases}$$

$$f_{ijk}^{\odot;\sim i}\big(X_{abc}^{(t,1)}\big) = \begin{cases} \mathbf{0}_d & \text{if } X_{abc}^{(t,1),1} = 0, \\ X_{abc}^{(t,1),d+1:2d} & \text{otherwise.} \end{cases}$$

$$f_{iij}^{\odot;\sim}\big(X_{aab}^{(t,1)}\big) = \begin{cases} \mathbf{0}_d & \text{if } X_{aab}^{(t,1),2} = 0, \\ X_{aab}^{(t,1),2d+1:} & \text{otherwise.} \end{cases} \quad f_{iji}^{\odot;\sim}\big(X_{aba}^{(t,1)}\big) = \begin{cases} \mathbf{0}_d & \text{if } X_{aba}^{(t,1),2} = 0, \\ X_{aba}^{(t,1),2d+1:} & \text{otherwise.} \end{cases}$$

$$f_{ijk}^{\odot;\sim}\big(X_{abc}^{(t,1)}\big) = \begin{cases} \mathbf{0}_d & \text{if } X_{abc}^{(t,1),2} = 0, \\ X_{abc}^{(t,1),2d+1:} & \text{otherwise.} \end{cases}$$

and construct the corresponding datasets:

$$D_{iij}^{\sim i} = \Big\{ \big(x, f_{iij}^{\odot;\sim i}(x)\big) \,\big|\, x = X_{(1),aab}^{(t,1)} \ \forall a,b \in [n_1], a \neq b \Big\} \cup$$

$$\cup \Big\{ \big(x, f_{iij}^{\odot;\sim i}(x)\big) \,\big|\, x = X_{(2),aab}^{(t,1)} \ \forall a,b \in [n_2], a \neq b \Big\}$$

$$D_{iji}^{\sim i} = \Big\{ \big(x, f_{iji}^{\odot;\sim i}(x)\big) \,\big|\, x = X_{(1),aba}^{(t,1)} \ \forall a,b \in [n_1], a \neq b \Big\} \cup$$

$$\cup \Big\{ \big(x, f_{iij}^{\odot;\sim i}(x)\big) \,\big|\, x = X_{(2),aba}^{(t,1)} \ \forall a,b \in [n_2], a \neq b \Big\}$$

$$D_{ijk}^{\sim i} = \Big\{ \big(x, f_{ijk}^{\odot;\sim i}(x)\big) \,\big|\, x = X_{(1),abc}^{(t,1)} \ \forall a,b,c \in [n_1], a \neq b \neq c \Big\} \cup$$

$$\cup \Big\{ \big(x, f_{ijk}^{\odot;\sim i}(x)\big) \,\big|\, x = X_{(2),abc}^{(t,1)} \ \forall a,b,c \in [n_2], a \neq b \neq c \Big\}$$

$$D_{iij}^{\sim} = \Big\{ \big(x, f_{iij}^{\odot;\sim}(x)\big) \,\big|\, x = X_{(1),aab}^{(t,1)} \ \forall a,b \in [n_1], a \neq b \Big\} \cup$$

$$\cup \Big\{ \big(x, f_{iij}^{\odot;\sim}(x)\big) \,\big|\, x = X_{(2),aab}^{(t,1)} \ \forall a,b \in [n_2], a \neq b \Big\}$$

$$D_{iji}^{\sim} = \Big\{ \big(x, f_{iji}^{\odot;\sim}(x)\big) \,\big|\, x = X_{(1),aba}^{(t,1)} \ \forall a,b \in [n_1], a \neq b \Big\} \cup$$

$$\cup \Big\{ \big(x, f_{iij}^{\odot;\sim}(x)\big) \,\big|\, x = X_{(2),aba}^{(t,1)} \ \forall a,b \in [n_2], a \neq b \Big\}$$

$$D_{ijk}^{\sim} = \Big\{ \big(x, f_{ijk}^{\odot;\sim}(x)\big) \,\big|\, x = X_{(1),abc}^{(t,1)} \ \forall a,b,c \in [n_1], a \neq b \neq c \Big\} \cup$$

$$\cup \Big\{ \big(x, f_{ijk}^{\odot;\sim}(x)\big) \,\big|\, x = X_{(2),abc}^{(t,1)} \ \forall a,b,c \in [n_2], a \neq b \neq c \Big\}$$

These, by Proposition 14, are memorised by, respectively, MLPs $\varphi^{(\odot_{iij}^{\widetilde{i}})}$, $\varphi^{(\odot_{iji}^{\widetilde{i}})}$, $\varphi^{(\odot_{ijk}^{\widetilde{i}})}$, $\varphi^{(\odot_{iij}^{\widetilde{i}})}$, $\varphi^{(\odot_{iji}^{\widetilde{i}})}$, $\varphi^{(\odot_{ijk}^{\widetilde{i}})}$. We let our 3-IGN model apply these:

$$X_{iij}^{(t,2)} = \begin{bmatrix} \boldsymbol{\kappa}_{:d}^{:d} & \varphi_{d+1:2d}^{(\odot_{iij}^{\widetilde{i}}):} & \varphi_{2d+1:}^{(\odot_{iij}^{\widetilde{i}}):} \end{bmatrix} X_{iij}^{(t,1)}$$

$$X_{iji}^{(t,2)} = \begin{bmatrix} \boldsymbol{\kappa}_{:d}^{:d} & \varphi_{d+1:2d}^{(\odot_{iji}^{\widetilde{i}}):} & \varphi_{2d+1:}^{(\odot_{iji}^{\widetilde{i}}):} \end{bmatrix} X_{iji}^{(t,1)}$$

$$X_{ijk}^{(t,2)} = \begin{bmatrix} \boldsymbol{\kappa}_{:d}^{:d} & \varphi_{d+1:2d}^{(\odot_{ijk}^{\widetilde{i}}):} & \varphi_{2d+1:}^{(\odot_{ijk}^{\widetilde{i}}):} \end{bmatrix} X_{ijk}^{(t,1)}$$

It is only left to aggregate messages via global pooling:

$$X_{iii}^{(t,3)} = \boldsymbol{\kappa}_{:2d/(4d)}^{:2d} X_{iii}^{(t,2)} + \boldsymbol{\kappa}_{2d+1:4d}^{d+1:} \boldsymbol{\beta}_{iii} \boldsymbol{\pi}_i X_{iij}^{(t,2)}$$

$$X_{ijj}^{(t,3)} = \boldsymbol{\kappa}_{:2d/(4d)}^{:2d} X_{ijj}^{(t,2)} + \boldsymbol{\kappa}_{2d+1:4d}^{d+1:} \boldsymbol{\beta}_{ijj} X_{iji}^{(t,2)} + \boldsymbol{\kappa}_{2d+1:4d}^{d+1:} \boldsymbol{\beta}_{ijj} \boldsymbol{\pi}_{ij} X_{ijk}^{(t,2)}$$

([4] *Update*) We describe the statements implementing the final linear transformations:

$$X_{iii}^{(t+1)} = \sigma\left( [W_{1,t}^1 \| W_{1,t}^2 \| W_{2,t}^1 \| W_{2,t}^2] X_{iii}^{(t,3)} \right)$$

$$X_{ijj}^{(t+1)} = \sigma\left( [W_{1,t}^1 \| W_{1,t}^2 \| W_{2,t}^1 \| W_{2,t}^2] X_{ijj}^{(t,3)} \right)$$

$$X_{iij}^{(t+1)} = \boldsymbol{\kappa}_{:2/(d)}^{:2} X_{iij}^{(t,3)}$$

$$X_{iji}^{(t+1)} = \boldsymbol{\kappa}_{:2/(d)}^{:2} X_{iji}^{(t,3)}$$

$$X_{ijk}^{(t+1)} = \boldsymbol{\kappa}_{:2/(d)}^{:2} X_{ijk}^{(t,3)}$$

[**GNN-AK-ctx**] When equipped with Morris et al. [36] base-encoder, GNN-AK-ctx updates representation of node $i$ in subgraph $k$ as:

$$x_i^{k,(t,0)} = x_i^{k,(t)}$$

$$x_i^{k,(t,l+1)} = \sigma\left( W_{1,t,l} \cdot x_i^{k,(t,l)} + W_{2,t,l} \cdot \sum_{j \sim_k i} x_j^{k,(t,l)} \right), \quad l = 0, \dots, L-1 \qquad [S] \qquad (29)$$

$$x_i^{k,(t+1)} = x_i^{i,(t,L)} + \sum_j x_i^{j,(t,L)} + \sum_j x_j^{i,(t,L)} \qquad\qquad\qquad [A] \qquad (30)$$

([$S$]) In order to implement block [$S$], it is sufficient to repeat steps [1–3] in the DS-GNN derivation $L$ times, i.e. the desired number of message-passing steps. We obtain representations $X^{(t,L)}$.

([$A$]) Block [$A$] is implemented as:

$$X_{iii}^{(t+1)} = 3 \cdot X_{iii}^{(t,L)} + \boldsymbol{\beta}_{i,i,i} \boldsymbol{\pi}_i X_{ijj}^{(t,L)} + \boldsymbol{\beta}_{j,j,j} \boldsymbol{\pi}_j X_{ijj}^{(t,L)}$$

$$X_{ijj}^{(t+1)} = 3 \cdot \boldsymbol{\beta}_{*,i,i} \cdot X_{iii}^{(t,L)} + \boldsymbol{\beta}_{*,i,i} \boldsymbol{\pi}_i X_{ijj}^{(t,L)} + \boldsymbol{\beta}_{*,j,j} \boldsymbol{\pi}_j X_{ijj}^{(t,L)}$$

In the original paper [61], the second and third terms in block [$A$] only operate on the nodes which are members of the ego-networks at hand:

$$x_i^{k,(t+1)} = x_i^{i,(t,L)} + \sum_{j \in V^i} x_i^{j,(t,L)} + \sum_{j \in V^i} x_j^{i,(t,L)} \qquad\qquad [A]$$

We show that 3-IGNs can implement this block formulation as well, by resorting to the same sparsification technique employed in the derivation of DS-GNN. Let us recall that, as already mentioned above, the *ego-networks* policy can store reachability patterns in orbit representation $X_{iij}$: they convey, for each node $j$, its membership to subgraph $i$. This information can be used to sparsify node representations being aggregated by the global pooling operations taking place in the equations above. We start by placing node representations $X_{ijj}$ onto $X_{iij}, X_{iji}$. Let us also replicate reachability patterns into the second channel of $X_{iji}$.

$$X_{iij}^{(t,p)} = \boldsymbol{\kappa}_{:d}^{:d} X_{iij}^{(t,L)} + \boldsymbol{\kappa}_{d+1:2d}^{:d} \boldsymbol{\beta}_{i,i,j} X_{ijj}^{(t,L)}$$

$$X_{iji}^{(t,p)} = \boldsymbol{\kappa}_{:d}^{:d} X_{iji}^{(t,L)} + \boldsymbol{\kappa}_{2:2}^{2:2} \boldsymbol{\beta}_{i,j,i} X_{iij}^{(t,L)} + \boldsymbol{\kappa}_{d+1:2d}^{:d} \boldsymbol{\beta}_{j,i,j} X_{ijj}^{(t,L)}$$

We would like to memorise the following sparsification functions:

$$f_{iij}^{\odot,V^i}\big(X_{aab}^{(t,p)}\big) = \begin{cases} \mathbf{0}_d & \text{if } X_{aab}^{(t,p),2} = 0, \\ X_{aab}^{(t,p),d+1:} & \text{otherwise.} \end{cases} \quad f_{iji}^{\odot,V^i}\big(X_{aba}^{(t,p)}\big) = \begin{cases} \mathbf{0}_d & \text{if } X_{aba}^{(t,p),2} = 0, \\ X_{aba}^{(t,p),d+1:} & \text{otherwise.} \end{cases}$$

so we construct the following datasets:

$$D_{iij}^{V^i} = \Big\{ \big(x, f_{iij}^{\odot,V^i}(x)\big) \,\big|\, x = X_{(1),aab}^{(t,p)} \ \forall a,b \in [n_1], a \neq b \Big\} \cup$$
$$\cup \Big\{ \big(x, f_{iij}^{\odot,V^i}(x)\big) \,\big|\, x = X_{(2),aab}^{(t,p)} \ \forall a,b \in [n_2], a \neq b \Big\}$$
$$D_{iji}^{V^i} = \Big\{ \big(x, f_{iji}^{\odot,V^i}(x)\big) \,\big|\, x = X_{(1),aba}^{(t,p)} \ \forall a,b \in [n_1], a \neq b \Big\} \cup$$
$$\cup \Big\{ \big(x, f_{iji}^{\odot,V^i}(x)\big) \,\big|\, x = X_{(2),aba}^{(t,p)} \ \forall a,b \in [n_2], a \neq b \Big\}$$

Again, we invoke Proposition 14, which guarantees the existence of MLPs $\varphi^{(\odot_{iij}^{V^i})}, \varphi^{(\odot_{iji}^{V^i})}$. Let the 3-IGN model implement them:

$$X_{iij}^{(t,p+1)} = [\boldsymbol{\kappa}_{:d}^{:d} \ \ \varphi_{d+1:2d}^{(\odot_{iij}^{V^i}):}] \, X_{iij}^{(t,p)}$$
$$X_{iji}^{(t,p+1)} = [\boldsymbol{\kappa}_{:d}^{:d} \ \ \varphi_{d+1:2d}^{(\odot_{iji}^{V^i}):}] \, X_{iji}^{(t,p)}$$

Then, we perform the last global pooling step to complete the implementation of block $[A]$:

$$X_{iii}^{(t+1)} = 3 \cdot \boldsymbol{\kappa}_{:d}^{:d} X_{iii}^{(t,p+1)} + \boldsymbol{\kappa}_{:d}^{d+1:2d} \boldsymbol{\beta}_{i,i,i} \boldsymbol{\pi}_i X_{iij}^{(t,p+1)} + \boldsymbol{\kappa}_{:d}^{d+1:2d} \boldsymbol{\beta}_{i,i,i} \boldsymbol{\pi}_i X_{iji}^{(t,p+1)} \tag{31}$$
$$X_{ijj}^{(t+1)} = 3 \cdot \boldsymbol{\kappa}_{:d}^{:d} \boldsymbol{\beta}_{*,i,i} X_{iii}^{(t,p+1)} + \boldsymbol{\kappa}_{:d}^{d+1:2d} \boldsymbol{\beta}_{*,i,i} \boldsymbol{\pi}_i X_{iij}^{(t,p+1)} + \boldsymbol{\kappa}_{:d}^{d+1:2d} \boldsymbol{\beta}_{*,i,i} \boldsymbol{\pi}_i X_{iji}^{(t,p+1)} \tag{32}$$
$$X_{iij}^{(t+1)} = \boldsymbol{\kappa}_{1:2/(d)}^{1:2} X_{iij}^{(t,p+1)}$$
$$X_{iji}^{(t+1)} = \boldsymbol{\kappa}_{1:1/(d)}^{1:1} X_{iji}^{(t,p+1)}$$
$$X_{ijk}^{(t+1)} = \boldsymbol{\kappa}_{1:1/(d)}^{1:1} X_{ijk}^{(t,p+1)}$$

[**GNN-AK**] In the case of $[A]$ operating only on those nodes in the ego-networks at hand, it is sufficient to rewrite Equations 31, 32 as:

$$X_{iii}^{(t+1)} = 2 \cdot \boldsymbol{\kappa}_{:d}^{:d} X_{iii}^{(t,p+1)} + \boldsymbol{\kappa}_{:d}^{d+1:2d} \boldsymbol{\beta}_{i,i,i} \boldsymbol{\pi}_i X_{iij}^{(t,p+1)}$$
$$X_{ijj}^{(t+1)} = 2 \cdot \boldsymbol{\kappa}_{:d}^{:d} \boldsymbol{\beta}_{*,i,i} X_{iii}^{(t,p+1)} + \boldsymbol{\kappa}_{:d}^{d+1:2d} \boldsymbol{\beta}_{*,i,i} \boldsymbol{\pi}_i X_{iij}^{(t,p+1)}$$

These equations would also implement the more general block $[A]$ in Equation (30).

[**ID-GNN**] With Morris et al. [36] base-encoder, ID-GNN updates node representations as:

$$x_i^{k,(t+1)} = \sigma\big(W_{1,t} x_i^{k,(t)} + W_{2,t} \sum_{j \sim_k i, j \neq k} x_j^{k,(t)} + \mathbb{1}_{[k \sim_k i]} \cdot W_{3,t} x_k^{k,(t)}\big) \tag{33}$$

Message passing is performed according to the same 3-IGN programme as in DS-GNN, with the only modifications required to Equations 24,25,26,27. Equations 24 and 25 are rewritten as:

$$X_{iii}^{(t,3)} = \boldsymbol{\kappa}_{:d}^{:d} X_{iii}^{(t,2)} + \boldsymbol{\kappa}_{d+1:}^{:d} W_{2,t} \boldsymbol{\kappa}_{:d}^{d+1:} \boldsymbol{\beta}_{iii} \boldsymbol{\pi}_i X_{iij}^{(t,2)}$$
$$X_{ijj}^{(t,3)} = \boldsymbol{\kappa}_{:d}^{:d} X_{ijj}^{(t,2)} + \boldsymbol{\kappa}_{d+1:}^{:d} W_{3,t} \boldsymbol{\kappa}_{:d}^{d+1:} \boldsymbol{\beta}_{ijj} X_{iji}^{(t,2)} + \boldsymbol{\kappa}_{d+1:}^{:d} W_{2,t} \boldsymbol{\kappa}_{:d}^{d+1:} \boldsymbol{\beta}_{ijj} \boldsymbol{\pi}_{ij} X_{ijk}^{(t,2)}$$

whereas Equations 26, 27 as:

$$X_{iii}^{(t+1)} = \sigma\big([W_{1,t} \,\|\, I_d] X_{iii}^{(t,3)}\big)$$
$$X_{ijj}^{(t+1)} = \sigma\big([W_{1,t} \,\|\, I_d] X_{ijj}^{(t,3)}\big)$$

[**NGNN**] Using a Morris et al. [36] base-encoder, the update equation for the *inner* siamese GNN in a Nested GNN [59] exactly match that of Equation 23. It is therefore sufficient for the 3-IGN to execute the same programme employed in the DS-GNN derivation. $\qquad\square$

## B.4 Upperbounding Subgraph GNNs

*Proof of Thereom 6.* Subgraph GNN $\mathcal{N}_\Theta$ distinguishes $G_1, G_2$ if they are assigned distinct representations, that is: $y_{G_1} = \mathcal{N}_\Theta(A_1, X_1) \neq \mathcal{N}_\Theta(A_2, X_2) = y_{G_2}$. Naturally, a 3-IGN instance $\mathcal{M}_\Omega$ implementing $\mathcal{N}_\Theta$ on the same pair of graphs would distinguish them as well. We prove the theorem by showing that such an instance exists.

We seek to find a 3-IGN model $\mathcal{M}_\Omega$ in the form of Equation 16 such that: $\mathcal{M}_\Omega(A_1, X_1) = \mathcal{N}_\Theta(A_1, X_1) = y_{G_1}$ and $\mathcal{M}_\Omega(A_2, X_2) = \mathcal{N}_\Theta(A_2, X_2) = y_{G_2}$. According to Equation 1, $\mathcal{N}_\Theta(\cdot) = (\mu \circ \rho \circ \mathcal{S} \circ \pi)_\Theta(\cdot)$. We will show how to construct $\mathcal{M}_\Omega$ as an appropriate stacking of 3-IGN layers exactly implementing each of the components $\pi, \mathcal{S}, \rho, \mu$ when applied to graphs $G_1, G_2$. We assume, w.l.o.g., that stacking $\mathcal{S}$ has the form $\mathcal{S} = L^{(T)} \circ L^{(T-1)} \circ \ldots \circ L^{(1)}$, where $L$'s are $\mathcal{N}$-layers.

By the definition of class $\Upsilon$, $\pi$ in $\mathcal{N}_\Theta$ is such that $\pi \in \Pi$, thus, by Lemma 4, there exists a stacking of 3-IGN layers $\mathcal{M}_\pi$ implementing $\pi$. At the same time, for each $\mathcal{N}$-layer $L^{(t)}$, Lemma 5 has its hypotheses satisfied, hence there exists a 3-IGN-stacking $\mathcal{M}^{(t)}$ implementing $L^{(t)}$ on both $G_1, G_2$. We can compose such stacks so that, overall, we have: $(\mathcal{M}^{(T)} \circ \ldots \circ \mathcal{M}^{(1)} \circ \mathcal{M}_\pi) \cong_{\{G_1, G_2\}} (\mathcal{S} \circ \pi)$, $\cong_{\{G_1, G_2\}}$ denoting *implementation* over set $\{G_1, G_2\} \subset \mathcal{G}$.

We are left with implementing blocks $\mu, \rho$. We show this for every Subgraph GNN in $\Upsilon$.

[**DS-GNN & DSS-GNN**] perform graph readout on each subgraph and then apply a Deep Sets network to these obtained representations:

$$x^{k,(T)} = \sum_i x_i^{k,(T)}$$

$$y_G = \psi\Big(\sum_k \phi(x^{k,(T)})\Big)$$

The following instruction implements subgraph readout as a 3- to 1- equivariant layer:

$$X_i^{\rho,(1)} = \boldsymbol{\pi}_i\, X_{ijj}^{(T)} + X_{iii}^{(T)}$$

Transformation $\phi$ is implemented by a stacking of 1-IGN layers:

$$X_i^{\rho,(2)} = \boldsymbol{\varphi}^{(\phi)}\, X_i^{\rho,(1)} \tag{34}$$

Finally, we let module $h$ in the 3-IGN model implement summation $\sum_k$, and choose MLP $m$ in the 3-IGN such that $m \equiv \psi$:

$$x_G = h\big(X_i^{\rho,(2)}\big) = \sum_i X_i^{\rho,(2)} \tag{35}$$

$$y_G = m\big(x_G\big) = \psi\big(x_G\big)$$

It is possible for the DeepSets network to implement a late invariant-aggregation strategy, so that $\mu \circ \rho$ is realised as:

$$x^{k,(T)} = \sum_i x_i^{k,(T)}$$

$$x^{k,(T+1)} = \sigma\big(W_T^1 x^{k,(T)} + \sum_h W_T^2 x^{h,(T)}\big)$$

$$\ldots$$

$$x^{k,(T+L)} = \sigma\big(W_{T+L-1}^1 x^{k,(T+L-1)} + \sum_h W_{T+L-1}^2 x^{h,(T+L-1)}\big)$$

$$y_G = \psi\big(\sum_k x^{k,(T+L)}\big)$$

In this case, it is sufficient to rewrite Equation 34 as:

$$X_i^{\rho,(1+l)} = \sigma\big(W_l^1 X_i^{\rho,(l)} + W_l^2 \boldsymbol{\beta}_* \boldsymbol{\pi}\, X_i^{\rho,(l)}\big)$$

with $l$ ranging from 1 to $L$, so that then Equation (35) becomes:

$$x_G = h\big(X_i^{\rho,(1+L)}\big) = \sum_i X_i^{\rho,(1+L)}$$

[**GNN-AK, GNN-AK-ctx** & **ID-GNN**] do not perform subgraph pooling, rather pool the representations of root nodes directly:

$$y_G = \mu\Big(\sum_h x_h^{h,(T)}\Big)$$

Block $h$ implements pooling on roots:

$$x_G = h(\mathcal{Y}^{(T)}) = \sum_i X_{iii}^{(T)}$$

and it is then sufficient to choose block $m$ such that $m \equiv \mu$.

[**NGNN**], in its most general form, performs $L$ layers of message passing on subgraph pooled representations over the original graph connectivity:

$$x_v^{(T)} = \sum_{w \in V^v} x_w^{v,(T)}$$

$$x_v^{(T+1)} = \sigma\big(W_T^1 x_v^{(T)} + W_T^2 \sum_{w \sim v} x_w^{(T)}\big)$$

$$\dots$$

$$x_v^{(T+L)} = \sigma\big(W_{T+L-1}^1 x_v^{(T+L-1)} + W_{T+L-1}^2 \sum_{w \sim v} x_w^{(T+L-1)}\big)$$

$$y_G = \mu\big(\sum_w x_w^{(T+L)}\big)$$

We assume the original graph connectivity has been retained in the third channel of orbit representations $X_{iij}, X_{iji}, X_{ijk}$, while the second channel in $X_{iij}$ hosts reachability patterns (see Section B.2, [**ego-networks(**$h$**)**] and [**Retaining original connectivity**]). First, we pool representations of nodes in each subgraph, excluding those nodes not belonging to the ego-nets. We need to extend the summation only to those nodes belonging to the ego-networks. This information is stored in the reachability pattern in $X_{iij}$, and we make use of this information to mask node representations before aggregating them. First, we place node representations in $X_{ijj}$ over $X_{iij}$:

$$X_{iij}^{\rho,(1)} = \boldsymbol{\kappa}_{:d/(2d)}^{:d} X_{iij} + \boldsymbol{\kappa}_{d+1:2d}^{:} \boldsymbol{\beta}_{iij} X_{ijj}^{(T)}$$

We note that it is needed to memorise the following sparsification function:

$$f_{iij}^{\odot,V^i}\big(X_{aab}^{\rho,(1)}\big) = \begin{cases} \mathbf{0}_d & \text{if } X_{aab}^{\rho,(1),2} = 0, \\ X_{aab}^{\rho,(1),d+1:} & \text{otherwise.} \end{cases}$$

and construct the following dataset:

$$D_{iij}^{V^i} = \Big\{\big(x, f_{iij}^{\odot,V^i}(x)\big) \,\big|\, x = X_{(1),aab}^{\rho,(1)} \quad \forall a, b \in [n_1], a \neq b\Big\} \cup$$

$$\cup \Big\{\big(x, f_{iij}^{\odot,V^i}(x)\big) \,\big|\, x = X_{(2),aab}^{\rho,(1)} \quad \forall a, b \in [n_2], a \neq b\Big\}$$

Proposition 14 can be invoked, guaranteeing the existence of MLP $\varphi^{(\odot_{iij}^{V^i})}$ memorising such dataset. We let the 3-IGN implement it:

$$X_{iij}^{\rho,(2)} = [\boldsymbol{\kappa}_{:d}^{:d} \quad \boldsymbol{\varphi}_{d+1:2d}^{(\odot_{iij}^{V^i}):}] X_{iij}^{\rho,(1)}$$

and complete the subgraph readout via a global pooling operation:

$$X_{iii}^{\rho,(3)} = \boldsymbol{\kappa}_{:d}^{:d} \, X_{iii}^{\rho,(3)} + \boldsymbol{\kappa}_{:d}^{d+1:} \, \boldsymbol{\beta}_{iii} \, \boldsymbol{\pi}_i \, X_{iij}^{\rho,(2)}$$

At this point it is left to perform message passing on these pooled representations for $L$ steps on the original connectivity. We note that it is sufficient to broadcast these onto $X_{ijj}$ and run the same message passing steps in parallel on each subgraph, using the same original graph connectivity:

$$X_{ijj}^{\rho,(4)} = \boldsymbol{\beta}_{*,i,i} \, X_{iii}^{\rho,(3)}$$

Such message-passing steps are implemented with the same programme provided in the Proof of Lemma 5 for DS-GNN, with the only difference being that the sparsification functions are defined based on the third channel of $X_{iij}, X_{iji}, X_{ijk}$:

$$f_{iij}^{\odot}\big(X_{aab}^{\rho,(4+l)}\big) = \begin{cases} \mathbf{0}_d & \text{if } X_{aab}^{\rho,(4+l),3} = 0, \\ X_{aab}^{\rho,(4+l),d+1:} & \text{otherwise.} \end{cases}$$

$$f_{ijk}^{\odot}\big(X_{abc}^{\rho,(4+l)}\big) = \begin{cases} \mathbf{0}_d & \text{if } X_{abc}^{\rho,(4+l),3} = 0, \\ X_{abc}^{\rho,(4+l),d+1:} & \text{otherwise.} \end{cases}$$

$$f_{iji}^{\odot}\big(X_{aba}^{\rho,(4+l)}\big) = \begin{cases} \mathbf{0}_d & \text{if } X_{aba}^{\rho,(4+l),3} = 0, \\ X_{aba}^{\rho,(4+l),d+1:} & \text{otherwise.} \end{cases}$$

Datasets to be memorised are defined accordingly. This construction is repeated $L$ times. Afterwards, blocks $h$ and $m$ in the `3-IGN` model pool the root representations and apply MLP $\mu$ on the obtained embedding. These are implemented as shown above for GNN-AK, GNN-AK-ctx, ID-GNN. The proof concludes. □

*Proof of Corollary 7.* We prove the corollary by contradiction. Suppose there exist non-isomorphic but `3-WL`-equivalent graphs $G_1, G_2$ distinguished by instance $\mathcal{N}_\Theta$ of $\mathcal{N} \in \Upsilon$. That is, $\mathcal{N}_\Theta(G_1) \neq \mathcal{N}_\Theta(G_2)$. In view of Theorem 6, there must exists a `3-IGN` instance $\mathcal{M}_\Omega$ such that $\mathcal{M}_\Omega(G_1) \neq \mathcal{M}_\Omega(G_2)$. We note that the expressive power of `k-IGNs` has been fully characterised by Azizian and Lelarge [5], Geerts [21]. In particular, let us report Geerts [21, Theorem 2]:

**Theorem 15** (Expressive power of k-IGNs, Theorem 2 in Geerts [21]). *For any two graphs $G_1$ and $G_2$, if `k-WL` does not distinguish $G_1, G_2$ then any `k-IGN` does not distinguish them either, i.e., it assigns $G_1, G_2$ the same (tensorial) representations.*

This theorem equivalently asserts that if there exists a `k-IGN` distinguishing $G_1, G_2$, then these two graphs must be distinguished by the `k-WL` algorithm. Thus, given the existence of `3-IGN` model $\mathcal{M}_\Omega$, the theorem ensures us that `3-WL` distinguishes graphs $G_1, G_2$, against our hypothesis. □

Let us conclude this section by reporting the following

**Remark 16.** *Any Subgraph Network $\mathcal{N} \in \Upsilon$ equipped with policy $\pi_{\mathrm{EGO}(h)}$ (or $\pi_{\mathrm{EGO}+(h)}$) is at most as expressive as `3-WL`, for any $h > 0$.*

In other words, given the results proved above, deeper ego-networks may increase the expressive power of a model, but not in a way to exceed that of `3-WL`.

## C   Illustrated comparison of Subgraph GNNs

In this section we explain in detail Figure 6 (corresponding to Figure 3 in the main paper) by linking the coloured updates of each Subgraph GNN to its corresponding formulation in Appendix A. We consider grids of $n$ subgraphs with $n$ nodes. The figure shows the aggregation and update rules in Subgraph GNNs for both diagonal $(i, i)$ and off-diagonal $(k, i)$ entries, which correspond respectively to updates of root and non-root nodes. Each color represents a different parameter. We use squares to indicate global pooling and triangles for local pooling.

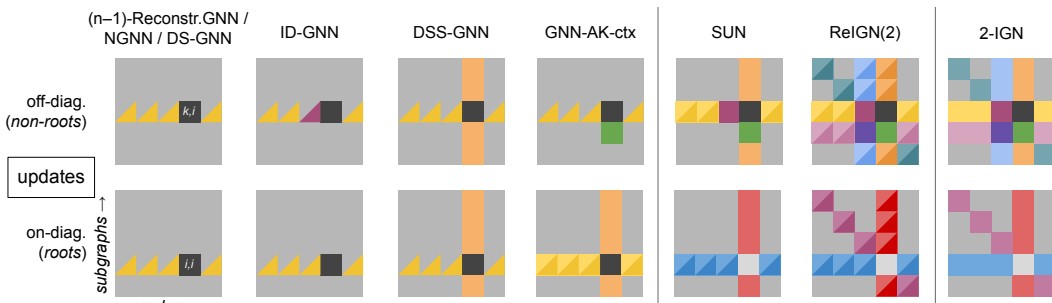

Figure 6: A comparison of aggregation and update rules in Subgraph GNNs, illustrated on an $n \times n$ matrix holding $n$ subgraphs with $n$ node features. Top row shows off-diagonal updates, bottom row shows diagonal (root node) updates. Each colour represents a different parameter. Full squares represent global sum pooling; triangles represent local pooling. Two triangles represent both local and global pooling.

**Reconstruction GNN / NGNN / DS-GNN.** These methods do not distinguish between root and non-root nodes, effectively sharing the parameters between the two (same yellow colour). The representation of a node in a subgraph is obtained via message passing and aggregation within the subgraph, thus, locally.

**ID-GNN.** ID-GNN performs message passing on each subgraph but distinguishes messages coming from the root (purple instead of yellow), resulting in an additional parameter for non-root updates.

**DSS-GNN.** DSS allows information sharing between subgraphs. Indeed, it does not only perform message passing within each subgraph (yellow), but also on the aggregated adjacency matrix. The message passing on the aggregated adjacency matrix uses the original connectivity and it is therefore still local (triangle, yellow). However, it uses node representations obtained by aggregating node representations globally across subgraphs (orange).

**GNN-AK-ctx.** GNN-AK-ctx distinguishes between root and non-root updates. Non-root node are updated by first copying the diagonal representation of the corresponding node (green) and then performing message passing locally (yellow). Root nodes are updated first by performing a local message passing (yellow), and then by also considering the subgraph readout (yellow squares) and the aggregated representation of the node across subgraphs (orange).

**SUN.** SUN distinguishes root and non-root updates. Each non-root node is updated with: (1) the representation of the node at the previous iteration (black), (2) the representation of the root of the subgraph in which the current node is located (purple), (3) the representation of the node in the subgraph where it is root (green), (4) the readout on the subgraph (yellow squares), (5) message passing on the subgraph (yellow triangles), (6) message passing on the aggregated connectivity (yellow triangles and orange squares). For root nodes many of these terms collapse and the node is updated by only considering (1) (bright green), (4) (blue squares), (5) (blue triangles), (6) (blue triangles and red squares).

**2-IGN and ReIGN(2).** The node representations are updated according to Equation (4). In the 2-IGN case, the operations are all global (squares), whilst for ReIGN(2) each aggregation can also be performed locally (triangles), as prescribed by the connectivity of the subgraph or by the original connectivity.

# D  Proofs for Section 6 – Subgraph GNNs and ReIGN(2)

## D.1  ReIGN(2) expansion of aggregation terms

We report in Table 3 the expansion rules which allow to derive the ReIGN(2) equations from the 2-IGN ones (Equation 4), which we copy here below for convenience.

$$x_i^{i,(t+1)} = v_{\theta_1}\left(x_i^{i,(t)}, \square_j x_j^{j,(t)}, \square_{j \neq i} x_j^{i,(t)}, \square_{h \neq i} x_i^{h,(t)}, \square_{h \neq j} x_j^{h,(t)}\right)$$

$$x_i^{k,(t+1)} = v_{\theta_2}\left(x_i^{k,(t)}, x_k^{i,(t)}, \square_{h \neq j} x_j^{h,(t)}, \square_{h \neq i} x_i^{h,(t)}, \square_{j \neq k} x_j^{k,(t)}, \square_{j \neq i} x_j^{i,(t)}, \square_{h \neq k} x_k^{h,(t)}, x_k^{k,(t)}, x_i^{i,(t)}, \square_j x_j^{j,(t)}\right)$$

In Table 3 we assign each term an identifier (id.) which will allow us to easily refer to specific terms in the proofs we report below. We additionally provide an interpretation for each of the three expanded terms in the last column. Global and local 'vertical' aggregations are dubbed 'needle's in analogy with what the authors in Bevilacqua et al. [7, Definition 5] define as 'needle' colours in their DSS- Weisfeiler-Leman variant.

Table 3: `ReIGN(2)` expansion rules. For each of the `2-IGN` global aggregation terms in Equation 4, `ReIGN(2)` additionally considers two more *local* aggregation terms, which sparsify the aggregation to only include factors adjacent according to the subgraph or original graph connectivities.

| target | id. | 2-IGN term | ReIGN(2) expansion | interpretation |
|---|---|---|---|---|
| $x_i^i$ | #1.on | $\square_j\, x_j^j$ | $\left[\sum_{j/j\sim_i i/j\sim i} x_j^j\right]$ | root-readout / root-msg for $i$ (on $i$) |
| | #2.on | $\square_{j\neq i}\, x_j^i$ | $\left[\sum_{j\neq i/j\sim_i i/j\sim i} x_j^i\right]$ | $i$-readout / $i$-msg for $i$ (on $i$) |
| | #3.on | $\square_{h\neq i}\, x_i^h$ | $\left[\sum_{h\neq i/h\sim_i i/h\sim i} x_i^h\right]$ | $i$-'needle' / local $i$-'needle' for $i$ (on $i$) |
| | #4.on | $\square_{h\neq j}\, x_j^h$ | $\left[\sum_{h/h\sim_i i/h\sim i}\sum_{j/j\sim_h h/j\sim h} x_j^h\right]$ | non-root-readout / joint local needle for $i$ (on $i$) and msg for $h$ (on $h$) |
| $x_i^k$ | #1.off | $\square_{h\neq j}\, x_j^h$ | $\left[\sum_{h/h\sim_k i/h\sim i}\sum_{j/j\sim_h i/j\sim i} x_j^h\right]$ | non-root-readout / joint local needle and msg for $i$ (on $k$ and $h$) |
| | #2.off | $\square_{h\neq i}\, x_i^h$ | $\left[\sum_{h\neq i/h\sim_k i/h\sim i} x_i^h\right]$ | $i$-'needle' / local $i$-'needle' for $i$ (on $k$) |
| | #3.off | $\square_{j\neq k}\, x_j^k$ | $\left[\sum_{j\neq k/j\sim_k i/j\sim i} x_j^k\right]$ | $k$-readout / $k$-msg for $i$ (on $k$) |
| | #4.off | $\square_{h\neq i}\, x_h^i$ | $\left[\sum_{h\neq i/h\sim_k i/h\sim i} x_h^i\right]$ | $i$-readout / $i$-msg for $i$ (on $k$) |
| | #5.off | $\square_{h\neq k}\, x_k^h$ | $\left[\sum_{h\neq k/h\sim_k i/h\sim i} x_k^h\right]$ | $k$-'needle' / local $k$-'needle' for $i$ (on $k$) |
| | #6.off | $\square_j\, x_j^j$ | $\left[\sum_{j/j\sim_k i/j\sim i} x_j^j\right]$ | root-readout / root-msg for $i$ (on $k$) |

Interestingly, expansions in Table 3 could be extended to also include global summations extending only over subgraph nodes as already proposed in Zhang and Li [59], Zhao et al. [61]. For example, term [#3.off] could also include summation $\sum_{j\in V^k} x_j^k$, where $V^k$ is the vertex set of subgraph $k$. As a last note, we remark how all subgraph-local aggregations updating target $x_i^k$ (second expansions) consider the connectivity of subgraph $k$. We believe it could be possible to extend `ReIGN(2)` to also make use of a different subgraph connectivity, e.g. to define a variant of term [#2.off] which includes expansion $\sum_{h\sim_i i} x_i^h$. We defer these enquiries to future works.

## D.2 Proofs for Section 6.1

*Proof of Theorem 8.* Any Subgraph GNN in class $\Upsilon$ has the following form:

$$\mathcal{N} = \big(\mu \circ \rho \circ \mathcal{S} \circ \pi\big)$$

with $\mu$ any MLP, $\rho$ a permutation invariant pooling function, $\pi \in \Pi$ and $\mathcal{S}$ a stacking $\mathcal{S} = L_T \circ \ldots L_1$. Any `ReIGN(2)` model has exactly the same form, with the only (important) difference that layers $L_t$'s in the stacking are `ReIGN(2)` layers. Therefore, the theorem is proved by showing that, for any $\mathcal{N} \in \Upsilon$, the $\mathcal{N}$-layer equations can be implemented by an appropriate `ReIGN(2)` layer stacking. This effort consists in describing (a series of) linear functions $\upsilon_1, \upsilon_2$ as in Equation 4 implementing the layer equation for model $\mathcal{N} \in \Upsilon$. In practice, this will involve specifying which linear transformation $W$ is applied to each of the terms in Equation 4 after expanding each summation according to the rules in Table 3. For convenience, we will directly omit terms assigned a 'nullifying' linear transformation $\mathbf{0}$.

[**DS-GNN**] has its layer equations in the form of Equation 23. These are recovered with one `ReIGN(2)` layer by linearly transforming the second expansion for aggregated terms [#2.on] and [#3.off], and by sharing parameters between off- and on-diagonal updates:

$$x_i^{i,(t+1)} = \sigma\Big(W_{1,t} x_i^{i,(t)} + W_{2,t} \sum_{j\sim_i i} x_j^{i,(t)}\Big)$$

$$x_i^{k,(t+1)} = \sigma\Big(W_{1,t} x_i^{k,(t)} + W_{2,t} \sum_{j\sim_k i} x_j^{k,(t)}\Big)$$

[**DSS-GNN**] has its update rule in the form of Equation 28. It is needed to stack two linear `ReIGN(2)` layers to recover these. The first layer computes message passing on each subgraph as in DS-GNN

and cross-bag aggregation of features. We expand the hidden dimension to $2d$ so that the first half stores the result from the former, the second that of the latter. We 'utilise' terms [#2.on] and [#3.off] in their second expansion for the message-passing operation and terms [#3.on] and [#2.off] in their global version for cross-bag aggregation:

$$x_i^{i,(t,1)} = [W_{1,t}^1 \; I_d] \, x_i^{i,(t)} + \boldsymbol{\kappa}_{:d}^{:} \, W_{2,t}^1 \sum_{j\sim_i i} x_j^{i,(t)} + \boldsymbol{\kappa}_{d+1:2d}^{:} \sum_{h\neq i} x_i^{h,(t)}$$

$$x_i^{k,(t,1)} = \boldsymbol{\kappa}_{:d}^{:} \, W_{1,t}^1 x_i^{k,(t)} + \boldsymbol{\kappa}_{:d}^{:} \, W_{2,t}^1 \sum_{j\sim_k i} x_j^{k,(t)} + \boldsymbol{\kappa}_{d+1:2d}^{:} \sum_{h\neq i} x_i^{h,(t)} + \boldsymbol{\kappa}_{d+1:2d}^{:} \, x_i^{i,(t)}$$

The second layer reduces the dimensionality back to $d$ and completes the implementation by performing message-passing of the cross-bag-aggregated representations over the original graph. This is realised by employing terms [#2.on] and [#3.off] in their third expansion.

$$x_i^{i,(t+1)} = \sigma\Big( [I_d \,||\, W_{1,t}^2] \, x_i^{i,(t,1)} + W_{2,t}^2 \boldsymbol{\kappa}_{:d}^{d+1:2d} \sum_{j\sim i} x_j^{i,(t,1)} \Big)$$

$$x_i^{k,(t+1)} = \sigma\Big( [I_d \,||\, W_{1,t}^2] \, x_i^{k,(t,1)} + W_{2,t}^2 \boldsymbol{\kappa}_{:d}^{d+1:2d} \sum_{j\sim i} x_j^{k,(t,1)} \Big)$$

[**GNN-AK-ctx & GNN-AK**] Equations 29 and 30 describe the update equations for these models. First, block [S] performs independent message-passing on each subgraph for $L$ steps. The $l$-th update, $l = 1, \ldots, (L-1)$ is implemented as for DS-GNN, that is by employing terms [#2.on] and [#3.off] as follows:

$$x_i^{i,(t,l+1)} = \sigma\Big( W_{1,t,l} x_i^{i,(t,l)} + W_{2,t,l} \sum_{j\sim_i i} x_j^{i,(t,l)} \Big)$$

$$x_i^{k,(t,l+1)} = \sigma\Big( W_{1,t,l} x_i^{k,(t,l)} + W_{2,t,l} \sum_{j\sim_k i} x_j^{k,(t,l)} \Big)$$

Block [A] in GNN-AK-ctx is implemented by one `ReIGN(2)` layer via aggregated terms [#2.on], [#3.on], [#2.off], [#4.off] all in their global version:

$$x_i^{i,(t+1)} = 3 \cdot I_d \, x_i^{i,(t,L)} + \sum_{h\neq i} x_i^{h,(t,L)} + \sum_{j\neq i} x_j^{i,(t,L)}$$

$$x_i^{k,(t+1)} = 3 \cdot I_d \, x_i^{i,(t,L)} + \sum_{h\neq i} x_i^{h,(t,L)} + \sum_{j\neq i} x_j^{i,(t,L)}$$

In the case of GNN-AK, Block [A] is implemented more simply as

$$x_i^{i,(t+1)} = 2 \cdot I_d \, x_i^{i,(t,L)} + \sum_{j\neq i} x_j^{i,(t,L)}$$

$$x_i^{k,(t+1)} = 2 \cdot I_d \, x_i^{i,(t,L)} + \sum_{j\neq i} x_j^{i,(t,L)}$$

where terms [#3.on], [#2.off] are nullified.

[**ID-GNN**] Implements the update rule in Equation 33, which we report here for convenience:

$$x_i^{k,(t+1)} = \sigma\big( W_{1,t} x_i^{k,(t)} + W_{2,t} \sum_{j\sim_k i, j\neq k} x_j^{k,(t)} + \mathbb{1}_{[k\sim_k i]} \cdot W_{3,t} x_k^{k,(t)} \big)$$

We observe that we can rewrite this equation as follows:

$$x_i^{k,(t+1)} = \sigma\big( W_{1,t} x_i^{k,(t)} + \sum_{j\sim_k i} W_{(\mathbb{1}_{[j=k]}+2),t} x_j^{k,(t)} \big) \tag{36}$$

where $(\mathbb{1}_{[j=k]} + 2) = 3$ if $j = k$, 2 otherwise. That is: (i) we do not explicitly exclude the root node from the set of $i$'s neighbours; (ii) messages are computed via $W_{3,t}$ if from root nodes, via $W_{2,t}$ otherwise. This update equation can be implemented by two `ReIGN(2)` layers. Similarly as in

the DSS-GNN derivation, the first layer expands the hidden dimension to $2d$. The first $d$ channels will store node representations transformed according to $W_{1,t}$, whilst the remaining $d+1$ to $2d$ channels will store node representations transformed according to $W_{3,t}, W_{2,t}$ for, respectively, on- and off-diagonal terms, namely root and non-root nodes:

$$x_i^{i,(t,1)} = [W_{1,t} \ W_{3,t}] \, x_i^{i,(t)}$$
$$x_i^{k,(t,1)} = [W_{1,t} \ W_{2,t}] \, x_i^{k,(t)}$$

The second `ReIGN(2)` layer effectively completes implementing message-passing and brings back the dimension to $d$. It combines and aggregates the resulting transformations via terms [#2.on] and [#3.off] in their second expansion as follows:

$$x_i^{i,(t+1)} = \sigma\Big( \boldsymbol{\kappa}_{:d}^{:d} \, x_i^{i,(t,1)} + \boldsymbol{\kappa}_{:d}^{d+1:2d} \sum_{j \sim_i i} x_j^{i,(t,1)} \Big)$$
$$x_i^{k,(t+1)} = \sigma\Big( \boldsymbol{\kappa}_{:d}^{:d} \, x_i^{k,(t,1)} + \boldsymbol{\kappa}_{:d}^{d+1:2d} \sum_{j \sim_k i} x_j^{k,(t,1)} \Big)$$

[**NGNN**] Computes independent subgraph-wise message passing as in DS-GNN: the same `ReIGN(2)` layer implements this one. $\qquad\square$

*Proof of Proposition 9.* `ReIGN(2)` model $\mathcal{R}_{\rho,\Theta,\pi}$ distinguishes $G_1, G_2$ if they are assigned distinct representations, that is: $y_{G_1} = \mathcal{R}_{\rho,\Theta,\pi}(A_1, X_1) \neq \mathcal{R}_{\rho,\Theta,\pi}(A_2, X_2) = y_{G_2}$. A `3-IGN` instance $\mathcal{M}_\Omega$ implementing $\mathcal{R}_{\rho,\Theta,\pi}$ on the same pair of graphs would distinguish the same graphs as well. We will show the existence of such a model instance.

We seek to find a `3-IGN` model $\mathcal{M}_\Omega$ in the form of Equation 16 such that: $\mathcal{M}_\Omega(A_1, X_1) = \mathcal{R}_{\rho,\Theta,\pi}(A_1, X_1) = y_{G_1}$ and $\mathcal{M}_\Omega(A_2, X_2) = \mathcal{R}_{\rho,\Theta,\pi}(A_2, X_2) = y_{G_2}$, where $\mathcal{R}_{\rho,\Theta,\pi}(\,\cdot\,) = (\mu \circ \rho \circ \mathcal{S} \circ \pi)_\Theta(\cdot)$. From the hypotheses of the theorem, block $\rho$ is `3-IGN`-computable, so there must exist a `3-IGN` layer stacking $\mathcal{M}_\rho$ such that $\mathcal{M}_\rho \cong \rho$. At the same time, $\pi \in \Pi$ so Lemma 4 ensures the existence of a `3-IGN` stacking $\mathcal{M}_\pi$ such that $\mathcal{M}_\pi \cong \pi$. It is left to show the existence of a `3-IGN` stacking $\mathcal{M}_\mathcal{S}$ implementing the `ReIGN(2)` stacking $\mathcal{S}$ on the same pair of graphs. Without loss of generality, it is sufficient to show the existence of a `3-IGN` stacking $\mathcal{M}_L$ implementing one single `ReIGN(2)` intermediate layer $L$ defined as per Equations 4 and the aggregated term expansions in Table 3. We will then show how to construct $\mathcal{M}_L$ for a generic step $t$ in the following. In order to more explicitly reflect the index notation used in Table 3 ($i$ refers to nodes, $k$ to subgraphs) we will refer to the `3-IGN` orbit representations with different subscripts: $\mathcal{Y} \cong X_{iii} \sqcup X_{iij} \sqcup X_{kik} \sqcup X_{kii} \sqcup X_{kij}$, that is we rename $X_{iji}$ as $X_{kik}$, $X_{ijj}$ as $X_{kii}$, $X_{ijk}$ as $X_{kij}$.

We note that the summation of all non-aggregated and globally aggregated terms is recovered by just one `3-IGN` layer, including their linear transformations. In the yet-to-construct stacking, the first layer performs this computation and stores the result into $d$-auxiliary channels. The layer also replicates the current representations $x^{k,(t),i}, x^{i,(t),i}$ in the first $d$ channels. The implementation of local aggregations, i.e. second and third expansions in Table 3, will require a larger number of layers: this aforementioned input replication allows them to operate on the original representations even after having implemented non-aggregated globally aggregated terms, as required. The result of their computation will then be used to update the intermediate term in channels $d+1$ to $2d$, as we shall see next. Let us start by describing the first `3-IGN` layer, constructed as follows:

$$X_{iii}^{(t,1)} = [I_d \ \theta_{1,t}^{(i,i)}] \, X_{iii}^{(t)} + \sum_{o_1 \in O_1} \boldsymbol{\kappa}_{d+1:2d}^{:} \, o_1$$
$$X_{kii}^{(t,1)} = [I_d \ \theta_{2,t}^{(k,i)}] \, X_{kii}^{(t)} + \sum_{o_2 \in O_2} \boldsymbol{\kappa}_{d+1:2d}^{:} \, o_2$$

with set $O_1$ being:

$$O_1 = \Big\{ \theta_{1,t}^{(j,j)} \boldsymbol{\beta}_{*i,*i,*i} \, \boldsymbol{\pi} \, X_{iii}^{(t)}, \ \theta_{1,t}^{(i,j)} \boldsymbol{\beta}_{i,i,i} \, \boldsymbol{\pi}_i \, X_{ijj}^{(t)}, \ \theta_{1,t}^{(h,i)} \boldsymbol{\beta}_{j,j,j} \, \boldsymbol{\pi}_j \, X_{ijj}^{(t)}, \ \theta_{1,t}^{(h,j)} \boldsymbol{\beta}_{*i,*i,*i} \, \boldsymbol{\pi} \, X_{ijj}^{(t)} \Big\}$$

and set $O_2$ being:

$$O_2 = \Big\{ \theta_{2,t}^{(i,k)} \boldsymbol{\beta}_{i,k,k} \ X_{kii}^{(t)}, \theta_{2,t}^{(h,j)} \boldsymbol{\beta}_{*k,*i,*i} \ \boldsymbol{\pi} \ X_{kii}^{(t)}, \theta_{2,t}^{(h,i)} \boldsymbol{\beta}_{*,i,i} \ \boldsymbol{\pi}_i \ X_{kii}^{(t)}, \theta_{2,t}^{(k,j)} \boldsymbol{\beta}_{k,*i,*i} \ \boldsymbol{\pi}_k \ X_{kii}^{(t)}$$

$$\theta_{2,t}^{(i,j)} \boldsymbol{\beta}_{*,k,k} \ \boldsymbol{\pi}_k \ X_{kii}^{(t)}, \theta_{2,t}^{(h,k)} \boldsymbol{\beta}_{i,*k,*k} \ \boldsymbol{\pi}_i \ X_{kii}^{(t)}, \theta_{2,t}^{(k,k)} \boldsymbol{\beta}_{i,*k,*k} \ X_{iii}^{(t)}, \theta_{2,t}^{(i,i)} \boldsymbol{\beta}_{*,i,i} \ X_{iii}^{(t)},$$

$$\theta_{2,t}^{(j,j)} \boldsymbol{\beta}_{*k,*i,*i} \ \boldsymbol{\pi} \ X_{iii}^{(t)} \Big\}$$

We now focus on the following layers. The $l$-th local aggregation in Table 3 (second and third expansions) can be obtained by a 3-IGN layer stacking implementing the steps of: (1) *Message broadcasting*, (2) *Message sparsification*, (3) *Message aggregation*, (4) *Update*, similarly to what already shown for the Proof of Lemma 5. More in detail, the underlying construction will be such such that: first, messages are placed on the third axis of the cubed tensor on which the 3-IGN operates (broadcasting, 1); they are then sparsified consistently with the (sub)graph connectivity (sparsification, 2); aggregated via pooling operations on the same axis (aggregation, 3); finally, linearly transformed with their specific linear operator and used to update the intermediate representation(s) in channels $d+1$ to $2d$ (update, 4). We note that for each local aggregation term, these steps essentially differ in the way messages are propagated (1) and in the specific linear transformations applied (4), while the same computation is shared for the sparsification (2) and aggregation (3) steps. Thus, we deem it convenient to first describe these steps and then show how specific choices for (1), (4) recover each desired term.

Here, we assume that $\mathcal{M}_\pi$ writes in $X_{kij}, i \neq j$, the connectivity between nodes $i, j$ in subgraph $k$ (first channel) as well as that prescribed by the original graph connectivity (second channel) — once more, see discussion [***Retaining original connectivity***] in the Proof of Lemma 4. In order to implement this step on the input graph pair, it is sufficient to have 3-IGN layers applying an MLP which sparsifies messages according to the aforementioned connectivities and then to aggregate the sparsified messages by global summation, effectively realising *both* the second and third expansion for terms in Table 3. Step (2) is realised as:

$$X_{iij}^{(t,l,\mathrm{sp.})} = \big[ \boldsymbol{\kappa}_{:d}^{:d} \ \boldsymbol{\varphi}_{d+1:2d}^{(\odot_{iij}^{\widetilde{i}}):} \ \boldsymbol{\varphi}_{2d+1:}^{(\odot_{iij}^{\widetilde{i}}):} \big] X_{iij}^{(t,l,\mathrm{broad.})} \tag{37}$$

$$X_{kik}^{(t,l,\mathrm{sp.})} = \big[ \boldsymbol{\kappa}_{:d}^{:d} \ \boldsymbol{\varphi}_{d+1:2d}^{(\odot_{kik}^{\widetilde{k}}):} \ \boldsymbol{\varphi}_{2d+1:}^{(\odot_{kik}^{\widetilde{k}}):} \big] X_{kik}^{(t,l,\mathrm{broad.})} \tag{38}$$

$$X_{kij}^{(t,l,\mathrm{sp.})} = \big[ \boldsymbol{\kappa}_{:d}^{:d} \ \boldsymbol{\varphi}_{d+1:2d}^{(\odot_{kij}^{\widetilde{k}}):} \ \boldsymbol{\varphi}_{2d+1:}^{(\odot_{kij}^{\widetilde{}}):} \big] X_{kij}^{(t,l,\mathrm{broad.})} \tag{39}$$

where $X_{iij}^{(t,l,\mathrm{broad.})}, X_{kik}^{(t,l,\mathrm{broad.})}, X_{kij}^{(t,l,\mathrm{broad.})}$ are computed by the yet-to-describe step (1) and MLPs $\boldsymbol{\varphi}^{(\odot_{iij}^{\widetilde{i}})}, \boldsymbol{\varphi}^{(\odot_{iij}^{\widetilde{}})}, \boldsymbol{\varphi}^{(\odot_{kik}^{\widetilde{k}})}, \boldsymbol{\varphi}^{(\odot_{kik}^{\widetilde{}})}, \boldsymbol{\varphi}^{(\odot_{kij}^{\widetilde{k}})}, \boldsymbol{\varphi}^{(\odot_{kij}^{\widetilde{}})}$ compute the required sparsifications. We do not describe how to construct such MLPs, but their existence is guaranteed by Proposition 14, which we can invoke by constructing the same datasets as shown in the Proof of Lemma 5 for the DSS-GNN derivation. Step (3) aggregates these sparsified messages; concurrently the same layer linearly transforms the result and adds it to the current, intermediate node representations, performing step (4):

$$X_{iii}^{(t,l+1)} = \boldsymbol{\kappa}_{:2d}^{:2d} X_{iii}^{(t,l,\mathrm{sp.})} + \boldsymbol{\kappa}_{d+1:2d}^{:} \big[ \mathbf{0} \ || \ \theta_{1,l,t}^{\widetilde{i}} \ || \ \theta_{1,l,t}^{\widetilde{}} \big] \boldsymbol{\beta}_{iii} \ \boldsymbol{\pi}_i \ X_{iij}^{(t,l,\mathrm{sp.})}$$

$$X_{kii}^{(t,l+1)} = \boldsymbol{\kappa}_{:2d}^{:2d} X_{kii}^{(t,l,\mathrm{sp.})} + \boldsymbol{\kappa}_{d+1:2d}^{:} \big[ \mathbf{0} \ || \ \theta_{2,l,t}^{\widetilde{k}} \ || \ \theta_{2,l,t}^{\widetilde{}} \big] \boldsymbol{\beta}_{kii} \ X_{kik}^{(t,l,\mathrm{sp.})} +$$

$$+ \boldsymbol{\kappa}_{d+1:2d}^{:} \big[ \mathbf{0} \ || \ \theta_{2,l,t}^{\widetilde{k}} \ || \ \theta_{2,l,t}^{\widetilde{}} \big] \boldsymbol{\beta}_{kii} \ \boldsymbol{\pi}_{ki} \ X_{kij}^{(t,l,\mathrm{sp.})}$$

Here, parameters $\theta_{1,l,t}^{\widetilde{i}}, \theta_{1,l,t}^{\widetilde{}}, \theta_{2,l,t}^{\widetilde{k}}, \theta_{2,l,t}^{\widetilde{}}$ will depend on the specific term being implemented.

As for step (1), one 3-IGN layer suffices to properly broadcast the current input representations $X_{iii}^{(t,l)}, X_{kii}^{(t,l)}$ over, respectively, $X_{iji}^{(t,l,\mathrm{broad.})}$ and $X_{kik}^{(t,l,\mathrm{broad.})}, X_{kij}^{(t,l,\mathrm{broad.})}$. We will now show the required broadcasting operations in a way that, then, the following layers described above will

effectively implement the second and third expansions of the terms in Table 3:

$$[\#1.\text{on}] \quad X_{iij}^{(t,l,\text{broad.})} = \boldsymbol{\kappa}_{:d}^{:d} X_{iij}^{(t,l)} + \boldsymbol{\kappa}_{d+1:3d}^{:2d} [\boldsymbol{\kappa}_{:d}^{:d} \ \boldsymbol{\kappa}_{:d}^{:d}] \boldsymbol{\beta}_{*,*,i} X_{iii}^{(t,l)}$$

$$[\#2.\text{on}] \quad X_{iij}^{(t,l,\text{broad.})} = \boldsymbol{\kappa}_{:d}^{:d} X_{iij}^{(t,l)} + \boldsymbol{\kappa}_{d+1:3d}^{:2d} [\boldsymbol{\kappa}_{:d}^{:d} \ \boldsymbol{\kappa}_{:d}^{:d}] \boldsymbol{\beta}_{k,k,i} X_{kii}^{(t,l)}$$

$$[\#3.\text{on}] \quad X_{iij}^{(t,l,\text{broad.})} = \boldsymbol{\kappa}_{:d}^{:d} X_{iij}^{(t,l)} + \boldsymbol{\kappa}_{d+1:3d}^{:2d} [\boldsymbol{\kappa}_{:d}^{:d} \ \boldsymbol{\kappa}_{:d}^{:d}] \boldsymbol{\beta}_{i,i,k} X_{kii}^{(t,l)}$$

$$[\#2.\text{off}] \quad X_{kik}^{(t,l,\text{broad.})} = \boldsymbol{\kappa}_{:d}^{:d} X_{kik}^{(t,l)} + \boldsymbol{\kappa}_{d+1:3d}^{:2d} [\boldsymbol{\kappa}_{:d}^{:d} \ \boldsymbol{\kappa}_{:d}^{:d}] \boldsymbol{\beta}_{*,i,*} X_{iii}^{(t,l)}$$

$$X_{kij}^{(t,l,\text{broad.})} = \boldsymbol{\kappa}_{:d}^{:d} X_{kij}^{(t,l)} + \boldsymbol{\kappa}_{d+1:3d}^{:2d} [\boldsymbol{\kappa}_{:d}^{:d} \ \boldsymbol{\kappa}_{:d}^{:d}] \boldsymbol{\beta}_{*,i,k} X_{kii}^{(t,l)}$$

$$[\#3.\text{off}] \quad X_{kik}^{(t,l,\text{broad.})} = \boldsymbol{\kappa}_{:d}^{:d} X_{kik}^{(t,l)} + \boldsymbol{\kappa}_{d+1:3d}^{:2d} [\boldsymbol{\kappa}_{:d}^{:d} \ \boldsymbol{\kappa}_{:d}^{:d}] \boldsymbol{\beta}_{i,*,i} X_{iii}^{(t,l)}$$

$$X_{kij}^{(t,l,\text{broad.})} = \boldsymbol{\kappa}_{:d}^{:d} X_{kij}^{(t,l)} + \boldsymbol{\kappa}_{d+1:3d}^{:2d} [\boldsymbol{\kappa}_{:d}^{:d} \ \boldsymbol{\kappa}_{:d}^{:d}] \boldsymbol{\beta}_{k,*,i} X_{kii}^{(t,l)}$$

$$[\#4.\text{off}] \quad X_{kik}^{(t,l,\text{broad.})} = \boldsymbol{\kappa}_{:d}^{:d} X_{kik}^{(t,l)} + \boldsymbol{\kappa}_{d+1:3d}^{:2d} [\boldsymbol{\kappa}_{:d}^{:d} \ \boldsymbol{\kappa}_{:d}^{:d}] \boldsymbol{\beta}_{*,i,*} X_{iii}^{(t,l)}$$

$$X_{kij}^{(t,l,\text{broad.})} = \boldsymbol{\kappa}_{:d}^{:d} X_{kij}^{(t,l)} + \boldsymbol{\kappa}_{d+1:3d}^{:2d} [\boldsymbol{\kappa}_{:d}^{:d} \ \boldsymbol{\kappa}_{:d}^{:d}] \boldsymbol{\beta}_{*,k,i} X_{kii}^{(t,l)}$$

$$[\#5.\text{off}] \quad X_{kik}^{(t,l,\text{broad.})} = \boldsymbol{\kappa}_{:d}^{:d} X_{kik}^{(t,l)} + \boldsymbol{\kappa}_{d+1:3d}^{:2d} [\boldsymbol{\kappa}_{:d}^{:d} \ \boldsymbol{\kappa}_{:d}^{:d}] \boldsymbol{\beta}_{i,*,i} X_{iii}^{(t,l)}$$

$$X_{kij}^{(t,l,\text{broad.})} = \boldsymbol{\kappa}_{:d}^{:d} X_{kij}^{(t,l)} + \boldsymbol{\kappa}_{d+1:3d}^{:2d} [\boldsymbol{\kappa}_{:d}^{:d} \ \boldsymbol{\kappa}_{:d}^{:d}] \boldsymbol{\beta}_{i,*,k} X_{kii}^{(t,l)}$$

$$[\#6.\text{off}] \quad X_{kik}^{(t,l,\text{broad.})} = \boldsymbol{\kappa}_{:d}^{:d} X_{kik}^{(t,l)} + \boldsymbol{\kappa}_{d+1:3d}^{:2d} [\boldsymbol{\kappa}_{:d}^{:d} \ \boldsymbol{\kappa}_{:d}^{:d}] \boldsymbol{\beta}_{i,*,i} X_{iii}^{(t,l)}$$

$$X_{kij}^{(t,l,\text{broad.})} = \boldsymbol{\kappa}_{:d}^{:d} X_{kij}^{(t,l)} + \boldsymbol{\kappa}_{d+1:3d}^{:2d} [\boldsymbol{\kappa}_{:d}^{:d} \ \boldsymbol{\kappa}_{:d}^{:d}] \boldsymbol{\beta}_{*,*,i} X_{iii}^{(t,l)}$$

Terms [#4.on], [#1.off] involve two sparse summations and thus require a slightly different construction. In particular, they are concurrently implemented by computing steps (1,2,3) twice, in sequence, followed by computation of step (4). The first brodcasting step (1) is realised as follows:

$$X_{iij}^{(t,l,1^{st}\text{broad.})} = \boldsymbol{\kappa}_{:d}^{:d} X_{iij}^{(t,l)} + \boldsymbol{\kappa}_{d+1:3d}^{:2d} [\boldsymbol{\kappa}_{:d}^{:d} \ \boldsymbol{\kappa}_{:d}^{:d}] \boldsymbol{\beta}_{k,k,i} X_{kii}^{(t,l)}$$

$$X_{kik}^{(t,l,1^{st}\text{broad.})} = \boldsymbol{\kappa}_{:d}^{:d} X_{kik}^{(t,l)} + \boldsymbol{\kappa}_{d+1:3d}^{:2d} [\boldsymbol{\kappa}_{:d}^{:d} \ \boldsymbol{\kappa}_{:d}^{:d}] \boldsymbol{\beta}_{i,*,i} X_{iii}^{(t,l)}$$

$$X_{kij}^{(t,l,1^{st}\text{broad.})} = \boldsymbol{\kappa}_{:d}^{:d} X_{kij}^{(t,l)} + \boldsymbol{\kappa}_{d+1:3d}^{:2d} [\boldsymbol{\kappa}_{:d}^{:d} \ \boldsymbol{\kappa}_{:d}^{:d}] \boldsymbol{\beta}_{k,*,i} X_{kii}^{(t,l)}$$

that is, in the same way as in the implementation of terms [#2.on], [#3.off]. Then, the first sparsification step (2) is computed as per Equations (37) to (39), generating $X_{iij}^{(t,l,1^{st}\text{sp.})}, X_{kik}^{(t,l,1^{st}\text{sp.})}, X_{kij}^{(t,l,1^{st}\text{sp.})}$. The first aggregation step (3) is performed jointly with the second broadcasting step (1) as:

$$X_{iij}^{(t,l,2^{nd}\text{broad.})} = \boldsymbol{\kappa}_{:d}^{:d} X_{iij}^{(t,l,1^{st}\text{sp.})} + \boldsymbol{\kappa}_{d+1:3d}^{d+1:3d} \boldsymbol{\beta}_{*,*,i} \boldsymbol{\pi}_i X_{iij}^{(t,l,1^{st}\text{sp.})}$$

$$X_{kik}^{(t,l,2^{nd}\text{broad.})} = \boldsymbol{\kappa}_{:d}^{:d} X_{kik}^{(t,l,1^{st}\text{sp.})} + \boldsymbol{\kappa}_{d+1:3d}^{d+1:3d} \boldsymbol{\beta}_{k,i,k} \boldsymbol{\pi}_{k,i} X_{kij}^{(t,l,1^{st}\text{sp.})} +$$

$$+ \boldsymbol{\kappa}_{d+1:3d}^{:2d} [\boldsymbol{\kappa}_{:d}^{:d} \ \boldsymbol{\kappa}_{:d}^{:d}] \boldsymbol{\beta}_{i,*,i} X_{iii}^{(t,l,1^{st}\text{sp.})}$$

$$X_{kij}^{(t,l,2^{nd}\text{broad.})} = \boldsymbol{\kappa}_{:d}^{:d} X_{kij}^{(t,l,1^{st}\text{sp.})} + \boldsymbol{\kappa}_{d+1:3d}^{d+1:3d} \boldsymbol{\beta}_{*,i,k} X_{kik}^{(t,l,1^{st}\text{sp.})} + \boldsymbol{\kappa}_{d+1:3d}^{d+1:3d} \boldsymbol{\beta}_{*,i,k} \boldsymbol{\pi}_{k,i} X_{kij}^{(t,l,1^{st}\text{sp.})}$$

where, crucially, the results from pooling are broadcast back into orbit tensors $X_{iij}, X_{kik}, X_{kij}$, given that one more local summation is required. Next, one more sparsification takes place in the form of Equations (37) to (39), generating $X_{iij}^{(t,l,2^{nd}\text{sp.})}, X_{kik}^{(t,l,2^{nd}\text{sp.})}, X_{kij}^{(t,l,2^{nd}\text{sp.})}$. Finally, second aggregation step (3) is performed jointly with the final update step (4), which writes back into orbit tensors $X_{iii}, X_{kii}$:

$$X_{iii}^{(t,l+1)} = \boldsymbol{\kappa}_{:2d}^{:2d} X_{iii}^{(t,l,2^{nd}\text{sp.})} + \boldsymbol{\kappa}_{d+1:2d}^{:} [\mathbf{0} \ || \ \theta_{1,l,t}^{\sim i} \ || \ \theta_{1,l,t}^{\sim}] \boldsymbol{\beta}_{i,i,i} \boldsymbol{\pi}_i X_{iij}^{(t,l,2^{nd}\text{sp.})}$$

$$X_{kii}^{(t,l+1)} = \boldsymbol{\kappa}_{:2d}^{:2d} X_{kii}^{(t,l,2^{nd}\text{sp.})} + \boldsymbol{\kappa}_{d+1:2d}^{:} [\mathbf{0} \ || \ \theta_{2,l,t}^{\sim k} \ || \ \theta_{2,l,t}^{\sim}] \boldsymbol{\beta}_{kii} X_{kik}^{(t,l,2^{nd}\text{sp.})} +$$

$$+ \boldsymbol{\kappa}_{d+1:2d}^{:} [\mathbf{0} \ || \ \theta_{2,l,t}^{\sim k} \ || \ \theta_{2,l,t}^{\sim}] \boldsymbol{\beta}_{kii} \boldsymbol{\pi}_{k,i} X_{kij}^{(t,l,2^{nd}\text{sp.})}$$

When all terms are implemented and combined together, it is only left to bring back the dimensionality to $d$, overwriting the previous node representations with the newly computed one:

$$X_{iii}^{(t+1)} = \sigma\left(\boldsymbol{\kappa}_{:d}^{d+1:} X_{iii}^{(t,L)}\right)$$

$$X_{kii}^{(t+1)} = \sigma\left(\boldsymbol{\kappa}_{:d}^{d+1:} X_{kii}^{(t,L)}\right)$$

$\square$

*Proof of Corollary 10.* We proceed by contradiction as in the Proof of Corollary 7. Suppose there exist non-isomorphic but 3-WL-equivalent graphs $G_1, G_2$ distinguished by instance $\mathcal{R}_{\rho,\Theta,\bar{\pi}}$. That is, $\mathcal{R}_{\rho,\Theta,\bar{\pi}}(G_1) \neq \mathcal{R}_{\rho,\Theta,\bar{\pi}}(G_2)$. In view of Theorem 9, there must exists a 3-IGN instance $\mathcal{M}_\Omega$ such that $\mathcal{M}_\Omega(G_1) \neq \mathcal{M}_\Omega(G_2)$. By Theorem 15, if there exists a 3-IGN distinguishing $G_1, G_2$, then these two graphs must be distinguished by the 3-WL algorithm, against our hypothesis. $\square$

### D.3 Proofs for Section 6.2

SUN **in linear form**

$$x_i^{i,(t+1)} = \sigma\Big(U_{r,t}^2 \cdot x_i^{i,(t)} + U_{r,t}^3 \cdot \sum_j x_j^{i,(t)} +$$

$$+ U_{r,t}^4 \cdot \sum_{j\sim_i i} x_j^{i,(t)} + U_{r,t}^5 \cdot \sum_h x_i^{h,(t)} + U_{r,t}^6 \cdot \sum_{j\sim i}\sum_h x_j^{h,(t)}\Big) \quad (40)$$

$$x_i^{k,(t+1)} = \sigma\Big(U_t^0 \cdot x_i^{i,(t)} + U_t^1 \cdot x_k^{k,(t)} + U_t^2 \cdot x_i^{k,(t)} + U_t^3 \cdot \sum_j x_j^{k,(t)} +$$

$$+ U_t^4 \cdot \sum_{j\sim_k i} x_j^{k,(t)} + U_t^5 \cdot \sum_h x_i^{h,(t)} + U_t^6 \cdot \sum_{j\sim i}\sum_h x_j^{h,(t)}\Big) \quad (41)$$

*Proof of Proposition 11.* We construct a stacking of 2 ReIGN(2) layers implementing one SUN layer as per Equations 40 and 41. The first layer expands the dimension of the hidden representations to $2d$. The first $d$ channels store the sum of linear transformations operated by $U_{r,t}^2, U_{r,t}^3, U_{r,t}^4$ in Equation 40 and those operated by $U_t^0, U_t^1, U_t^2, U_t^3, U_t^4$ in Equation 41. Channels $d + 1$ to $2d$ will store terms $\sum_h x_i^{h,(t)}$:

$$x_i^{i,(t,1)} = [(U_{r,t}^2 + I_d)\ \ I_d]\, x_i^{i,(t)} + \boldsymbol{\kappa}_{:d/(2d)}^{:}\, U_{r,t}^3 \cdot \sum_{j\neq i} x_j^{i,(t)} +$$

$$+ \boldsymbol{\kappa}_{:d/(2d)}^{:}\, U_{r,t}^4 \cdot \sum_{j\sim_i i} x_j^{i,(t)} + \boldsymbol{\kappa}_{d+1:2d}^{:} \sum_{h\neq i} x_i^{h,(t)}$$

$$x_i^{k,(t,1)} = [U_t^0\ \ I_d]\, x_i^{i,(t)} + \boldsymbol{\kappa}_{:d/(2d)}^{:}\, (U_t^1 + I_d) \cdot x_k^{k,(t)} + \boldsymbol{\kappa}_{:d/(2d)}^{:}\, U_t^2 \cdot x_i^{k,(t)} +$$

$$+ \boldsymbol{\kappa}_{:d/(2d)}^{:}\, U_t^3 \cdot \sum_{j\neq k} x_j^{k,(t)} + \boldsymbol{\kappa}_{:d/(2d)}^{:}\, U_t^4 \cdot \sum_{j\sim_k i} x_j^{k,(t)} + \boldsymbol{\kappa}_{d+1:2d}^{:} \sum_{h\neq i} x_i^{h,(t)}$$

where we have used the following aggregated terms. First equation: [#2.on] in its global version and second expansion, [#3.on] in its global version. Second equation: [#2.off] in its global version, [#3.off] in its global version and second expansion. Non-appearing ReIGN(2) terms are nullified. The second ReIGN(2) layer completes the computation by implementing linear transformations $U_{t,r}^5, U_{t,r}^6, U_t^5, U_t^6$, and by contracting the dimensionality back to $d$:

$$x_i^{i,(t+1)} = \sigma\Big([I_d \,\|\, U_{r,t}^5]\, x_i^{i,(t,1)} + U_{r,t}^6 \cdot \boldsymbol{\kappa}_{:d}^{d+1:2d} \sum_{j\sim i} x_j^{i,(t,1)}\Big)$$

$$x_i^{k,(t+1)} = \sigma\Big([I_d \,\|\, U_t^5]\, x_i^{k,(t,1)} + U_t^6 \cdot \boldsymbol{\kappa}_{:d}^{d+1:2d} \sum_{j\sim i} x_j^{k,(t,1)}\Big)$$

where we have used aggregated terms [#2.on] and [#3.off] in their third expansion. $\square$

*Proof of Proposition 12.* We describe how a stacking of `SUN` layers in the form of Equations 40 and 41 implements layers of models in $\Upsilon$ with Morris et al. [36] base-encoders. As usual, we proceed model by model.

[**DS-GNN**] updates root and non-root nodes in the same manner. Thus, we seek to find a choice of linear operators in Equations 40 and 41 in a way that the two coincide and exactly correspond to Equation 23. To this aim, it is sufficient to set:

- $U_{r,t}^2 = U_t^2 = W_{1,t}$

- $U_{r,t}^4 = U_t^4 = W_{2,t}$

and all other weight matrices $U$ to $\mathbf{0}$.

[**DSS-GNN**] implements Equation 28. We proceed similarly as above, setting:

- $U_{r,t}^2 = U_t^2 = W_{1,t}^1$

- $U_{r,t}^4 = U_t^4 = W_{2,t}^1$

- $U_{r,t}^5 = U_t^5 = W_{1,t}^2$

- $U_{r,t}^6 = U_t^6 = W_{2,t}^2$

and all other weight matrices $U$ to $\mathbf{0}$.

[**GNN-AK-ctx & GNN-AK**] We seek to recover Equations 29 ([S]) and 30 ([A]). Each message passing layer in [S] is implemented by one `SUN` layer similarly as above, that is by setting:

- $U_{r,t,l}^2 = U_{t,l}^2 = W_{1,t,l}$

- $U_{r,t,l}^4 = U_{t,l}^4 = W_{2,t,l}$

and all other weight matrices $U$ to $\mathbf{0}$. Then, block [A] is implemented by one `SUN` layer by setting:

- $U_{r,t}^2 = U_{r,t}^3 = U_{r,t}^5 = I$

- $U_t^0 = U_t^3 = U_t^5 = I$

and all other weight matrices $U$ to $\mathbf{0}$. In the case of GNN-AK, we instead require $U_{r,t}^5 = U_t^5 = \mathbf{0}$. No activation $\sigma$ is applied after this layer.

[**ID-GNN**] Two `SUN` layers can implement one ID-GNN layer as in Equation 36, similarly as shown for `ReIGN(2)` in the Proof of Theorem 8. The first layer doubles the representation dimension and applies projections $W_{1,t}, W_{2,t}, W_{3,t}$, by setting:

- $U_{r,t,1}^2 = [W_{1,t} \ \ W_{3,t}]$

- $U_{t,1}^2 = [W_{1,t} \ \ W_{2,t}]$

and all other weight matrices $U$ to $\mathbf{0}$. Here, as usual, $[\,\cdot\,\cdot\,]$ indicates vertical concatenation. No activation $\sigma$ is applied after this layer.

The second layer has its weight matrices set to:

- $U_{r,t,2}^2 = U_{t,2}^2 = \boldsymbol{\kappa}_{:d}^{:d}$

- $U_{r,t,2}^4 = U_{t,2}^4 = \boldsymbol{\kappa}_{:d}^{d+1:2d}$

and all other weight matrices $U$ to $\mathbf{0}$.

[*NGNN*] layers perform independent message passing on each subgraph. `SUN` implements these as shown for DS-GNN. □

# E   Future research directions

The following are promising directions for future work:

1. *Extension to higher-order node policies.* The prior works of Cotta et al. [14], Papp et al. [43] suggested using more complex policies that depend on tuples of nodes rather than a single node. Since there are exactly $n^k$ distinct $k$-tuples, and each subgraph is defined by a second-order adjacency tensor, we conjecture that Subgraph GNNs applied to such policies are bounded by `(k+2)-WL`. See Appendix F for additional details.

2. *Beyond `3-WL`.* Our results suggest two directions for breaking the 3-WL representational limit: (i) Using policies not computable by `3-IGNs` (ii) Using higher-order node-based policies as mentioned above.

3. *Layers vs. policies.* We make an interesting observation regarding the relationship between layer structure and subgraph selection policies: Having a non-shared set of parameters for root and non-root nodes, `SUN` may be capable of learning the policies $\pi_{\mathrm{NM}}, \pi_{\mathrm{ND}}$ by itself. This raises the question of whether we should let the model learn a policy or specify one in advance.

4. *Lower bound on `SUN` and `ReIGN(2)`.* In this work we have proved a `3-WL` *upper bound* on the expressive power of SUN, `ReIGN(2)` and other node-based Subgraph GNNs by showing their computation on a given graph pair can be simulated by a `3-IGN`. It is natural to ask whether a (tighter) *lower bound* exists as well. In this sense, it is reasonable to believe that node-based Subgraph GNNs are not capable of implementing 3-IGNs, as they inherently operate on a second-order object. However, this does not necessarily imply these models are less expressive than `3-IGNs`, when considering graph separation. For example, they may still be able to distinguish between the same pairs of graphs distinguished by 3-IGNs, hence attaining 3-WL expressive power. It is because of this reason that we believe studying the expressivity gap between `ReIGN(2)` (or any of the subsumed Subgraph GNNs) and `3-WL` represents an interesting open question that could be addressed in future work.

5. *Expressive power of subgraph selection policies.* Another interesting direction for future work would be to better characterise the impact of subgraph selection policies on the expressive power of Subgraph GNNs. Bevilacqua et al. [7] already showed that the DS-GNN model can distinguish some Strongly Regular graphs in the same family when equipped with edge-deletion policy, but not with node-deletion or depth-$n$ ego-networks [7, Proposition 3]. However, edge-deletion is *not* a node-based policy since subgraphs are not in a bijection with nodes in the original graph. It still remains unclear whether a stratification in expressive power exists amongst node-based policies in particular, and under which conditions – if any – this last holds.

We note that, related to 3. and concurrently to the present work, Qian et al. [46] experiment with directly learning policies by back-propagating through discrete structures via perturbation-based differentiation [41].

# F   Extension to higher-order node policies

Constructing a higher-order subgraph selection policy (Appendix E, direction 1.), amounts to defining *selection function* $f$ on a graph and a $k$-*tuple* of its nodes: For a graph $G \in \mathcal{G}$, the subgraphs of such a policy are obtained as $G_{(v_1, \ldots, v_k)} = f\big(G, (v_1, \ldots, v_k)\big)$. The policy contains a subgraph for each possible tuple $(v_1, \ldots, v_k)$. We refer to such policies as $k$-*order node policies*. The $k$-node deletion policy suggested by Cotta et al. [14] is a natural example as the bag of subgraphs contains all subgraphs that are obtained by removing $k$ distinct nodes from the original graph. Since there are exactly $n^k$ distinct tuples, and each subgraph is defined by a second-order adjacency tensor in $\mathbb{R}^{n^2}$, these bags of subgraphs can be arranged into tensors in $\mathbb{R}^{n^{k+2}}$. Noting that the symmetry of

Table 4: TUDatasets. The top three are highlighted by **First**, **Second**, **Third**.

| Dataset | MUTAG | PTC | PROTEINS | NCI1 | NCI109 | IMDB-B | IMDB-M |
|---|---|---|---|---|---|---|---|
| DCNN [4] | N/A | N/A | 61.3±1.6 | 56.6±1.0 | N/A | 49.1±1.4 | 33.5±1.4 |
| DGCNN [60] | 85.8±1.8 | 58.6±2.5 | 75.5±0.9 | 74.4±0.5 | N/A | 70.0±0.9 | 47.8±0.9 |
| IGN [33] | 83.9±13.0 | 58.5±6.9 | 76.6±5.5 | 74.3±2.7 | 72.8±1.5 | 72.0±5.5 | 48.7±3.4 |
| PPGNs [32] | 90.6±8.7 | 66.2±6.6 | **77.2**±4.7 | 83.2±1.1 | **82.2**±1.4 | 73.0±5.8 | 50.5±3.6 |
| NATURAL GN [15] | 89.4±1.6 | 66.8±1.7 | 71.7±1.0 | 82.4±1.3 | N/A | 73.5±2.0 | 51.3±1.5 |
| GSN [11] | **92.2**±7.5 | **68.2**±7.2 | 76.6±5.0 | 83.5±2.0 | N/A | **77.8**±3.3 | **54.3**±3.3 |
| SIN [10] | N/A | N/A | 76.4±3.3 | 82.7±2.1 | N/A | 75.6±3.2 | 52.4±2.9 |
| CIN [9] | **92.7**±6.1 | **68.2**±5.6 | **77.0**±4.3 | 83.6±1.4 | **84.0**±1.6 | 75.6±3.7 | 52.7±3.1 |
| GIN [55] | 89.4±5.6 | 64.6±7.0 | 76.2±2.8 | 82.7±1.7 | **82.2**±1.6 | 75.1±5.1 | 52.3±2.8 |
| ID-GNN (GIN) [56] | 90.4±5.4 | 67.2±4.3 | 75.4±2.7 | 82.6±1.6 | 82.1±1.5 | 76.0±2.7 | 52.7±4.2 |
| DROPEDGE [48] | 91.0±5.7 | 64.5±2.6 | 73.5±4.5 | 82.0±2.6 | **82.2**±1.4 | 76.5± 3.3 | 52.8± 2.8 |
| DS-GNN (GIN) (ND) [7] | 89.4±4.8 | 66.3±7.0 | **77.1**±4.6 | **83.8**±2.4 | 82.4±1.3 | 75.4±2.9 | 52.7±2.0 |
| DS-GNN (GIN) (EGO) [7] | 89.9±6.5 | 68.6±5.8 | 76.7±5.8 | 81.4±0.7 | 79.5±1.0 | 76.1±2.8 | 52.6±2.8 |
| DS-GNN (GIN) (EGO+) [7] | 91.0±4.8 | 68.7±7.0 | 76.7±4.4 | 82.0±1.4 | 80.3±0.9 | **77.1**±2.6 | 53.2±2.8 |
| DSS-GNN (GIN) (ND) [7] | 91.0±3.5 | 66.3±5.9 | 76.1±3.4 | 83.6±1.5 | **83.1**±0.8 | 76.1±2.9 | **53.3**±1.9 |
| DSS-GNN (GIN) (EGO) [7] | 91.0±4.7 | 68.2±5.8 | 76.7±4.1 | 83.6±1.8 | 82.5±1.6 | 76.5±2.8 | **53.3**±3.1 |
| DSS-GNN (GIN) (EGO+) [7] | 91.1±7.0 | **69.2**±6.5 | 75.9±4.3 | 83.7±1.8 | 82.8±1.2 | **77.1**±3.0 | 53.2±2.4 |
| GIN-AK+ [61] | 91.3±7.0 | 67.8±8.8 | **77.1**±5.7 | **85.0**±2.0 | N/A | 75.0±4.2 | N/A |
| **SUN (GIN) (NULL)** | 91.6±4.8 | 67.5±6.8 | 76.8±4.4 | 84.1±2.0 | 83.0±0.9 | 76.2±1.9 | 52.6±3.2 |
| **SUN (GIN) (NM)** | 91.0±4.7 | 67.0±4.8 | 75.7±3.4 | 84.2±1.5 | **83.1**±1.5 | 76.1±2.9 | **53.1**±2.5 |
| **SUN (GIN) (EGO)** | **92.7**±5.8 | 67.2±5.9 | 76.8±5.0 | 83.7±1.3 | 83.0±1.0 | **76.6**±3.4 | 52.7±2.3 |
| **SUN (GIN) (EGO+)** | 92.1±5.8 | 67.6±5.5 | 76.1±5.1 | 84.2±1.5 | **83.1**±1.0 | 76.3±1.9 | 52.9±2.8 |

these tensors can be naturally defined by the diagonal action of $S_n$ on $\{1, \ldots, n\}^{k+2}$ we raise the following generalisation of Corollary 7:

**Conjecture 1.** *Subgraph GNNs equipped with $k$-order node-deletion, $k$-order node-marking or $k$-order ego-networks policies are bounded by $(k + 2)$-WL.*

We believe that proving this conjecture can be accomplished by following the same steps as our proof, i.e., by showing that `(k+2)-IGN` can implement the bag and the update steps of Subgraph GNNs. We also note that `ReIGN(k)`, a higher order analogue of `ReIGN(2)`, can be obtained by following the steps in Section 6. We leave both directions for future work.

We end this section by noting that the statement in our conjecture is considered in a work concurrent to ours by Qian et al. [46].

## G   Experimental details and additional results

Table 5: Test mean metric on the Graph Properties dataset. All Subgraph GNNs employ a GIN base-encoder.

| Method | Graph Properties ($\log_{10}$(MSE)) | | |
|---|---|---|---|
| | IsConnected | Diameter | Radius |
| GCN [27] | -1.7057 | -2.4705 | -3.9316 |
| GIN [55] | -1.9239 | -3.3079 | -4.7584 |
| PNA [13] | -1.9395 | -3.4382 | -4.9470 |
| PPGN [32] | -1.9804 | -3.6147 | -5.0878 |
| GNN-AK [61] | -1.9934 | -3.7573 | -5.0100 |
| GNN-AK-CTX [61] | -2.0541 | -3.7585 | -5.1044 |
| GNN-AK+ [61] | -2.7513 | -3.9687 | -5.1846 |
| SUN (EGO) | -2.0001 | -3.6671 | -5.5720 |
| SUN (EGO+) | -2.0651 | -3.6743 | -5.6356 |

### G.1   Additional experiments

**TUDatasets.** We experimented the performances of `SUN` on the widely used datasets from the TUD repository [37], and include a comparison of different subgraph selection policies. Marking a first,

Table 6: Test mean and std for the corresponding metric on the synthetic tasks. A comparison with other methods can be found in Tables 1 and 5.

| Method | Counting Substructures (MAE) | | | |
|--------|---------|-------------|------|---------|
|        | Triangle | Tailed Tri. | Star | 4-Cycle |
| SUN (EGO) | 0.0092±0.0002 | 0.0105±0.0010 | 0.0064±0.0006 | 0.0140±0.0014 |
| SUN (EGO+) | 0.0079±0.0003 | 0.0080±0.0005 | 0.0064±0.0003 | 0.0105±0.0006 |

| Method | Graph Properties ($\log_{10}$(MSE)) | | |
|--------|-------------|----------|--------|
|        | IsConnected | Diameter | Radius |
| SUN (EGO) | -2.0001±0.0211 | -3.6671±0.0078 | -5.5720±0.0423 |
| SUN (EGO+) | -2.0651±0.0533 | -3.6743±0.0178 | -5.6356±0.0200 |

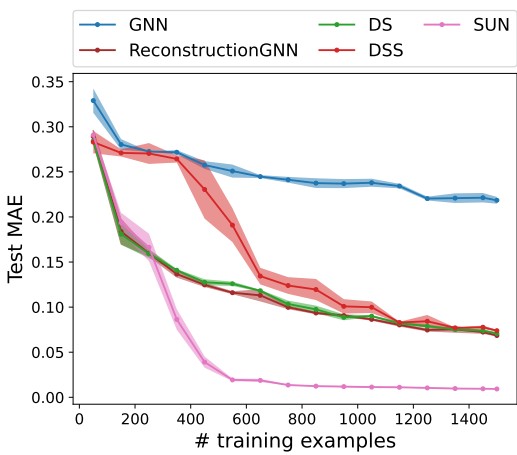

Figure 7: Generalisation capabilities of Subgraph GNNs in the counting prediction task with the node-marking (NM) selection policy.

preliminary, step in the future research direction (3) in Appendix E, we also experiment with a 'NULL' policy, i.e. by constructing bags by simply replicating the original graph $n$ times, without any marking or connectivity alteration. Results are reported in Table 4. Notably, the EGO policies obtain the best results in 6 out of 7 datasets, while the NULL policy does not seem an advantageous strategy on this benchmarking suite. On average, SUN compares well with best performing approaches across domains, while featuring smaller result variations w.r.t. to GNN-AK+ [61].

**Synthetic – Graph property prediction.** Table 5 reports mean test $\log_{10}$(MSE) on the Graph Properties dataset. SUN achieves state-of-the art results on the "Radius" task, where each target is defined as the largest (in absolute value) eigenvalue of the graph's adjacency matrix. Table 6 gathers the standard deviation on the results for these benchmarks as well as "Counting Substructures" ones over 3 seeds as in Zhao et al. [61].

**Generalisation from limited data – Node Marking.** Figure 7 tests the generalisation abilities of Subgraph GNNs on the 4-Cycles task using the node-marking selection policy (NM). Similarly to Figures 4a and 4b, SUN outperforms all other Subgraph GNNs by a large margin and, except for a short initial phase where all Subgraph architectures perform similarly, SUN generalises better.

**Analysis of generalisation.** The GNN's poor performance on this set of experiments may be due to different reasons, e.g. underfitting vs. overfitting behaviours. In this sense, here we deepen our understanding on the 4-Cycles counting task by additionally inspecting characteristics of the train samples and performance thereon. Table 7 reports evaluation results on training and validation sets at the epoch of best validation performance, as well as the best overall training performance. We include the GNN and SUN models, along with a trivial predictor which always outputs the mean training target.

Table 7: Performances for the 4-Cycles counting task. The Trivial Predictor always outputs the mean training target.

|  | Best Train | Train | Val | Test |
|---|---|---|---|---|
| Trivial Predictor | 0.9097 | 0.9097 | 0.9193 | 0.9275 |
| GIN [55] | 0.0283±0.0032 | 0.1432±0.0526 | 0.2148±0.0051 | 0.2185±0.0061 |
| SUN (EGO+) | 0.0072±0.0002 | 0.0072±0.0001 | 0.0097±0.0005 | 0.0105±0.0002 |

Table 8: Test results on ZINC dataset (GIN base-encoder). Each row reports a particular ablation applied on top of the ones in the upper rows.

| Method | ZINC (MAE ↓) | |
|---|---|---|
|  | EGO | EGO+ |
| **SUN** | 0.083±0.003 | 0.084±0.002 |
| w/o $x_i^{i,(t)}, x_k^{k,(t)}$ | 0.089±0.004 | 0.089±0.002 |
| $\theta_1 = \theta_2$ | 0.093±0.003 | 0.093±0.004 |
| w/o $\sum_j x_j^{k,(t)}$ | 0.093±0.004 | 0.090±0.004 |
| w/o $\sum_h x_i^{h,(t)}, \sum_{j \sim i} \sum_h x_j^{h,(t)}$ | 0.111±0.005 | 0.101±0.007 |

First, we observe that the GNN exhibits a relatively large gap between the two reported training MAEs if compared to SUN. This rules out scenarios of complete underfitting, especially considering the trivial predictor performs much worse than the GNN. This led us to evaluate the expressiveness class required to disambiguate all training and test samples: out of all possible graph pairs, we only found one not distinguished by a 1-WL test running 6 colour refinement rounds — same as the number of message passing layers in our GNN. As a consequence, the GNN baseline can effectively assign unique representations to almost all graphs, this justifying its superior performance w.r.t. the trivial predictor. Yet, SUN achieves much better results on all sets, while displaying a smaller train-test gap. This is an indication that, although the hypothesis class of the GNN is sufficiently large to avoid underfitting, it renders the overall learning procedure difficult, leading to suboptimal solutions and partial memorisation phenomena. These results are in line with the observations in Cotta et al. [14, Appendix G.1], where the authors have performed a similar analysis on real-world benchmarks.

**Ablation study.** To assess the impact of the terms in the SUN layer, we perform an ablation study by making sequential changes to Equations (5) and (6) until recovering an architecture similar to NGNN, DS-GNN. We considered the ZINC-12k molecular dataset, using GIN as base graph encoder. Table 8 reports the performances for the EGO and EGO+ policies. As it can be seen, each ablation generally produces some performance degradation, with the removal of $\sum_j x_j^{k,(t)}$ having no significant impact (EGO policy) or even being beneficial when the other changes are made (EGO+ policy). Interestingly, in the EGO+ policy case, although root nodes are explicitly marked, the architecture seems to still benefit from not sharing parameters between root and non-root updates. Indeed, imposing the weight sharing $\theta_1 = \theta_2$ deteriorates the overall performance, which gets similar to the one obtained for the EGO policy. These results indicates that, in the SUN layer, most of the terms concur to the strong empirical performance of the architecture, including the choice of not sharing parameters between root and non-root updates.

### G.2 Experimental details

We implemented our model using Pytorch [44] and Pytorch Geometric [18] (available respectively under the BSD and MIT license). We ran our experiments on NVIDIA DGX V100, GeForce 2080, and TITAN V GPUs. We performed hyperparameter tuning using the Weight and Biases framework [8]. The time spent on each run depends on the dataset under consideration, with the largest being ogbg-molhiv which takes around 8 hours for 200 epochs and asam optimizer. The time for a single ZINC run is 1 hour and 10 minutes for 400 epochs. SUN uses the mean aggregator for the feature matrix and directly employs the adjacency matrix of the original graph as the aggregated adjacency (Equations (5) and (6)). We used the sum aggregator for all the other terms. Unless

otherwise specified, SUN uses the following update equations:

$$x_v^{v,(t+1)} = \sigma\Big(\mu_{t,r}^2\big(x_v^{v,(t)}\big) + \mu_{t,r}^3\big(\sum_w x_w^{v,(t)}\big) +$$
$$+ \gamma_{t,r}^0\big(x_v^{v,(t)}, \sum_{w\sim_v v} x_w^{v,(t)}\big) + \gamma_{t,r}^1\big(\sum_h x_v^{h,(t)}, \sum_{w\sim v}\sum_h x_w^{h,(t)}\big)\Big) \tag{42}$$

$$x_v^{k,(t+1)} = \sigma\Big(\mu_t^0\big(x_v^{v,(t)}\big) + \mu_t^1\big(x_k^{k,(t)}\big) + \mu_t^2\big(x_v^{k,(t)}\big) + \mu_t^3\big(\sum_w x_w^{k,(t)}\big) +$$
$$+ \gamma_t^0\big(x_v^{k,(t)}, \sum_{w\sim_k v} x_w^{k,(t)}\big) + \gamma_t^1\big(\sum_h x_v^{h,(t)}, \sum_{w\sim v}\sum_h x_w^{h,(t)}\big)\Big) \tag{43}$$

where $\mu$'s are two-layer MLPs and each $\gamma$ consists of one GIN [55] convolutional layer whose internal MLP matches the dimensionality of $\mu$'s, e.g.,

$$\gamma_t^0\big(x_v^{k,(t)}, \sum_{w\sim_k v} x_w^{k,(t)}\big) = \hat\mu_t^0\Big((1+\epsilon)x_v^{k,(t)} + \sum_{w\sim_k v} x_w^{k,(t)}\Big)$$

where $\hat\mu_t^0$ is an MLP. Details on the hyperparameter grid and architectural choices specific for each dataset are reported in the following subsections.

### G.2.1 Synthetic datasets

We used the dataset splits and evaluation procedure of Zhao et al. [61]. We considered a batch size of 128 and used Adam optimiser with a learning rate of 0.001 which is decayed by 0.5 every 50 epochs. Training is stopped after 250 epochs. We used GIN as base encoder, and tuned the number of layers in $\{5, 6\}$, and the embedding dimension in $\{64, 96, 110\}$. The depth of the ego-networks is set 2 and 3 in, respectively, the Counting and Graph Property tasks, in accordance with Zhao et al. [61]. Results of existing baselines reported in Tables 1 and 5 are taken from Zhao et al. [61].

### G.2.2 ZINC-12k

We used the same dataset splits of Dwivedi et al. [17], and followed the evaluation procedure prescribed therein. We used Mean Absolute Error as training loss and evaluation metric. We considered batch size of 128, and Adam optimizer with initial learning rate of 0.001 which is decayed by 0.5 after the validation metric does not improve for a patience that we set of 40 epochs. Training is stopped after the learning rate reaches the value of 0.00001, at which time we compute the test metric. We re-trained all Subgraph GNNs to comply with the 500k parameter budget, and also to the above standard procedure in the case of GNN-AK and GNN-AK-ctx. For GNN-AK+, we reported the result 0.086±??? specified by the authors in the rebuttal phase on Openreview, where the question marks indicate that the standard deviation was not provided. We also re-ran GNN-AK+ with the aforementioned standard procedure (learning rate decay and test at the time of early stopping) and obtained $0.091 \pm 0.011$. All Subgraph GNNs use 6 layers and ego-networks of depth 3. We use GIN as the base encoder and we set the embedding dimension to 128 for NGNN, DS- and DSS-GNN, to 100 for GNN-AK variants and to 64 for SUN. DS-GNN employs invariant deep sets layers [58] of the form $\rho(\frac{1}{n}\sum_{i=1}^n \phi(x_i))$ where $x_i$ denotes the representation of subgraph $i$. We tuned $\phi$ and $\rho$ to be either a 2-layers MLP or a single layer with dimensions in $\{64, 128\}$. All other parameters are left as in the original implementation of the corresponding method. We repeat the experiments with 10 different initialisation seeds, and report mean and standard deviation.

### G.2.3 OGBG-molhiv dataset

We used the evaluation procedure proposed in Hu et al. [23], which prescribes running each experiment with 10 different seeds and reporting the results at the epoch achieving the best validation metric. Following Zhao et al. [61], we disabled the subgraph aggregation components $\mu_{t,r}^3\big(\sum_w x_w^{v,(t)}\big)$ and $\mu_t^3\big(\sum_w x_w^{k,(t)}\big)$ in Equations (42) and (43). We used the same architectural choices of Zhao et al. [61], namely depth-3 ego-networks, 2 GIN layers, residual connections and dropout of 0.3. We set the embedding dimension of the GNNs to be 64. Early experimentation with the common Adam optimiser revealed large fluctuations in the validation metric, which we found to considerably

oscillate across optimisation steps even for small learning rate values. Thus, given the non-uniform strategy adopted to generate train, validation and test splits, we considered employing the ASAM optimiser [29]. ASAM considers the sharpness of the training loss in each gradient descent step, effectively driving the optimisation towards flatter minima. We left its $\rho$ parameter to its default value of $0.5$. Additionally, to further prevent overfitting, we adopted linear layers in place of MLPs, as shown in Equations (44) and (45). These choices showed to greatly reduce the aforementioned fluctuations. Finally, we tuned the learning rate in $\{0.01, 0.005\}$ and the batch size in $\{32, 64\}$. The result in Table 2 corresponds to the configuration attaining best overall validation performance (ROC AUC $85.19 \pm 0.82$), with a batch size of 32 and a learning rate of $0.01$. We note that other configurations performed comparably well. Amongst others, the configuration with a batch size of 64 and a learning rate of $0.005$ attained a Test ROC AUC of $80.41 \pm 0.76$ with a Validation ROC AUC of $84.87 \pm 0.55$. We remark how these SUN configurations perform comparably well when contrasted with state-of-the-art GNN approaches which explicitly model (molecular) rings, crucially, *both* on test and validation sets, despite the non-uniform splitting procedure. As an example, CIN [9] reports a Test ROC AUC of $80.94 \pm 0.57$ with a Validation ROC AUC of $82.77 \pm 0.99$.

$$x_v^{v,(t+1)} = \sigma\Big(U_{t,r}^2 \cdot x_v^{v,(t)} + \gamma_{t,r}^0\big(x_v^{v,(t)}, \sum_{w \sim_v v} x_w^{v,(t)}\big) + \gamma_{t,r}^1\big(\sum_h x_v^{h,(t)}, \sum_{w \sim v} \sum_h x_w^{h,(t)}\big)\Big) \quad (44)$$

$$\begin{aligned} x_v^{k,(t+1)} = \sigma\Big(&U_t^0 \cdot x_v^{v,(t)} + U_t^1 \cdot x_k^{k,(t)} + U_t^2 \cdot x_v^{k,(t)} + \\ &+ \gamma_t^0\big(x_v^{k,(t)}, \sum_{w \sim_k v} x_w^{k,(t)}\big) + \gamma_t^1\big(\sum_h x_v^{h,(t)}, \sum_{w \sim v} \sum_h x_w^{h,(t)}\big)\Big) \end{aligned} \quad (45)$$

### G.2.4 TUDatasets

We followed the evaluation procedure described in Xu et al. [55]. We conducted 10-fold cross validation and reported the performances at the epoch achieving the best averaged validation accuracy across the folds. We used the same hyperparameter grid of Bevilacqua et al. [7]. We used GIN as base encoder, setting the number of layers to 4 and tuning its embedding dimension in $\{16, 32\}$. We used Adam optimizer with batch size in $\{32, 128\}$, and initial learning rate in $\{0.01, 0.001\}$, which is decayed by $0.5$ every 50 epochs. Training is stopped after 350 epochs. All ego-networks are of depth 2.

### G.2.5 Generalisation from limited data

We select each architecture by tuning the hyperparameters with the entire training and validation sets, and choosing the configuration achieving the best validation performances. The hyperparameter grids for SUN are the ones in Appendices G.2.1 and G.2.2. In the 4-Cycles task, for NGNN, DS- and DSS-GNN we used the same grid but we tuned the embedding dimension in $\{64, 128, 256\}$ to allow them to have a similar number of parameters as SUN. For GNN-AK variants we used the best performing parameters as provided in Zhao et al. [61].

### G.2.6 Ablation study

For every ablation we tuned the embedding dimension in $\{64, 96, 110, 128\}$ and chose the model obtaining the lowest validation MAE while still being complaint with the 500K parameter budget. The evaluation procedure and all the other hyperparameters are as specified in Appendix G.2.2.