# OpenReview forum: "Understanding and Extending Subgraph GNNs by Rethinking Their Symmetries"
_NeurIPS.cc/2022/Conference — NeurIPS 2022 Accept_

### Official Review · Reviewer_qAH9 · 2022-06-25

**Rating:** 7
**Confidence:** 4
**Soundness:** 4 excellent
**Presentation:** 3 good
**Contribution:** 4 excellent

**Summary:**

This paper proposes a unified analysis of subgraph GNNs based on node selection (e.g., ego-networks, node deletion, etc), and shows that these GNNs map directly to 3-IGNs (invariant graph networks) by representing their different components (subgraph selection, layers, pooling, MLP) as 3-IGN layers. Based on this correspondence, the paper then shows that any subgraph GNN based on node selection (where every subgraph is computed through a bijection over the selected node and the original input graph) can be implemented through a 3-IGN. This result is shown first by proving that all known graph selection policies can be emulated by a 3-IGN (Lemma 4) and then showing how the following components of subgraph GNNs can also be captured by this same model (Lemma 5). Based on this result, the paper proves that all subgraph GNNs based on node selection have expressive power upper-bounded by 3-WL, the upper bound for 3-IGNs.

Building on this insight, the paper then considers potentially novel designs for invariant/equivariant operations over nodes and subgraphs, and draws on the potential operations of 3-IGN and 2-IGN. In particular, it looks for equivariant functions with an at-most quadratic memory footprint (like 2-IGN) and then extends 2-IGNs with local node neighborhood aggregation (as standard in MPNNs) so as to naturally emulate subgraph GNNs. This extension, called ReIGN, is sufficient to capture all known subgraph GNNs. Finally, the paper proposes subgraph union networks (SUNs), which build on the ReIGN framework, and evaluate this model empirically on subgraph counting benchmarks, demonstrating strong performance.

**Questions:**

None at the moment

**Limitations:**

Limitations: The model properties are thoroughly discussed in this paper, and the corresponding limitations of ReIGN are clear. However, SUN could have been better  analyzed in terms of the limitations stemming from its model choices (See ''Weaknesses" section for more details.)

Societal Impact: Not applicable

**Strengths And Weaknesses:**

Strengths:
- The unified analysis of subgraph GNNs is very valuable, and the upper-bound is quite useful to understand the limitations of current models. I have briefly checked the 3-IGN construction in the appendix, and all proofs appear sound.
- The proposed ReIGN framework offers interesting avenues to extend subgraph GNNs, and the coverage of existing models is very good.

Weaknesses:
- The experimental section is rather limited, as the paper makes no ablation analyses of the SUN model, nor does it conduct case studies to further validate its hypotheses. I believe an interesting question is how SUN performance changes relative to the different components in its equations (i.e., how much gain comes from, e.g., using different update functions v_theta1, v_theta2, using all the different components fed into the update equation).
- The intuition provided in the paper is not sufficient to fully understand the result. In particular, I had to consult the appendix on multiple occasions to understand orbits and the constructions for the proof. Therefore, I strongly recommend a slightly more detailed coverage of the 3-IGN construction and of orbits to facilitate the explanation
-  It seems that the font used in the submission differs from what is expected from the NeurIPS template: To demonstrate, the "Do not distribute" footer stretches into a second line, which isn't the case in the normal template. This could well be a bug, and clearly is to the disadvantage of the authors (i.e., the font is less generous in terms of space than it should be). Hence, I strongly suggest you revisit your source file. The added space could help you more clearly explain your proofs as suggested above.

---

> ### Author Response · Authors · 2022-08-02
> **Official author response to Reviewer qAH9**
>
> We are delighted to see the Reviewer recognised the value in our theoretical contribution as well as in the framework we proposed for the design of novel Subgraph GNN layers. At the same time, the Reviewer raised important points we address in the following.
>
> _**“The experimental section is rather limited, as the paper makes no ablation analyses” / “SUN could have been better analyzed in terms of the limitations stemming from its model choices”**_
>
> We agree with the Reviewer that a principled ablation analysis could be beneficial in understanding the impact of each term in the update equations of the SUN layer: we have proceeded by performing the study we detail in the following.
>
> We considered the ZINC molecular dataset, using GIN as base graph encoder, and we made sequential changes to the SUN layer equations (Equations 5 and 6) until recovering an architecture similar to NGNN, DS. We performed hyperparameter tuning for every change, while maintaining the 500K parameter budget. The table below reports the test performance for the EGO policy.
>
> As it can be seen, each ablation generally produces some performance degradation, with the removal of $\sum_{j} x^{k,(t)}_{j}$ having no significant impact.
>
> | Method                                                                       | ZINC (MAE $\downarrow$) |
> |----------------------------------------------------------------------|:----------------------------------:|
> | SUN                                                                                              | 0.083 ± 0.003                  |
> | w/o $x_{i}^{i,(t)}, x_{k}^{k,(t)}$                                                       | 0.089 ± 0.004                  |
> | $\theta_1 = \theta_2$                                                                    | 0.093 ± 0.003                  |
> | w/o $\sum_{j} x^{k,(t)}_{j}$                                                            | 0.093 ± 0.004                  |
> | w/o $\sum_{h} x_{i}^{h,(t)}, \sum_{j \sim i} \sum_{h} x_{j}^{h,(t)}$ | 0.111 ± 0.005                  |
>
>
> The next table reports the test results obtained for the EGO+ policy.  Interestingly, although root nodes are explicitly marked, the architecture seems to still benefit from not sharing parameters between root and non-root updates: imposing the weight sharing $\theta_1 = \theta_2$, deteriorates the overall performance, which gets similar to the one obtained for the EGO policy. In this case, term $\sum_{j} x^{k,(t)}_{j}$ is even proved to be detrimental when the other changes are made.
>
> | Method                                                                       | ZINC (MAE $\downarrow$) |
> |----------------------------------------------------------------------|:----------------------------------:|
> | SUN                                                                                              | 0.084 ± 0.002                  |
> | w/o $x_{i}^{i,(t)}, x_{k}^{k,(t)}$                                                       | 0.089 ± 0.002                  |
> | $\theta_1 = \theta_2$                                                                    | 0.093 ± 0.004                  |
> | w/o $\sum_{j} x^{k,(t)}_{j}$                                                            | 0.090 ± 0.004                  |
> | w/o $\sum_{h} x_{i}^{h,(t)}, \sum_{j \sim i} \sum_{h} x_{j}^{h,(t)}$ | 0.101 ± 0.007                  |
>
> Overall, this ablation analysis indicates that, in the SUN layer, most of the terms concur to the strong empirical performance of the architecture, including the choice of not sharing parameters between root and non-root updates. We will make sure to include this analysis in the next paper revision.
>
> _**“I strongly recommend a slightly more detailed coverage of the 3-IGN construction and of orbits”**_
>
> We agree with the recommendation of the Reviewer. In a way compatible with space limitations, in the next revision of our manuscript we will make our best to introduce more thoroughly those aspects underpinning our theoretical results and proofs. We will try to better introduce 3-IGNs by describing their equivariant layers, how orbits partition the cubed tensor they are applied onto and their semantics when the tensor is interpreted as a node-based bag of subgraphs. Finally, we will properly refer readers to [Morris et al., 2021], which includes an articulated review of Invariant Graph Networks.
>
> _**“I strongly suggest you revisit your source file [...] clearly is to the disadvantage of the authors”**_
> After a careful check we found out that, likely due to a human error, we imported an unwanted package in our latex source code. As the Reviewer hypothesised, because of this, the current font is actually slightly more space-greedy, and fixing this problem will indeed free up some useful space which we will employ to address the comment above. Thanks!
>
> _**References**_
>
> [Morris et al., 2021] “Weisfeiler and Leman go Machine Learning: The Story so far”

---

### Official Review · Reviewer_F5Bg · 2022-07-09

**Rating:** 7
**Confidence:** 4
**Soundness:** 3 good
**Presentation:** 3 good
**Contribution:** 3 good

**Summary:**

This work provides a theoretical framework for subgraph GNNs. It first points out that the theoretical analysis of subgraph GNN with node-based subgraph policy can be simplified. Then, with the existing invariant graph networks (IGN) model, the author embeds node-based subgraph GNN into the 3-IGN and thus bounds the expressive power of node-based subgraph GNNs by 3-WL. Besides expressivity analysis, this work also provides a detailed illustration of the design space of subgraph GNN and a new subgraph GNN model, SUN. SUN exhibits good generalization ability in experiments.

**Questions:**

1. How large is the expressivity gap between IGN(3) and ReIGN(2)?

2. Time complexity of IGN(3)?

**Limitations:**

The societal impact is adequately addressed.

**Strengths And Weaknesses:**

Strength:
1. The paper is clear and well-written.
2. The connection between subgraph GNN and 3-WL is novel.
3. The topic is significant as using subgraph is an important method to boost expressivity.

Weakness:
1. The discussion is heavily based on IGN. However, this paper does not introduce it.
2. The design space of subgraph GNN is based on extended IGN(2) and has little connection to the theoretical analysis based on IGN(3).
3. Time complexity is not discussed.
4. Discussing how subgraph selection rules affect expressivity will also be very interesting.

---

> ### Author Response · Authors · 2022-08-02
> **Official author response to Reviewer F5Bg (1/2)**
>
> We gladly notice that, in their feedback, the Reviewer highlighted the significance of the research topic and the novelty of our approach. They also made some relevant comments and raised a few questions. We address these below.
>
> _**“The discussion is heavily based on IGN. However, this paper does not introduce it.”**_
>
> We thank the reviewer for bringing this point to our attention. We have already expanded on these models in Appendix B, but we agree that a more thorough introduction of Invariant Graph Networks in the main paper would improve the quality of our manuscript.
>
> In the next revision of our manuscript we will make all possible efforts to better describe these models in a way that is compatible with space limitations, giving more emphasis to those aspects with a pivotal role in our results and their proofs, i.e. the structure of IGN equivariant layers and of the tensorial object they process. Additionally, we will make sure to properly refer readers to the recent, comprehensive review of Invariant Graph Networks included in [Morris et al., 2021].
>
>
> _**“The design space of subgraph GNN is based on extended IGN(2) and has little connection to the theoretical analysis based on IGN(3).”**_
>
> In fact, it is exactly our theoretical analysis based on 3-IGNs that sparked the intuition on employing an extended 2-IGN model for a reduced, yet expressive, design space.
>
> In general, 3-IGN layers update all elements in their input third-order tensor. When interpreting these tensors as node-based bags of subgraphs, this corresponds to updating the representations of nodes, edges as well as non-edge node pairs across subgraphs. Subgraph GNNs, on the other hand, typically update only node representations. In the aforementioned third-order tensor, these correspond to only those elements in its main diagonal plane, and it was our intuition that a sensible approach to reduce the layer design space would be to restrict to operations updating these entries only (captured by those orbits which in Appendix B we refer to as $o_{iii}, o_{ijj}$). At the same time, we importantly noticed that this diagonal plane corresponded to a second-order object with symmetries described by the diagonal action of the group $S_n$ over $\mathbb{R}^{n^2}$, the same to which 2-IGNs are equivariant to. Given this context, we deemed it natural to consider 2-IGNs as the main computational framework in this reduced design space, to be further extended to support sparse message passing operations.
>
> Equations 2 and 3 describe the action of the symmetry groups separately over node connectivity ($\mathcal{A}$) and node representations ($\mathcal{X}$), even if, in fact, the two are both embedded in the same third-order tensor 3-IGNs operate on, as described above. This presentation choice has been made as it is customary in the Graph Neural Network community to distinguish these two entities. However, within the scope of the observation made by the Reviewer, we appreciate it may be at the cost of being somewhat deceptive. We will improve this presentation aspect in the next revision of our manuscript, and include a figure to better support the comprehension of the rationale described above.
>
>
> _**“Discussing how subgraph selection rules affect expressivity will also be very interesting.”**_
>
> The Reviewer is raising a very interesting point. However, we believe that to properly enquire into this aspect reasonably falls outside of the scope of the present work.
>
> Nonetheless, it may be interesting to report that some recent works have already marked some initial steps in this direction, that is in trying to characterise the impact of subgraph selection policies on the expressive power of a Subgraph GNN. For example, in [Bevilacqua et al., 2022], the authors have shown how edge-deletion may lead to superior expressive power than node-deletion or ego-network policies: contrary to the latter ones, edge-deletion allows to disambiguate pairs of Strongly Regular graphs in the same family. Notably, edge-deletion is not a node-based policy since subgraphs are not in a bijection with nodes in the original graph. Lastly, we note that the architecture proposed in [Papp and Wattenhofer., 2022] operates by node marking, and the authors showed this approach is strictly stronger than the popular node-deletion policy.

---

> > ### Author Response · Authors · 2022-08-02
> > **Official author response to Reviewer F5Bg (2/2)**
> >
> > _**“How large is the expressivity gap between IGN(3) and ReIGN(2)?”**_
> >
> > This is a very interesting aspect and we thank the reviewer for bringing it up. On one hand, we do not have a definite answer to this precise question; on the other hand, we believe it would be an important direction for future work to understand whether ReIGN(2), or any of the node-based Subgraph GNNs discussed in the paper, are already 3-WL expressive. Given that it intrinsically operates on a second-order object, it is our intuition that ReIGN(2) may not be able to _implement_ 3-IGNs. However, this does not necessarily imply the former model is less expressive than the latter when considering graph separation: ReIGN(2) may still be able to distinguish between the same pairs of graphs distinguished by 3-IGNs, hence attaining 3-WL expressive power. Either way, we believe this aspect would require a more focussed effort we are eager to make in a follow-up work.
> >
> >
> > _**“Time complexity is not discussed” / “Time complexity of IGN(3)?”**_
> >
> > We will discuss these aspects more thoroughly in the next version of our manuscript. The complexity of Subgraph GNNs has been studied in previous works [Bevilacqua et al., 2022; Zhao et al. 2022; Zhang et al., 2021]. For generic node-based subgraph selection policies it amounts to $T(n) = O(n^2 d)$, where $n$ is the number of nodes of an input graph, and $d$ is the maximum node degree. For subgraphs significantly smaller than the original graph (as it may be the case for shallow egonets) it can better be estimated with $T(n) = O(n c d)$ with $c$ the maximum subgraph size.
> >
> > As reported in [Bevilacqua et al., 2022], 3-IGNs have a time complexity which is cubic in the number of nodes in the input graph. Essentially, their equivariant layers update each element in a third-order tensor by means of shared pooling-broadcasting operations whose complexity is upper bounded by $O(n^3)$. We will stress this aspect in our next manuscript revision.
> >
> > _**References**_
> >
> > [Morris et al., 2021] “Weisfeiler and Leman go Machine Learning: The Story so far”
> >
> > [Bevilacqua et al., 2022] “Equivariant Subgraph Aggregation Networks”
> >
> > [Papp and Wattenhofer., 2022] “A Theoretical Comparison of Graph Neural Network Extensions”
> >
> > [Zhang et al., 2021] “Nested Graph Neural Networks”
> >
> > [Zhao et al., 2022] “From Stars to Subgraphs: Uplifting Any GNN with Local Structure Awareness”

---

### Official Review · Reviewer_SuEM · 2022-07-10

**Rating:** 7
**Confidence:** 4
**Soundness:** 4 excellent
**Presentation:** 4 excellent
**Contribution:** 3 good

**Summary:**

This paper extends and analyzes the class of Subgraph GNNs. Importantly, the authors proved that the expressive power of these Subgraph GNNs is bounded by 3-WL. Then, a new family of layers for the class of Subgraph GNNs is proposed, with better generalization abilities. Overall, this is a solid theory paper, and provides a deeper understanding for subgraph GNNs.

**Questions:**

How is the complexity of subgraph GNN?

**Limitations:**

The authors have adequately addressed the limitations and potential negative societal impact of their work.

**Strengths And Weaknesses:**

Strength: 1. Provide a deeper understanding of subgraph GNN
2. Design a novel Subgraph GNN, which unifies previous architectures and provides better empirical performance.
Weakness: 1. Scalability or computation/storage cost is not discussed.
2. The empirical improvement is not impressive

---

> ### Author Response · Authors · 2022-08-02
> **Official author response to Reviewer SuEM**
>
> We are thankful to the Reviewer for their constructive comments. We are pleased to notice they appreciated the solidity of our theoretical contribution as it offers a deeper understanding of Subgraph GNN models. We reply to more specific comments and questions here below.
>
> _**“Scalability or computation/storage cost is not discussed” / “How is the complexity of subgraph GNN?”**_
>
> We thank the reviewer for bringing this point to our attention. We acknowledge the fact that discussing the computational complexity of Subgraph GNNs would improve the quality of our manuscript and we will make sure to include this aspect in the next revision of our paper.
>
> The complexity of Subgraph GNNs has been discussed in previous papers, see e.g. [Bevilacqua et al., 2022; Zhang et al., 2021; Zhao et al., 2022]. In [Bevilacqua et al., 2022], the authors analyse the space and time complexity of a model performing traditional message passing on each of the subgraphs obtained by a generic subgraph selection policy. Let $n, d$ refer to, respectively, the number of nodes and maximum node degree of an input graph. If $b$ is the size of the subgraph bag, the asymptotic time complexity amounts to $O(b n d)$, while the memory complexity to $O(b (n + n d))$. For a node-based selection policy, these become, respectively, $O(n^2 d)$ and $O(n (n + n d))$.
>
> Other than local message passing operations, more sophisticated Subgraph GNNs may include “global” pooling terms in their layer equation: see, e.g. the “subgraph” and “context” econdings in GNN-AK+ [Zhao et al., 2022] or the aggregation over node representations across subgraphs that is operated by DSS-GNN [Bevilacqua et al., 2022]. In principle, these operations require a squared asymptotic computational complexity ($O(n^2)$). However, these terms are shared in the update equations of nodes / subgraphs: in practice, it is only sufficient to perform the computation once. As $T(n) = O(n^2 d + n^2)$ implies $T(n) = O(n^2 d)$, these Subgraph GNNs retain the same asymptotic complexity described above.
>
> SUN layers involve the same “local” message passing and “global” pooling operations. The above considerations are, thus, directly applicable yielding the same asymptotic bounds.
>
> It is worth noting that these bounds can be tightened in the case of ego-network policies. Let $c$ be the maximum ego-network size. The time complexity of a Subgraph GNN equipped with an ego-network policy becomes: $O(n c d)$. As observed in [Zhang et al., 2021], when ego-networks are of limited depth, the size of the subgraphs may be significantly smaller than that of the input graph; in other words $c \ll n$, reducing the overall message passing complexity.
>
> _**“The empirical improvement is not impressive”**_
>
> We respectfully believe that the performance attained by SUN on both synthetic and real-world benchmarks are, in fact, particularly solid. Generally, SUN tends to outperform all previous Subgraph GNNs, and on the ZINC and ogbg-molhiv datasets it approaches or outruns architectures which, contrary to SUN, explicitly model domain-relevant graph substructures (such as rings). This is extremely relevant for example on ZINC: here targets (penalised constrained solubility) linearly depend, amongst other terms, on the number of cycles whose length is at least 6 (see https://arxiv.org/pdf/2003.00982.pdf). SUN is a domain agnostic model, and we believe it is remarkable that, as such, it is able to attain such performance on these competitive and well-studied molecular benchmarks, where improvements are typically marginal.
>
> _**References**_
>
> [Bevilacqua et al., 2022] “Equivariant Subgraph Aggregation Networks”
>
> [Zhang et al., 2021] “Nested Graph Neural Networks”
>
> [Zhao et al., 2022] “From Stars to Subgraphs: Uplifting Any GNN with Local Structure Awareness”

---

### Official Review · Reviewer_GpAd · 2022-07-12

**Rating:** 7
**Confidence:** 2
**Soundness:** 3 good
**Presentation:** 3 good
**Contribution:** 3 good

**Summary:**

This paper focus on understanding the expressiveness of subgraph graph neural networks. Under its proposed node-based subgraph selection policy, the author demonstrates the representation power as strong as 3-WL test via bridging it with 3-IGNs. In the mean time, the algorithm is realized as Re-IGN and reports superior performance than other existing subgraph GNNs.

**Questions:**

1. Compared with traditional GNNs that achieves better performance in benchmark datasets, what's the potential reason? I understand it is not a disadvantage that SUN can not beat GIN/CIN on existing benchmarks? Some insights that why SUN with 1 MLP operators cannot beat existing methods with geometric group symmetries, might be helpful for the community.

2. Do you require pre-defined motifs in node-based selection policies?

**Limitations:**

N.A.

**Strengths And Weaknesses:**

**Strengths**
1. The presentation is clear and easy to follow.
2. Subgraph GNN is a uprising research topic in graph neural networks, theoretical understanding and its symmetries to existing invariant or equivariant GNNs help explain the design choices and applications.
3. The extensive experiments in main paper and appendix not only shows its superior performance in family of subgraph GNNs but also on par performance with established traditional graph neural networks.


**Weaknesses**
Not I can think of. I think the experiment table 1 is missing an entry.

---

> ### Author Response · Authors · 2022-08-02
> **Official author response to Reviewer GpAd**
>
> We thank the Reviewer for their feedback. We are glad they found the paper well-presented, while appreciating the provided theoretical and experimental analyses. We proceed by answering the questions raised by the Reviewer in the following.
>
>
> _**“I think the experiment table 1 is missing an entry”**_
>
> We believe the Reviewer refers to the triple question mark symbol (“???”) reported in place of the standard deviation for the GNN-AK+ model [Zhao et al., 2022]. As explained in Appendix G.2.2, the performance of this method has been reported directly from the authors’ rebuttal comment on Open Review, as the one compliant with the 500k parameter budget (see https://openreview.net/forum?id=Mspk_WYKoEH&noteId=2oeomvjT4eg). As the comment does not report the standard deviation, we used the question mark symbol (“???”).  In our next revision we will include this note directly in the main corpus of the manuscript.
>
>
> _**“Compared with traditional GNNs that achieves better performance in benchmark datasets, what's the potential reason? I understand it is not a disadvantage that SUN can not beat GIN/CIN on existing benchmarks?”**_
>
> We would like to bring to the attention of the Reviewer that SUN demonstrated to significantly outperform the “traditional” GIN model on the ZINC and ogbg-molhiv molecular benchmarks as well as the synthetic subgraph counting ones (see Tables 1 and 2).
>
> GSN [Bouritsas et al., 2022] and CIN [Bodnar et al., 2021] are the only provably expressive, non-traditional MPNN models with clearly superior performance on the ogbg-molhiv benchmark. On ZINC, SUN is outperformed by CIN, but, interestingly, not by GSN. Yet, in these cases, SUN is the best amongst all other baseline models. We believe these are particularly solid results, especially when put into context. While SUN is a domain agnostic model, CIN and GSN _explicitly model ring substructures_, which are known to play a pivotal role in molecular modelling. Because of this reason, we believe these results are particularly promising.
>
>
> _**“Do you require pre-defined motifs in node-based selection policies?”**_
>
> No, we do not. While being provably expressive, Subgraph GNNs can attain empirically competitive performance by employing generic, domain agnostic policies such as ego-networks or node-marking. This flexibility represents one of the main advantages of Subgraph GNNs in general and SUN in particular.
>
> _**References**_
>
> [Zhao et al., 2022] “From Stars to Subgraphs: Uplifting Any GNN with Local Structure Awareness”
>
> [Bouritsas et al., 2022] “Improving Graph Neural Network Expressivity with Subgraph Isomorphism Counting”
>
> [Bodnar et al., 2021] “Weisfeiler and Lehman Go Cellular: CW Networks”

---

### Author Response · Authors · 2022-08-02
**Official general author response**

We are deeply grateful to all reviewers for their feedback and constructive comments, while glad to notice that all Reviewers have positively welcomed our work.

The Reviewers have recognised the validity and importance of our theoretical contribution, considered _“solid”_ (**SuEM**), _“novel”_ (**F5Bg**), _“useful”_ (**qAH9**), and able to provide a _deeper understanding_ of the novel class of Subgraph GNN models (**GpAd**, **SuEM**). The reviewers have stressed the _significance_ of the research topic of Subgraph GNNs (**GpAd**, **F5Bg**), finding that our proposed _“unified analysis of subgraph GNNs is very valuable”_ (**qAH9**). As an important part of our contribution we have proposed a design framework for new Subgraph GNNs (ReIGN); this aspect has been positively appreciated as well. For example, Reviewer **qAH9** has reported this framework _“offers interesting avenues to extend subgraph GNNs”_ while featuring a very good _coverage of existing models_. Last, we are delighted to notice the Reviewers have found our presentation _“clear and easy to follow”_ (**GpAd**) and the paper to be _“well-written”_ (**F5Bg**).

In the next revision of our manuscript we will take into consideration the actionable feedback provided by the Reviewers to enhance the readability and overall quality of the manuscript. As suggested by Reviewers **F5Bg** and **qAH9**, we will make sure to devote a paragraph to better introduce Invariant Graph Networks, the structure of their linear equivariant layers and of the objects they process. Reviewers **SuEM** and **F5Bg** also pointed out how discussing the computational complexity of IGNs and Subgraph GNNs would be an important addition to the paper. We will add this analysis in the new revision of the manuscript, and refer to the relevant bibliographic sources which have previously studied this aspect.

Here below we provide responses to each Reviewer in a way to address their comments in more detail and answer their specific questions.

---

### Author Response · Authors · 2022-08-07
**New paper revision**

We kindly bring the attention of the Reviewers to a new manuscript revision we have just uploaded. The revision implements the additions discussed in the previous general comment and in specific responses to Reviewers.

Changes are visually signalled in _blue_; they include:
- A more thorough and detailed introduction of Invariant Graph Networks as recommended by Reviewers **F5Bg** and **qAH9** – pages 2-3;
- A more explicit reference to the 3-IGN construction and orbits as recommended by Reviewer **qAH9** – page 3 and 5, along with the new Figure 4 at page 16 (Supplementary Materials);
- An analysis of the computational complexity of Subgraph GNNs with reference to 3-IGNs, as recommended by Reviewers **SuEM** and **F5Bg** – page 15 (Supplementary Materials);
- An explicit mention and consideration of relevant directions for future developments of the work: the study of the gap between Subgraph GNNs and 3-WL and of the intrinsic expressive power of node-based policies – page 40 (Supplementary Materials);
- An ablation study on the terms in the SUN layer equations, as recommended by Reviewer **qAH9** – pages 43 and 45 (Supplementary Materials);
- Minor rephrasing of periods in the main corpus of the manuscript to accommodate the changes whilst respecting the given space limitations.

---

### Meta-Review · Area_Chair_sNpg · 2022-08-23

**Recommendation:** Accept
**Confidence:** Certain

**Metareview:**

This paper studies the recent hot topic in GNN, namely subgraph-based GNNs which apply GNN to each node-centered subgraph copy of the original graph instead of directly applying GNN to the full graph. These GNNs were shown to be more expressive than 1-WL but were unknown in terms of their upper bound of expressive power. This paper shows that all these subgraph-based GNNs, including Nested GNN, ID-GNN, reconstruction GNN, GNN-AK etc., can be implemented by 3-IGN which is upper bounded by 3-WL, thus giving an upper bound to subgraph-based GNNs' expressive power. The novel perspective that views subgraphs as an additional tensor dimension which is also equivariant to node permutation is very insightful, and is the key to the 3-IGN implementations. Overall, I believe this paper is of great theoretical contribution to the GNN community and opens up some new design space.

**Award:**

Yes

---

### Decision · Program_Chairs · 2022-09-14

Accept